# CodeClash: Benchmarking Goal-Oriented Software Engineering

**John Yang** [*1]  **Kilian Lieret** [*2]  **Joyce Yang** [3]  **Carlos Jimenez** [2]
**Muhtasham Oblokulov** [4]  **Aryan Siddiqui** [1]  **Ofir Press** [2]  **Ludwig Schmidt** [1]  **Diyi Yang** [1]

## Abstract

Existing coding benchmarks evaluate language models (LMs) on concrete, well-specified tasks such as fixing bugs or writing tests. However, human programmers do not spend all day addressing isolated GitHub issues. Instead, real-world software development is grounded in the pursuit of high-level goals. Evaluating whether LMs can iteratively develop code to accomplish open-ended objectives without explicit guidance remains an open challenge. We introduce Code-Clash, a benchmark where LMs compete in multi-round tournaments to build the best codebase for achieving a competitive objective. Each round proceeds in two parts: agents edit their code, then their codebases compete head-to-head in a code arena that determines winners based on objectives like score maximization, resource acquisition, or survival. Models must decide for themselves how to improve their code both absolutely and against their opponents. We run 1680 tournaments to evaluate 8 LMs across 6 arenas, revealing how models exhibit diverse development styles and share fundamental limitations in strategic reasoning. Models also struggle with long-term codebase maintenance; repositories become progressively messy and redundant. Top models lose every round against expert human programmers. We open-source CodeClash to advance the study of autonomous, goal-oriented code development.

## 1. Introduction

Existing coding benchmarks challenge language models (LMs) to complete small, focused tasks, such as implement-

ing an algorithm (Jain et al., 2024), fixing a specific bug in a single function (Jimenez et al., 2024), or writing a test for a target class (Mündler et al., 2024). Problem statements are straightforward and fine-grained in their description of a task. Given explicit instructions, models are evaluated on their ability to execute them correctly.

On the contrary, real world software development demands a much broader scope of agency. Instead of maintenance tasks, developers are driven by high-level goals like improving user retention, increasing revenue, or reducing costs. This requires fundamentally different capabilities; engineers must recursively decompose objectives into actionable steps, prioritize them, and make strategic decisions about which solutions to pursue. The process is a continuous loop – propose changes, deploy them, analyze feedback (e.g., metrics, user behavior, A/B test results), and repeat to inform the next move. Evaluating how models fare under such conditions remains an unaddressed challenge in benchmarking.

Therefore, we introduce CodeClash, a benchmark for goal-oriented software engineering. Specifically, multiple LM systems compete to build the best codebase for achieving a high-level objective over the course of a multi-round tournament. These codebases implement solutions that compete in a code arena, such as BattleSnake (grid-based survival), Poker (no-limit Texas Hold'em), and RoboCode (tank combat). Crucially, LMs do not play the games directly, unlike existing game-based benchmarks (Silver et al., 2016; OpenAI et al., 2019; Zhang et al., 2025a). Instead, they iteratively refine code that competes as their proxy.

As shown in Figure 1, each round proceeds in two phases: agents edit their code, then their codebases compete head-to-head in a code arena. The code arena then executes multiple implementations against one another and determines winners based on objectives like score maximization, resource acquisition, or survival.

Success in CodeClash requires models to determine their own improvement strategies. From the outset, LM agents receive only a brief description of the setting. While information like arena mechanics, example bots, and recommended strategies are available in the starter codebase, models must take initiative to proactively discover them. Each round, LMs receive gigabytes of logs from past rounds, which

---

[*]Equal contribution  [1]Stanford University, Stanford, USA [2]Princeton University, Princeton, USA [3]Cornell University, Ithaca, USA [4]Technical University of Munich, Munich, Germany. Correspondence to: John Yang <byjohnyang@gmail.com>, Kilian Lieret <klieret@princeton.edu>.

*Proceedings of the $43^{rd}$ International Conference on Machine Learning*, Seoul, South Korea. PMLR 306, 2026. Copyright 2026 by the author(s).

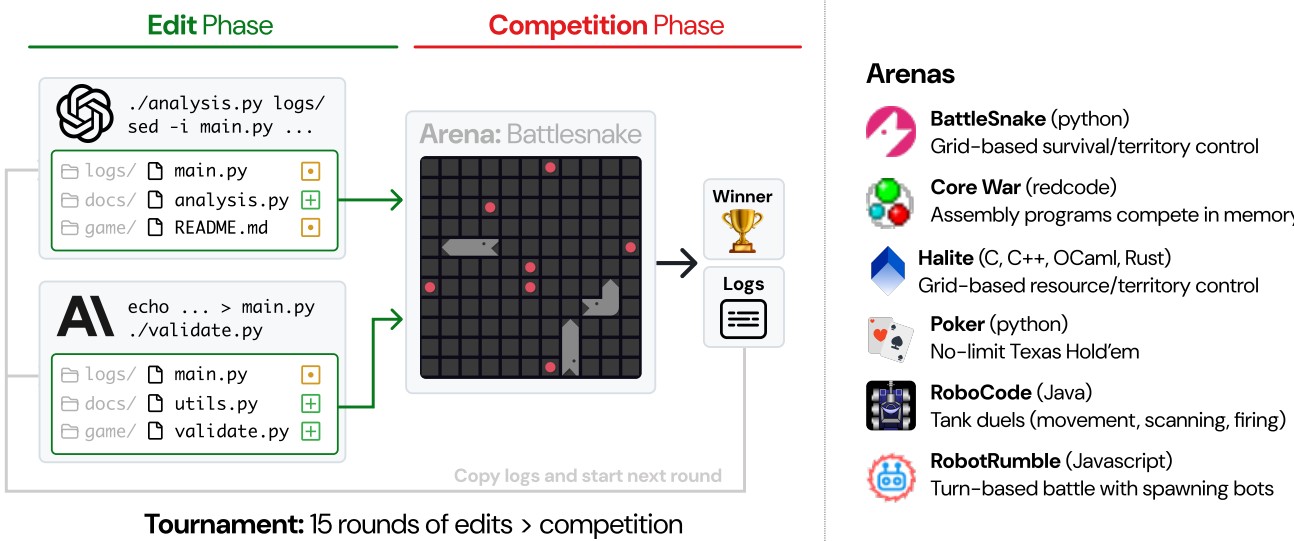

*Figure 1.* CodeClash is a benchmark where players (LMs as SWE-agents) compete in programming tournaments spanning multiple rounds. Per round, models edit their codebases (*edit* phase) before the codebases face off in a code arena (*competition* phase). Then, the competition logs are copied back into the codebases and the next round begins.

they can parse to extract insights about outcomes and opponents – or ignore entirely. Across the span of a tournament, CodeClash reveals whether and how models populate their codebases with notes, tests, and analyses.

We evaluate 8 frontier LMs across 6 arenas. We find Code-Clash elicits substantial creativity from models; across 1680 tournaments, we observe that a model's solutions become increasingly dissimilar round over round, even when facing the same opponent in the same arena. However, our results reveal that while models exhibit diverse development styles, they share common limitations in interpreting competitive feedback, validating changes, and maintaining organized codebases over time. Even top models hallucinate reasons for failure or modify code without confirming if these changes meaningfully improve performance. A substantial gap remains between model and human performance; the best model (Claude Sonnet 4.5) *fails to win a single round* against an expert human-written bot.

We release CodeClash as an open source toolkit to further the study of self-evolving, LM-based SWE-agents.

## 2. CodeClash

### 2.1. Formulation

CodeClash formalizes competitive coding as a tournament, where two or more players compete in a code arena for multiple rounds. *Player* refers to an LM equipped with an Agent Computer Interface (ACI) or scaffold that enables it to interact with a codebase (Yang et al., 2024a). Each player maintains their own codebase for the entire tournament. A

*code arena* is any competition platform that takes in multiple codebases and executes them against one another, producing measurable outcomes about relative performance on a designated objective (e.g., eliminating opponents, acquiring resources, maximizing profit).

Each round proceeds in two phases. In the *edit* phase, each player modifies their own codebase using whatever strategies they deem appropriate within a fixed budget of turns. During the *competition* phase, all codebases are compiled and executed within the code arena, where they interact and compete against each other. The arena determines a winner (or declares a tie) based on the codebases' performance.

CodeClash's formulation makes several key design decisions. *(1) Codebase-as-memory*: players have no explicit memory of actions from previous rounds. Their information is limited to whatever they chose to record in the codebase. *(2) Log-based feedback*: after each competition phase, the results and logs are copied into each player's codebase as the sole source of new information. *(3) Strategic opacity*: players cannot see each other's codebases, though we explore lifting this restriction in Section 4.1.

### 2.2. Technical Details

To implement a player, we use `mini-SWE-agent`, an agent computer interface (ACI) that enables an LM to interact with a codebase by issuing `bash` actions to a terminal (Yang et al., 2024a). Each turn, the LM generates a ReAct (Yao et al., 2023) style response containing a thought (in natural language) and a `bash` action, then receives standard output from the terminal environment in return.

We also define a lightweight, flexible interface for a code arena. An implementation only needs to define commands to run the competition and determine a winner. This minimal overhead enables us to fold many existing competitive programming games and tasks into CodeClash. More in §A.

## 2.3. Features

CodeClash features a suite of 6 code arenas, as listed in Figure 1. Each arena is covered thoroughly in §B. CodeClash introduces several distinctive properties that collectively push models beyond traditional code completion.

**Open-ended objectives.** CodeClash departs from the traditional reliance on unit tests or implementation correctness to measure success. Instead, players code to win competitive outcomes that vary dramatically across arenas, from maximizing profit to surviving the longest. This mirrors the ultimate objectives of real-world software more faithfully, where code is written to achieve practical outcomes (e.g., maximize resources, generate revenue, outperform competitors) rather than simply achieving technical correctness. A consequence of rich objectives is that models must then decompose a higher-order goal into actionable subtasks and measurable, intermediate metrics to inform improvements.

**Diverse arenas.** CodeClash's arenas vary significantly, with drastic differences in a codebase's structure, how a codebase interfaces with the arena engine, and the types of feedback generated. This contrasts sharply with existing benchmarks, where evaluation follows a consistent pattern of problem statement, code implementation, and test validation.

**Adversarial adaptation.** CodeClash's uniquely multiplayer, head-to-head setting adds a new layer of complexity to coding evaluations. While decent LMs may be capable of writing competent implementations, top-performing players will analyze opponent behaviors and incorporate countermeasures while remaining indecipherable in their own play. Early round wins do not ensure continued dominance. At some point, the challenge shifts from writing good code to writing code that consistently beats intelligent competition.

**Self-crafted memory.** As mentioned in Section 2.1, CodeClash does not maintain persistent memory for models across rounds; only ephemeral, within-round memory exists. To retain information for future use, models must explicitly add insights to the codebase; how to represent such knowledge is left entirely to the model's discretion.

**Self-directed improvement.** Beyond a brief description of the environment and arena, the initial system prompt provided to each player at the start of every edit phase contains *no* guidance beyond high level suggestions about how to enhance its codebase. All decisions and changes LMs make are necessarily autonomous. In practice, this may manifest as models writing analysis scripts to understand

competition logs, maintaining notes about past rounds, or generating multiple candidates to test against one another.

## 3. Experiments

**Models.** We select 8 strong LMs to evaluate, where strength is roughly estimated as performance on existing coding benchmarks. Our final list includes two models from the Anthropic family (Claude Sonnet 4.5 (Anthropic, 2025a), 4 (Anthropic, 2025b)), three models from the OpenAI family (GPT-5, GPT-5 mini (OpenAI, 2025a), o3 (OpenAI, 2025b)), Gemini 2.5 Pro (Comanici et al., 2025), Qwen3 Coder (Qwen, 2025), and Grok Code Fast (x.ai, 2025).

**Agent system.** As discussed in Section 2.2, we use `mini-SWE-agent`. We intentionally decide against using tool-heavy scaffolds such as SWE-agent or OpenHands (Wang et al., 2025b), as they are often optimized for models and benchmarks. By restricting interactions to bash commands, `mini-SWE-agent` avoids imposing predefined assumptions via tools about how LMs should approach codebase modifications or competitive play (Yang et al., 2024b). Per round, models are allotted a maximum of 30 turns for the *edit* phase, with automatic termination if exceeded. Player configurations are discussed in §C.1.

**Number of rounds run.** For our main leaderboard, we make models compete one-on-one. Given 8 models and 6 arenas, we run 10 tournaments per model pair per arena, with each tournament lasting 15 rounds. This yields $\binom{8}{2} \times 6 \times 10 \times 15 = 25,200$ total rounds. Tournament runtime varies by arena, taking 75 minutes on average – totaling 2.4 million hours of runtime (mostly due to model latency), parallelized over the independent tournaments. Tournament configuration details are covered in §C.2.

**Win rates.** Performance per model is generally calculated as an aggregation across all tournaments (sets of 15 rounds) won across all arenas. A single round is won by a model if it achieves a higher score in the arena than its opponent or if its opponent makes an invalid submission. A tournament is won by the model that wins more rounds than its opponent, or, if both models win equally many rounds, by the model that scores the last win. The win rate of a model is the fraction of tournaments it has won. For details, see §C.3.

**Elo metrics.** Inspired by the thread of prior work ranking LMs on the task of instruction following (Elo, 1967; Bai et al., 2022; Boubdir et al., 2024; Chiang et al., 2024), we use Elo scores with a base rating of R=1200 and a slope of 400 to quantify the overall strength of each model. Instead of calculating Elo scores using sequential updates (which require a choice of step size and depend on update order), we perform a more rigorous maximum likelihood fit to the win rates. We validate rank stability and our statistical treatment with both parametric and non-parametric boot-

*Table 1.* Elo ratings per model per arena.

| | BattleSnake | CoreWar | Halite | Poker | RoboCode | RobotRumble | **Overall** |
|---|---|---|---|---|---|---|---|
| Claude Sonnet 4.5 | 1470 | 1641 | 1408 | 1248 | 1361 | 1423 | **1389** |
| GPT-5 | 1339 | 1199 | 1522 | 1599 | 1409 | 1293 | **1360** |
| o3 | 1357 | 1348 | 1576 | 1277 | 1338 | 1309 | **1343** |
| Claude Sonnet 4 | 1253 | 1339 | 1111 | 1233 | 1033 | 1361 | **1223** |
| GPT-5 Mini | 1369 | 926 | 1185 | 1429 | 1217 | 1092 | **1200** |
| Gemini 2.5 Pro | 1115 | 1043 | 1186 | 978 | 1315 | 1044 | **1125** |
| Grok Code Fast | 833 | 1170 | 824 | 886 | 1033 | 1016 | **1004** |
| Qwen3 Coder | 860 | 929 | 784 | 945 | 890 | 1057 | **952** |

strapping experiments and observe more than 98% pairwise order agreement. For details, see §C.3.

# 4. Results

We present our main results in Table 1. Claude Sonnet 4.5 stands at the top, followed closely by o3 and GPT-5. After a gap of 100 Elo, the next best models are Claude Sonnet 4 and GPT-5 mini. Notably, no single model dominates across all arenas. Top ranked Claude Sonnet 4.5 places just 4th in Poker, emphasizing the importance of CodeClash's support for multiple arenas. Figure 2 shows win rates of specific matchups. Figure 3 reveals distinct performance trends across rounds – some models excel early before plateauing, while others improve steadily over time.

## 4.1. Ablations

**Models trail substantially behind expert human programmers.** To quantify the gap between models and human experts beyond a single arena and opponent, we introduce CodeClash Ladder (CC:Ladder), a progression-style evaluation where a model climbs a ranked ladder of human-authored solutions, advancing only when it defeats the current opponent. We construct ladders for RobotRumble (58 human solutions) and Core War (264 human solutions), with solutions ranked by Elo from all-pairs matchups. Each model plays 7 rounds against each opponent and advances if it wins a majority of rounds and wins the final round; the codebase carries over between opponents. CC:Ladder retains the core properties of the standard CodeClash evaluation (multi-round iterative development, codebase-as-memory, and log-based feedback) while replacing the model opponent with a static human solution, eliminating opponent variance and substantially reducing cost. Full details on the protocol and human solution rankings are in Appendix D.4.

Table 2 presents CC:Ladder scores for four models from the main leaderboard (Table 1); results for additional models are in Appendix D.4. No model completes either ladder, and a substantial gap remains between models and human

*Table 2.* CC:Ladder results. Models climb a ranked ladder of human-authored solutions, advancing only upon defeating the current opponent. Score = rank of the highest human opponent defeated (best of 5 runs). Column headers show total number of human solutions per ladder. No model completes either ladder.

| Model | RR (58) | Core War (264) |
|---|---|---|
| Claude Sonnet 4.5 | 43 | 205 |
| GPT-5 | 51 | 201 |
| GPT-5 mini | 57 | 260 |
| Gemini 2.5 Pro | 54 | 233 |

RR = RobotRumble, CW = Core War

experts. On RobotRumble, the best result is rank 57 out of 58 (GPT-5 mini); on Core War, rank 260 out of 264 (GPT-5 mini). Most other models plateau considerably earlier. The strategic reasoning limitations identified in Section 5.2, particularly ungrounded conclusions from logs and untested deployments, persist as the dominant causes of ladder termination. Most runs terminate because the model fails to win the final round against an opponent, rather than losing a majority of rounds, suggesting models are often competitive but unable to consolidate gains within the round window.

**Models have limited capacity for opponent analysis even with transparent codebases.** For each pairwise matchup among Claude Sonnet 4.5, GPT-5, and Gemini 2.5 Pro, we run 10 Core War tournaments of 15 rounds each, with one modification – before the *edit* phase of round n, each player receives a read-only copy of their opponent's code from round n-1. While the relative standings remain consistent with the default setting, the win rates change with GPT-5 securing 74.6% (+7.8%) of rounds, Claude Sonnet 4.5 at 53.2% (-1.8%), and Gemini 2.5 Pro at 22.7% (-5.5%). Curiously, GPT-5 only accesses its opponent's codebase in 12.8% of all rounds, far fewer than Claude Sonnet 4.5 (99.3%) and Gemini 2.5 Pro (52.9%), suggesting that frequent inspection of opponent code does not necessarily translate to competitive advantage, as our analysis later in Section 5.2 reaffirms. Additional insights in §D.2.

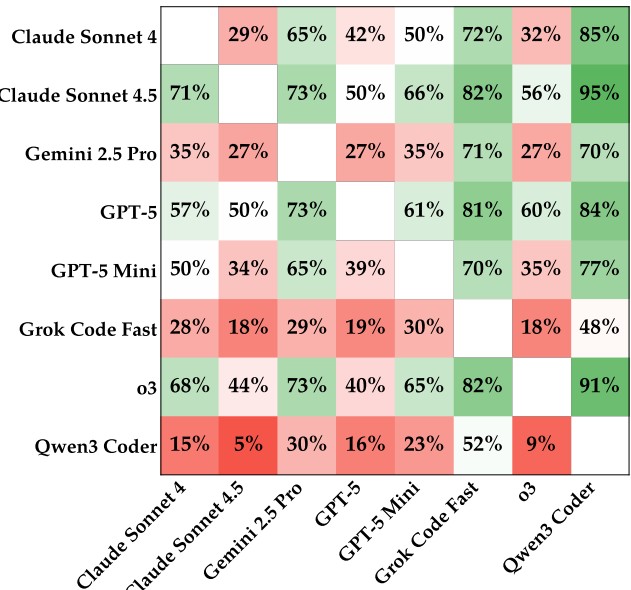

*Figure 2.* Model win rates (row beats column). Win rate is the proportion of tournaments (out of 240) won across all arenas. Claude Sonnet 4.5 has the highest average win rate at 69.9%.

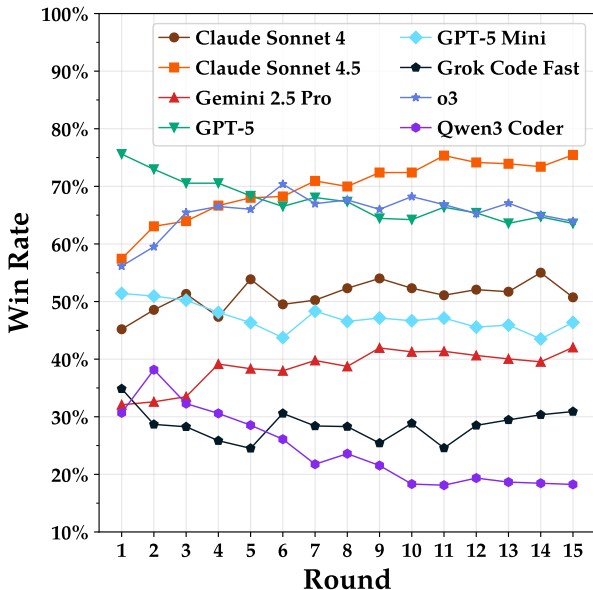

*Figure 3.* Win rates across rounds, illustrating how different models gain (Claude Sonnet 4.5) or lose momentum (GPT-5) over the course of the tournament.

Subsequent studies could more thoroughly investigate and enhance models' capacity for detecting opponents' weaknesses and designing tailored counter-strategies.

**Multi-agent competitions (3+ players) reflect similar rankings.** We run 20 Core War tournaments, 15 rounds each, with 6 of 8 models (excluding GPT-5 mini, Claude Sonnet 4). To quantify performance, as shown in Table 29b, we use the TrueSkill rating system (Herbrich et al., 2006) since Elo and win rate are limited to one-on-one settings. The results are similar to Core War ranks in Table 1, with GPT-5 and Grok Code Fast (two models of similar Elo ranking) switching positions. However, the 6 player tournaments exhibit far more competitive volatility. Lead changes (round n winner different from round n-1) occur 48.4% of the time in 6 player Core War, compared to just 18.2% in the two player setting. Winners of 6-player tournaments capture just 28.6% of total points on average versus 78.0% in 2-player settings. We provide some additional insights in §D. Looking forward, CodeClash's multi-player tournaments offer a promising testbed for future work on coalition dynamics, positional play, and risk management.

## 5. Analysis

### 5.1. Competitive Dynamics

Beyond overall win rates, we analyze how models interact with their codebases along with the resilience of models after losing individual rounds. We also investigate trends in models' solution diversity and codebase organization.

**Models interact with codebases in markedly different ways.** CodeClash's open-ended setting reveals striking differences in how models operate in the *edit* phase. For instance, while o3 and Gemini 2.5 Pro typically only edit an average of 2 files per round, GPT-5 usually changes 5 to 6. The size of edits also varies – on one end, o3 typically adds/removes a total of 51 lines per round, $8\times$ less than Qwen3 Coder or the Claude Sonnet family which usually modify more than 400 lines. Gemini 2.5 Pro stands out as a verbose thinker, generating an average of 105 words per thought, more than double the average. Claude Sonnet 4.5 usually takes 23 of the allotted 30 editing turns per round, whereas GPT-5 and o3 typically concludes after just 15 steps. Distributions visualizing these tendencies in §D.1.

Intriguingly, we did not find any correlations between any of these behaviors and win rates. Both minimalists (o3) and high activity editors (Claude Sonnet 4.5) succeed. Compared to existing benchmarks that terminate upon reaching a solution, CodeClash's multi-round competitive setting makes these distinctions even more salient.

**Even strong models struggle to recover after losing rounds.** In real-world software development, early choices are often made under uncertainty: the best approach might only become clear after testing, real world deployments, and observing competitors. Therefore, the ability to interpret noisy signals and reconsider core design decisions is an important factor for success. The round-based nature of CodeClash exposes how poorly LMs adapt once their initial strategies fail. Figure 4 shows that even for the Claude

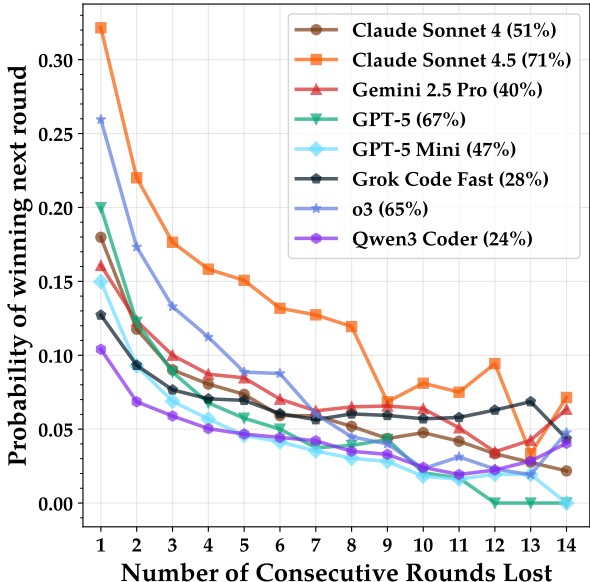

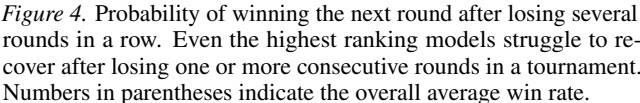

*Figure 4.* Probability of winning the next round after losing several rounds in a row. Even the highest ranking models struggle to recover after losing one or more consecutive rounds in a tournament. Numbers in parentheses indicate the overall average win rate.

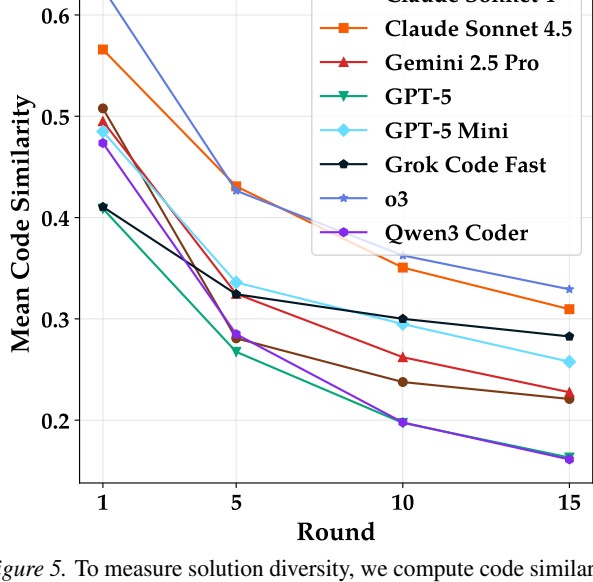

*Figure 5.* To measure solution diversity, we compute code similarity of each model's solutions to itself at the same round. Each data point represents the mean pairwise similarity between a model's solution (`main.py`) at round n across 70 BattleSnake tournaments.

Sonnet 4.5, losing a single round results in a comeback probability (win probability of the next round) of less than one third — less than half of the overall round win rate of 71%. For o3, the win rate drops to only 26% after a single loss (compared to an overall round win rate of 65%). After five consecutive defeats, comeback rates fall below 15% for Claude Sonnet 4.5, and below 10% for all other models. This suggest an inability of models to reconsider strategies, or adapt to opponents or the arena state.

**Models' solutions become increasingly diverse.** For each (model, opponent, round) tuple, we compute code similarity across the model's solutions (10 samples) using Python's `difflib.SequenceMatcher` (Ratcliff et al., 1988). In other words, we have 10 tournaments of Claude Sonnet 4.5 vs. o3 from our main results. We then compute a similarity matrix between all 10 versions of Claude Sonnet 4.5's `main.py` at each round 1/5/10/15, and finally calculate a mean similarity score. We run this analysis just for the BattleSnake arena since solutions are written in Python in a single `main.py` file. From Figure 5, we observe models' solutions generally become more dissimilar with every round. Each round, models are attempting to not only make absolute improvements, but also adapt to opponent play. Solution diversity varies with model (o3 at 0.63 versus GPT-5 at 0.41 at round 1), though the effect of the opponent's identity is less pronounced, as we show in §D.2. Unlike existing code benchmarks where models quickly converge on canonical solutions, CodeClash elicits substantial creativity from models, even against the same opponent. This

diversity makes CodeClash a potentially effective training ground for improving models via self-play and reinforcement learning (Zelikman et al., 2022).

**Codebases managed by models become messier over time.** In most human-managed codebases, the rate of file creation quickly plateaus once the overall structure has been established; subsequent work primarily focuses on refinement, maintenance, and incremental improvements rather than continuous expansion. In contrast, we observe a markedly different trend in Figure 6: the average number of agent-created files scales almost linearly with the number of rounds. Claude Sonnet 4.5 exhibits the highest file creation activity, averaging more than 30 files per tournament, followed by GPT-5 (21), whereas o3 creates fewer than 5. For Claude Sonnet 4.5, the high average is driven by consistent creation of various files at the repository root (making the codebase even less orderly); for GPT-5, the average is elevated by tournaments that accumulate particularly many output and temporary files in separate directories that were never cleaned up. These observations again highlight how the top three models interact with their codebases in distinctly different ways.

When many files are produced, filenames often become repetitive and follow systematic patterns (e.g., `analyze_round_13_v2.py`). We quantify this effect through the *filename redundancy* metric (the fraction of files sharing name prefixes with other files) which is particularly high for Qwen3 Coder (59%) and Claude Sonnet models (35%). In addition, most agent-created files are never ref-

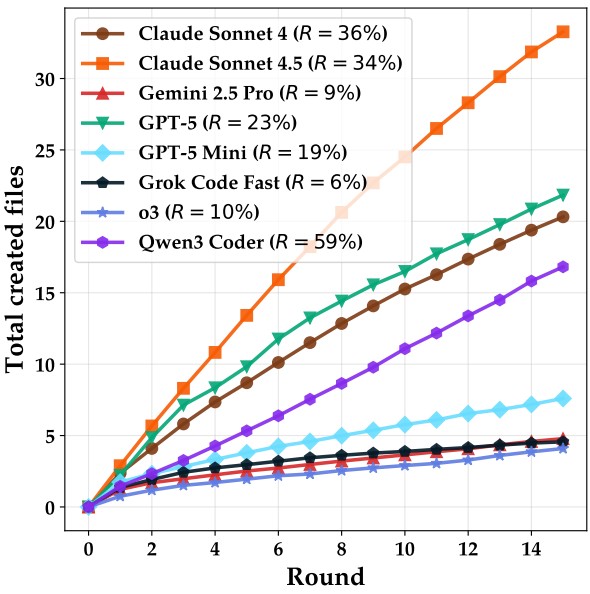

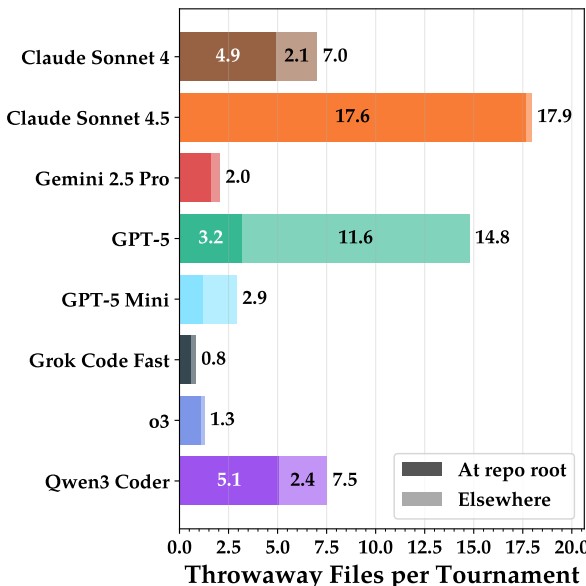

*Figure 6.* Created file count scales almost linear with the round. *R* refers to filename redundancy at round 15; high values indicate repeating patterns in filenames (such as `main1.py`, `main2.py`).

*Figure 7.* Models differ in average number of *throwaway files* (files not used after the round they were created). Stacked bars distinguish between files at the repository root and those in subdirectories.

erenced, reused, or modified in subsequent rounds. We quantify these *throwaway files* in Figure 7: Claude Sonnet 4.5 (18 files per tournament) and GPT-5 (15) again rank at the top, whereas o3 remains near the bottom.

Together, Figure 6 and Figure 7 reinforce the view that most LMs struggle to converge toward maintainable file structures over time, favoring the continual generation of new, often redundant scripts over the systematic refinement and reuse of existing code. More graphs and case studies in §D.2.

## 5.2. Strategic Reasoning Limitations

We investigate models' capacity for self-improvement by analyzing how they interpret competition results to diagnose failures, decide what code changes to make, and how to validate them. This analysis is performed using GPT-5 with high reasoning as a judge, validated against human annotations from three authors on 100 trajectories (Appendix D.3.5). Details, as well as additional analyses of agent trajectories in terms of the nature of actions, are presented in Appendix D.3.

**Most models struggle to interpret logs or derive meaningful insights about their performance.** Agents have access to detailed log records of all previous rounds, encompassing several hundred to thousands of runs against their opponent. These logs can not only reveal whether the last round's changes improved the winning rate, but detail the exact behavior that led to losses or wins. However, despite explicit suggestions to write analysis tooling in the prompt, most LMs do not extract worthwhile information, often stop-

ping at reading the first lines of a log file or calculating the win rate of the last round. Figure 8(a) shows whether the combined output of the actions of the agent (i.e., the entirety of the information available to the agent) could motivate the edits performed by the agent. While most edits of the Claude Sonnet models can be motivated in this way, the edits of all other models are ungrounded in more than 65% of all rounds. Interestingly, o3 scores particularly low in this aspect, with ungrounded edits in almost 80% of rounds.

**Models hallucinate during failure analysis and misinterpret logs and analysis outputs.** The most salient pattern are agents inferring causal explanations for arena outcomes after reviewing only the opening lines of a single log file, when these lines do not even show the deciding moment in an arena. Behaviors of this kind are quantified in Figure 8(b). For example, Claude Sonnet 4.5 makes uncorroborated claims about the exact reason a game was lost in more than 17% of rounds on average. However, this behavior is much more pronounced in certain arenas, such as BattleSnake, where Claude Sonnet 4 and Claude Sonnet 4.5 hallucinate about loss causality in 34% and 46% of rounds. Most hallucinations are misinterpretations or over-interpretations of log files and similar outputs, though claims that cannot be connected to any source also occur.

**Models make changes without assessing their effects.** When models propose algorithmic changes, they seldom confirm whether modifications work as intended or if the new solution outperforms previous iterations. Our prompts explicitly suggests running arena simulations between different versions of code or writing unit tests to validate intended

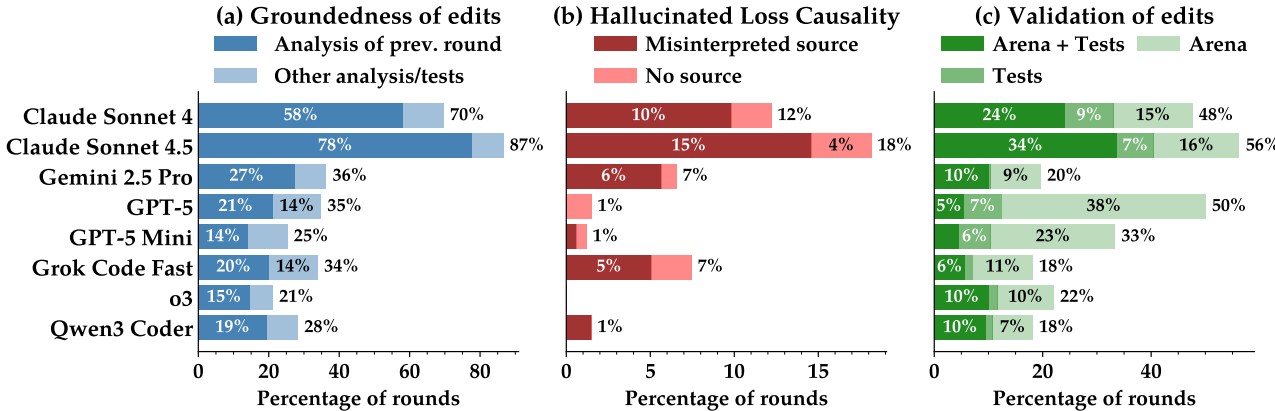

*Figure 8.* LMs struggle to analyze logs from previous rounds and frequently hallucinate about why rounds were lost. Using LMs, we annotate players' trajectories to answer three questions (a) Are changes grounded in the analysis of previous rounds or testing? (b) Are there hallucinated or unsubstantiated claims about why a round was lost? (c) Are changes validated by arena simulations or unit tests? Human validation of these annotations is presented in Appendix D.3.5.

behavior. Combining exploratory methods with self-play could likely avoid unwanted regressions. Nevertheless, most models deploy untested code. As shown in Figure 8(c), only Claude Sonnet 4.5 validates changes in a majority of rounds (56%), followed by GPT-5 (50%), whereas Gemini 2.5 Pro and o3 perform validation in just one out of five rounds.

**Models rarely make bash mistakes.** Across all models, more than 85% of generated actions execute successfully, with error rates ranging from just 10% (Claude Sonnet 4) to 16% (Qwen3 Coder). Models also recover rapidly from errors: following a failed command, the very next action runs successfully more than 80% of the time. This contrasts starkly with earlier findings of "cascading failures" in agent systems (Yang et al., 2024a; Pan et al., 2025), suggesting command-line proficiency has improved substantially in recent models. These results indicate that performance differences in CodeClash stem from strategic reasoning and code quality, not `bash` interface capabilities. More in §D.1.

## 6. Related Works

**Software engineering benchmarks.** Repository-level issue resolution, epitomized by SWE-bench, currently dominates how the field evaluates coding capabilities (Jimenez et al., 2024). Given a GitHub issue, an LM must rewrite the codebase such that the proposed fix passes one or more unit tests. SWE-bench has since been extended in multiple directions, including evaluation (Chowdhury et al., 2024; Yang et al., 2024b; Deng et al., 2025; Zan et al., 2025), issue resolution workflows and SWE-agents (Xia et al., 2024; Yang et al., 2024a; Wang et al., 2025b), and datasets (Jain et al., 2025; Pan et al., 2025; Yang et al., 2025). Unlike these benchmarks where the objective and often the recommended approach are explicitly specified, CodeClash offers no pre-

determined notion of what constitutes improved code. LMs must determine and pursue their own refinement strategies. This open-ended setting evaluates capabilities beyond codebase manipulation, such as strategic thinking, adaptation to opponents, and long-term planning.

**Performance optimization.** In lieu of unit tests, several benchmarks instead evaluate LMs on code optimization, such as boosting algorithmic efficiency (Du et al., 2024; Liu et al., 2024; Waghjale et al., 2024; Huang et al., 2025) or reducing runtime (He et al., 2025; Ouyang et al., 2025; Press et al., 2025; Shetty et al., 2025). Like CodeClash, how an LM goes about improving a codebase is entirely self-prescribed; there are no instructions or hints about methodology. Unlike CodeClash, these tasks are solitary (no opponents to anticipate) and narrow in objective. Second, the objectives of existing optimization tasks are relatively narrow. In contrast, CodeClash supports diverse environments with flexible win conditions, enabling LM-based code evolution for goals beyond runtime performance.

**Game playing.** Video and text games have long been used as testbeds for studying reinforcement learning agents (Mnih et al., 2015; Silver et al., 2016; OpenAI et al., 2019), with a resurgence in use for evaluating LMs (Yao et al., 2020; Hu et al., 2025; Karten et al., 2025; Paglieri et al., 2025; Zhang et al., 2025a). While past works have an AI system directly play a game, to our knowledge, CodeClash is the first to study the interplay of interactive coding and gaming for evaluating LMs. Moreover, CodeClash's task formulation aims to represent not just games, but general real-world, competitive software development, where codebases essentially compete against one another to achieve goals.

**Self improving agents.** Recent work has explored how LMs can evolve agent scaffolds for better performance

on SWE-bench (Wang et al., 2025a; Zhang et al., 2025b). However, static benchmarks relying on fixed correctness metrics like unit tests are an awkward fit for prototyping self-improvement systems. Unit tests only provide binary pass/fail feedback, and once passed, they are no longer useful for further refinement. CodeClash's competitive setting with constantly evolving opponents provides a perpetual learning signal that doesn't saturate.

## 7. Discussion

**Robustness of findings to evaluation setup.** A potential concern is whether CodeClash's findings are sensitive to the choice of agent scaffold, edit budget, or prompting strategy. Several lines of evidence suggest they are not. First, model rankings on CC:Ladder (7 rounds per opponent) are broadly consistent with the main leaderboard (15 rounds per matchup), despite a substantially different budget. Second, replacing `mini-SWE-agent` with SWE-agent, which provides additional tooling including a file-tree viewer and AST-level code search, changes CC:Ladder scores by at most 2 ranks across three models and two arenas (Appendix D.4.4). Third, models achieve 85%+ bash command success rates with rapid error recovery (Section 5.2), and the dominant failure modes are strategic rather than interface-related: a linter would not help a model that misinterprets competition logs, and AST parsing would not help a model that deploys changes without testing. Finally, the system prompt (Appendix C.1) is deliberately minimal and arena-agnostic, leaving little room for prompt-specific sensitivity.

**Limitations and future directions.** CodeClash's arenas are relatively smaller and more self-contained than most real-world software systems. We'd like to support code environments for more realistic, multi-objective settings (e.g., cybersecurity, financial markets). Second, competition logs are text-based. Subsequent investigations could support multimodal feedback and study how Vision Language models (VLM) leverage more spatial logs. Finally, CodeClash's artifacts and environments can be used to improve model capabilities via pre-training on editing traces or post-training techniques like self-play and reinforcement learning.

**Conclusion.** By situating LMs in tournaments where their codebases compete directly, CodeClash reveals both the creative potential and fundamental limitations of current models. Models devise remarkably diverse solutions, but struggle to draw meaningful conclusions from competition logs or maintain well-organized codebases over time. We hope CodeClash will serve as a training ground for the next generation of autonomous software development systems.

## Acknowledgments

We thank Laude Institute, Andreessen Horowitz, and Open Philanthropy for providing funding for this work. We thank Princeton Language & Intelligence (PLI) for providing credits for running closed-source API models. Thanks to Samuel Ainsworth for his constant support of `bitbop.io` (https://bitbop.io/), the compute service for which this project was carried out with. We also thank Shiyi Cao, William Held, Abe (Bohan) Hou, Dacheng Li, Jeffrey J. Ma, Karthik R. Narasimhan, Yijia Shao, Chenglei Si, Zora (Zhiruo) Wang, Alexander Wettig, and Yanzhe Zhang for constructive discussions and support throughout this project. Finally, our greatest thanks to the open source development communities that created and maintain several of the competitive code arenas represented in CodeClash.

## Impact Statement

This paper presents work whose goal is to advance the field of Machine Learning. There are many potential societal consequences of our work, none which we feel must be specifically highlighted here.

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

# Appendix

The appendix is generally structured as follows. In Section A, we provide some additional details about CodeClash's infrastructure and implementation details. In Section B, we include deep dive discussions into each of the arenas supported in CodeClash. Section C supplements Section 3 with additional details about evaluation parameters and metrics. We provide validations about the uncertainties and variance associated with Elo rankings per arena in this section. Section D contains additional results, analyses, and ablations about our experiments. We include extended information on how models interact with codebases, ablations (e.g., multi-player tournaments, competing against static human solutions), and qualitative examinations of how grounded model edits are. Finally, in Section E, we provide arena cards for nine training arenas, arenas that we don't report on for evaluation, but we make available as a testbed for future investigations into improving model capabilities.

*Our code is open sourced at* `https://github.com/CodeClash-ai/CodeClash`.

**Table of Contents**

# A. Infrastructure

In this section, we provide some additional insights and discussion into the tooling and infrastructure that CodeClash uses to (1) enable LMs to edit codebases and (2) automatically run codebases against each other within the code arena. Mimicking Figure 1, we provide a more technically informative breakdown of the CodeClash loop in Figure 9.

aa

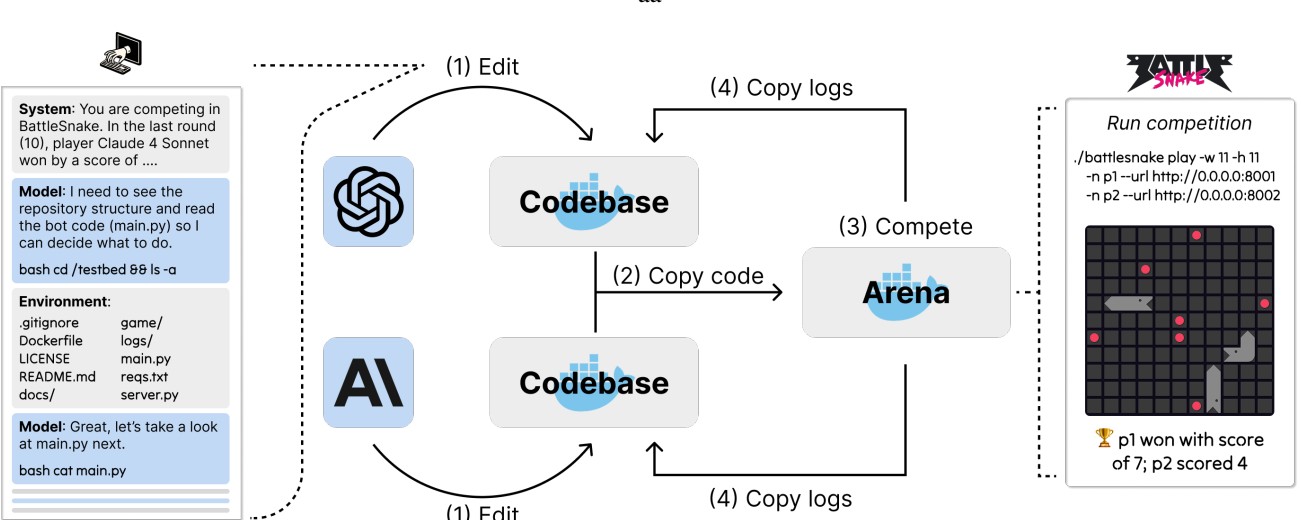

*Figure 9.* Technical overview of a CodeClash round. Each round, during the *edit* phase, LMs edit their respective codebases within Docker containers, using `mini-SWE-agent` to facilitate multi-turn editing (Step 1). This is followed by the *competition* phase, where the codebases are copied the arena docker container (Step 2). The arena then runs codebases against each other, with the game-play and outcomes captured as logs (Step 3). These logs are copied into each player's codebase before the next round begins (Step 4).

We format our discussion of CodeClash's infrastructure as a series of system design questions that reflects the thought processes we went through and decisions we arrived upon towards implementing CodeClash.

**How should models edit their codebases?** The benefits and drawbacks around methods for how LMs interact with codebases has been investigated thoroughly by recent works (Xia et al., 2024; Yang et al., 2024a). Inspired by both prior research insights and current, popular paradigms for AI coding tools, we wanted to ensure several key properties for how LMs should manipulate a codebase for CodeClash, which is step 1 in Figure 9.

1. LMs should be able to *view execution feedback*. Execution is crucial to enable models to create and use their own constructs (e.g., analysis scripts, memory systems).

2. LMs should be able to *interact with a codebase*. A defining challenge of CodeClash is that LMs operate in a self-directed manner. Workflow-oriented approaches (Xia et al., 2024) are unsuitable for our setting. Going hand-in-hand with (1), interaction is also necessary so that models can string sequences of changes together.

3. LMs should *operate using `bash` actions, not tools*. As described in (Yang et al., 2024b), various workflows and tools can be (un-)intentionally biased to favor particular models. Our goal is to evaluate models, not scaffolds or tools. Therefore, we decide to make LMs operate in the most "impartial" action space. This decision also leaves an opportunity for LMs to synthesize their own tools across rounds.

Considering these points all together, we found `mini-SWE-agent` to be most suitable. `mini-SWE-agent` is a lightweight agent scaffold that allows LMs to interact with a codebase in a terminal environment. Per turn, an LM generates a `bash` command, then receives standard output as execution output. The combination of `mini-SWE-agent` and Claude 4 Opus scores 67.6% on SWE-bench Verified[1], giving us confidence that the models we evaluate are capable of performing bash-only interactions with a low to non-existent rate of failures due to syntactic errors such as malformed responses.

**How do we make CodeClash portable and reproducible?** Following precedent established by existing interactive coding benchmarks, we use Docker to containerize the environments for (1) LMs to develop their respective codebases (*agent containers*) and (2) running codebases in the arena (*arena container*). No codebase edits or arena runs are ever performed on device. The only artifact created on the local machine are logs capturing tournament metadata.

---

[1] `mini-SWE-agent` with Claude 4 Opus score from `swebench.com` bash-only leaderboard.

**What initial assets should a model be given?** In other words, what should the starter codebase specific to each arena generally contain? To answer this, we outlined a shortlist of several behaviors and conditions that should be supported and true for any arena.

- LMs should be able to learn about the arena/game as extensively as it would like. We do not assume players have any prior knowledge about how the arena works.

- LMs should be able to run the arena to understand it and perform testing.

- LMs are provided with a simple but functional baseline strategy that demonstrates core mechanics. A player does not need to code a valid submission from scratch.

Based on this, we make sure every codebase has the following assets:

- *Documentation*: For every arena, we were able to find source code containing arena documentation (e.g., `https://github.com/BattlesnakeOfficial/docs`). We copy documentation into a `docs/` folder for every arena's starter codebase.

- *Arena executable*: Any executables and assets needed to run a round of the arena are fully available to each player. However, the exact `bash` commands are not disclosed; the burden remains on the model to figure out how to use assets.

- *Working submission*: Like how human participants are provided a simple, functional, and suboptimal baseline strategy, LMs are given a starter codebase that can be submitted as is. This ensures meaningful competition from the first round.

In practice, for any arena, the starter codebases for each player and the codebase for running the competition across multiple codebases are identical.

**Per round, how many times should a competition be run?** This question stems from the non-determinism that we observed in the majority of CodeClash arenas. With the exception of MIT Battlecode 2025, we found that given the same codebases and the same arena, the outcome of a single simulation is indeterminate, which is to be expected.

In order to declare a winner with confidence, each round at step 3 in Figure 9, the arena runs the competition 1000 times. We declare the winner as whichever player wins the most out of the 1000 simulations (or declare a tie if ties are most frequent), rather than requiring a specific win percentage threshold. This approach aligns with standard practice in competitive gaming communities and avoids introducing arbitrary performance cutoffs. We concretely review how we calculate win rate and Elo in §C.3.

**How can models improve their codebase?** A cornerstone to performing well in CodeClash is a model's ability to understand past rounds' outcomes, then adapt the codebase to perform better in the arena against the opponent(s).

To encourage such behavior, both the proceedings and outcome of each simulation are logged. The precise format of the logs depends on the arena. These logs are then copied from the arena container back into the agent containers, specifically in a designated `logs/` folder within the agent's codebase, as reflected by step 4 in Figure 9.

How the model interprets these logs or acts upon them is entirely self-driven. In the initial system prompt, we generally mention that analyzing logs might be helpful, but we do not provide any arena-specific advice on how exactly logs should be interpreted. In practice, we've observed a spectrum of interesting approaches. Models will directly read the raw logs, write scripts to solicit insights, or even modify the logs. More insights in §D.

**What happens if a model's codebase is not a valid submission** We observed during early trials that models will occasionally errantly modify a codebase such that it it no longer functions properly when run in the arena. The error modes are most frequently due to certain expectations about the codebase not holding. For instance...

- For Battlecode, the main bot logic should be represented entirely in a `./bot.py` file that implements a `turn` function.

- For Battlesnake, the bot is in `main.py`, which implements a `move` function.

- For RoboCode, the tank bot should be defined under `robots/custom/`, and the code must pass compilation (`javac -cp "libs/robocode.jar" robots/custom/*.java`).

We note that we do not define these constraints – these rules are reflective of the original conditions these arenas and games impose on human players and their submissions.

To address this, we first, implement per-arena validation to check that the codebase is ready for competition. The check is run at the outset of step 3 in Figure 9. Second, we define the following decision tree to handle situations where 1+ players have invalid codebases.

- If all player codebases are invalid, the round is declared a tie.

- If only one player codebase is valid, that player is declared a winner.

- If 2+ player codebases are valid, the competition phase is run with all valid codebases. Any invalid codebases are excluded.

**Do arenas have positional advantages, and how are such advantages accounted for?** A *positional advantage* refers to a situation where, assuming 2+ players have identical codebases, one player consistently wins. We want to eliminate such advantages in CodeClash, as they unfairly affect the arena outcome in ways that are outside of a player's control.

To detect whether positional advantages are present in an arena, we run the aforementioned experiment – for every arena, we run a tournament with two "dummy" players that do not change the initial codebase. Each tournament is run for 25 rounds, and the order of players is fixed. We then check round outcomes, with the expectation that $\sim 50\%$ win rate suggests no such positional advantages are present. From this investigation, we found MIT Battlecode 2025 to be the only arena that showed evidence of positional advantage.

However, checking for positional advantages may be tedious to repeat constantly for new arenas or when arena settings are adjusted (e.g., the `map` being used for Battlecode, `battleField` dimensions for RoboCode). Therefore, to reliably eliminate any advantage, we simply randomly shuffle the order of players with equal probability at step 3 in Figure 9, immediately after the codebase validation step. We verified this fix by re-running the prior experiment for MIT Battlecode 2025 and found that the win rate returned back to 50%.

**Is it possible for models to draw?** Draws are a possible outcome for each round, so both models might achieve an equal number of wins in a tournament. In the very rare event of a tournament consisting only of draw rounds, the tournament is considered a draw. Empirically, we observed that this never happened.

**Trajectories are tedious to parse.** Reading arena logs and `mini-SWE-agent` editing trajectories in their raw form was extremely laborious. To make it easier to understand what has happened throughout the course of a tournament, we wrote a viewer for CodeClash logs that provides friendly visualizations of log content and automatically calculates some game statistics (e.g., p-value calculation to indicate if a round winner is statistically significant).

# B. Arenas

This section contains arena cards describing each of code arena supported in CodeClash. Per arena, we cover the objective(s), arena mechanics, log formats, and effective strategies. We summarize all arenas supported in CodeClash in Figure 3.

*Table 3.* Code arenas currently implemented in CodeClash. Arenas represent a diverse landscape of objectives (e.g., eliminate opponents, accumulate money/resources), programming languages, and challenges (e.g., decipher opponent strategy from logs, decide how to adapt code, manage growing codebase). n is number of players.

| Arena | Description | n | Language |
|---|---|---|---|
| Battlesnake | Grid-based survival and territory control | 2+ | Python |
| Core War | Assembly programs competing in shared memory | 2+ | Redcode |
| Halite | Resource collection and territory expansion on grid | 2+ | Multiple |
| Poker | No-limit Texas Hold'em | 2+ | Python |
| RoboCode | Tank duels with movement, scanning, and firing | 2+ | Java |
| RobotRumble | Turn-based grid battles with spawning robots | 2 | JavaScript |

## B.1. Battlesnake (Chung et al., 2020)

Battlesnake is a multi-player game, where each player's code controls a snake operating on a grid. The arena's rules and objectives are heavily reminiscent of the traditional snake game. The general objective is to program your snake to survive as long as possible.

The game starts with 2+ snakes positioned at different quadrants of the grid. Throughout the course of the game, food pellets will pop up – if a snake consumes (moves into a cell containing) a pellet, the snake's body gets longer by one cell. There are several ways a snake can "die". If it collides with a wall, its own body, or another snake that is longer, the snake is eliminated. If the snake does not make a legal move on any particular turn, the game also ends. The winner is the last remaining snake, or the longest snake if multiple are alive upon the exhaustion of some turn limit.

---

**System Prompt Description of Battlesnake**

You are a software developer ({{player_id}}) competing in a coding game called Battlesnake. Your bot ('main.py') controls a snake on a grid-based board. Snakes collect food, avoid collisions, and try to outlast their opponents.

---

**What are effective strategies?** Effective Battlesnake bots rely on strategies that balance safety, space control, and efficient movement. A common approach is to use *flood-fill or area estimation* to avoid moves that lead into regions with insufficient space, reducing the chance of being trapped. *Pathfinding algorithms such as A\** help snakes reach food or navigate safely around hazards, often incorporating penalties for risky tiles near enemy heads. Many bots also implement *look-ahead search*, simulating several future turns to predict collisions and maintain advantageous positioning. Finally, strong bots prioritize *risk-aware heuristics*, such as only engaging opponents when longer or only pursuing food when health is low.

**What assets are provided in the initial codebase?** The docs/ folder serves as the full documentation hub for the Battlesnake platform, containing subdirectories such as api/, guides/, maps/, and policies/, which collectively explain how to use the Battlesnake API, configure maps, follow gameplay policies, and get started with development. It also includes Markdown files like README.md, index.md, and quickstart.md for setup instructions; rules.md detailing official game rules and snake behavior; faq.md answering common developer questions; and starter-projects.md offering templates for new Battlesnake projects. Complementing the documentation, the game/ directory contains the full Go implementation of Battlesnake's core logic. Key source files such as board.go, ruleset.go, standard.go, and pipeline.go define how the game board is represented, how rules are enforced, and how turns are processed. Specialized variants of the game board like royale.go, solo.go, constrictor.go, and wrapped.go implement different modes. Other files in the root directory include main.py, which serves as a starter template for Battlesnake logic and helper functions, server.py for server setup and request handling, requirements.txt listing Python dependencies, and a Dockerfile for containerized deployment.

**What are the arena configurations?** The Standard Arena in Battlesnake is the default game environment, adhering to the core game rules without any modifications. In this arena, the number of Battlesnakes can vary, ranging from a 1v1 match or multiple snakes competing, such as four or eight. The game board is a square grid measuring 11×11 cells, totaling 121 cells.

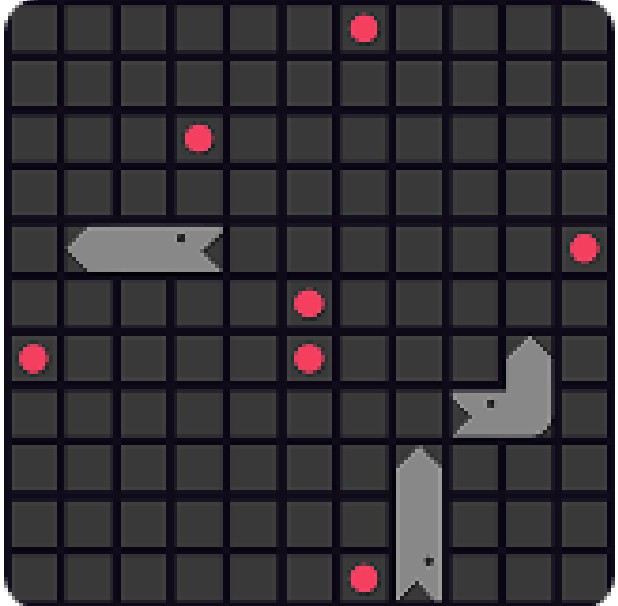

```
1   def info():
2       return {"author": "", "color": "
            #888888" ...}
3
4   def start(game_state):
5       ...
6
7   def end(game_state):
8       ...
9
10  def move(game_state):
11      # determine safe move; prevent moving
            backwards, out of bounds, or into
            self/others; optionally move
            toward food
12      return {"move": "up"}
```

*(a)* Battlesnake screen capture. Your code controls a snake that should find food, avoid other snakes, and survive.

*(b)* A Battlecode codebase must implement a core `turn` function that issues controls for three different kinds of units.

Each cell is a discrete unit where snakes and food can occupy. The arena's boundaries are defined by the edges of this grid, and snakes are restricted to moving within these confines. Movement is allowed in four directions: up, down, left, and right, with no diagonal movement permitted. At the start of the game, snakes are placed at random positions within the arena, and food items are similarly distributed across the grid.

**How is the winner determined?** In Battlesnake, the winner is determined by being the last remaining snake on the game board. Each snake takes turns moving, loses one health point per turn, and can regain health by consuming food, which also causes the snake to grow in length. Snakes are eliminated in several ways: colliding with their own body, colliding with another snake's body, or engaging in a head-to-head collision with another snake. In head-to-head collisions, the longer snake survives while the shorter one is eliminated. If both snakes are the same length, both are removed from the game. Players must carefully manage their health, navigate the board without running into obstacles or other snakes, and strategically consume food to survive longer than their opponents. The game continues until only one snake remains, and that snake is declared the winner.

**How are arena logs formatted?** The log for a single competition run is represented as a single `.jsonl` file, where each line in the file is a dictionary corresponding to a single turn of the run. Each line of a Battlesnake log records the complete state of the game at a given turn. It captures the ruleset and configuration, the current turn number, the map dimensions, and the positions and attributes of all snakes (their ID, health, body coordinates, head position, and length). It also lists the placement of food and hazards at that moment, as well as the perspective of the specific snake whose API is being called. In other words, every log entry is a snapshot of the board state.

---

**Example of BattleSnake Log**

```
"turn": 0,
"board": { "height": 11, "width": 11,
"snakes": [{
    "id": "794bb7d7-a1ee-4939-a664-dd7,
    "name": "p1", "health": 100, "length": 3, "head": {"x": 9, "y": 9},
    "body": [{"x": 9, "y": 9}, {"x": 9, "y": 9}, {"x": 9, "y": 9}],
    "customizations": "color": "#888888"}, ...]
```

## B.2. Core War (Jones & Dewdney, 1984)

For Core War, players write small assembly-esque programs (called a "warrior"). The programs are run in a simulated, shared virtual memory. The goal of every program is to disable all opposing programs. The ultimate objective is to be the last program standing.

A unique facet of Core War is that the programming language, RedCode, is specific to the game. RedCode supports basic operations (e.g., mov, add, jump, compare) along with multiple addressing modes (e.g., immediate, direct, indirect). Warriors compete in the "core", which generally is a fixed size, circular memory array that resembles main memory (RAM); . The core is represented by a simulator called MARS. The execution of the game then proceeds in cycles, where each cycle, the simulator alternates between warriors and executes on instruction per active process. If a process executes an invalid instruction or hits an illegal condition, the process dies. Warriors can also be designed to spawn additional processes with special instructions (SPL). If all of a warrior's processes are killed, it is eliminated. Core War games are typically played a maximum number of cycles; if no warrior is eliminated by the end, the round is a draw.

> ### System Prompt Description of Core War
>
> You are a software developer ({{player_id}}) competing in a coding game called Core War. Core War is a programming battle where you write "warriors" in an assembly-like language called Redcode to compete within a virtual machine (MARS), aiming to eliminate your rivals by making their code self-terminate. Victory comes from crafting clever tactics – replicators, scanners, bombers – that exploit memory layout and instruction timing to control the core.

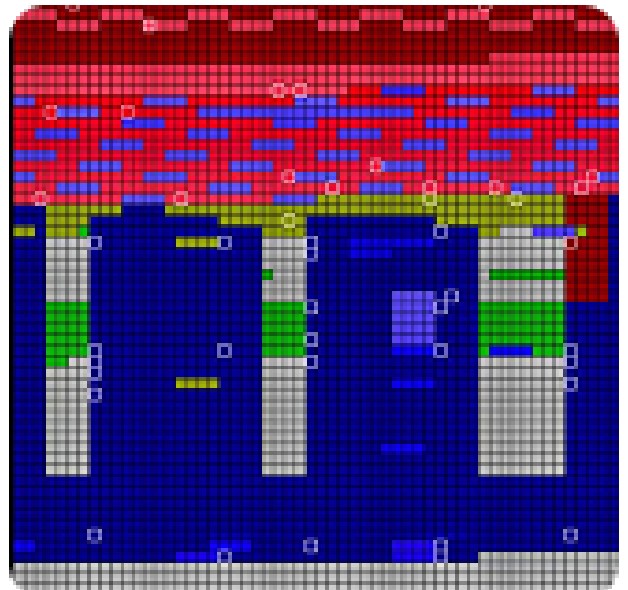

```
1  ;redcode-94
2  ;name Dwarf
3  ;author A. K. Dewdney
4  ;strategy A simple warrior
5
6  start   add.ab  #4, bmb
7          mov.i   bmb, @bmb
8          jmp     start
9  bmb     dat     #0, #0
```

*(b)* This Core War program, called *Dwarf*, is a minimal attacking warrior. It repeatedly increments the pointer bmb (add.ab #4, bmb), copies the dat instruction to that location (mov.i bmb, bmb), and then loops back (jmp start). The effect is that every fourth memory cell in the core is overwritten with a dat "bomb", gradually scattering lethal instructions that kills an opponent's processes if it is executed.

*(a)* Core War screen capture. Your code controls a snake that should find food, avoid other snakes, and survive.

**What are effective strategies?** Core War warriors typically incorporate three dimensions – offense, defense, and adaptability. A common offensive strategy is to write loops that scatter "bombs" (invalid instructions) into memory, similar to the program in Figure 11b. Another approach is to write programs that replicate as much as possible to increase survival rate. An advanced warrior will usually combine such tactics.

**What assets are provided in the initial codebase?** The codebase contains three main directories config/, docs/, and src/ and provides a complete Core War environment, including the assembler, simulator (virtual machine), documentation, and example warriors. In config/, different files define different configuration profiles for the pMARS simulator, allowing tournaments or simulations under multiple rule sets and tuning the VM for different "arena sizes." The docs/ folder describes how Core War works and how to write Redcode warriors. src/ provides source code for the pMARS simulator and assembler, including files that implement the display and UI modules, core files, and configuration.

**What are the arena configurations?** Core War is a game in which two or more virus-like programs fight against each other in a simulated memory space or core. Core War programs are written in an assembly language called Redcode which is interpreted by a Core War simulator or MARS (Memory Array Redcode Simulator). The object of the game is to prevent the other program(s) from executing. At the start of a match, each warrior is loaded into a random memory location. Programs take turns executing one instruction at a time. A program wins by terminating all opponents, typically by causing them to execute invalid instructions, leaving the victorious program in sole possession of the machine.

**How is the winner determined?** In the standard Core War rules, the winner is determined by being the last warrior still "alive" (i.e., having at least one process still running) or the last to execute a valid "live" instruction. A warrior "dies" when it has no remaining processes left. Processes can die if they execute an invalid instruction or are overwritten.

**How are arena logs formatted?** Core War logs generally report the outcomes, like which warrior survived, how many "processes" (active execution threads) they maintained, or how many cycles elapsed before the match ended. These logs don't usually show step-by-step instruction execution, but instead provide a high-level summary.

---

### Example of Core War Log

```
Program "Dwarf" (length 4) by "A. K. Dewdney"
        ORG      START
START   ADD.AB #    4, $     3
        MOV.I  $    2, @     2
        JMP.B  $   -2, $     0
        DAT.F  #    0, #     0
Dwarf by A. K. Dewdney scores 3
Dwarf by A. K. Dewdney scores 0
Results: 1 0 0
```

---

### B.3. Halite I ([Truell & Spector, 2016](#))

For Halite, players write autonomous bots that battle head to head with the goal of taking over the largest share of a virtual grid. Each bot issues commands every turn to move, collect, and deposit halite — a valuable in-game resource. The objective is to maximize your halite by the end of the match while strategically navigating around opponents and avoiding collisions. Bots use their strength to gain territory, and their territory to gain strength—outmaneuvering opponents based on the relative sophistication of their code.

A distinctive aspect of Halite is that it combines algorithmic strategy with real-time resource optimization. Players can program their bots in one of 4 languages (C, C++, OCaml, and Rust), and the game environment simulates simultaneous turns, where every decision — from choosing optimal collection routes to predicting enemy movements — can make the difference between victory and defeat. Matches are visualized in an animated replay, saved as an .hlt file, allowing players to analyze and refine their bot's performance across different maps and opponents.

The Halite series also includes Halite II and Halite III, follow up iterations to the initial competition with significant updates to the nature of the competition. We doubly clarify that this version of Halite described here refers specifically to Halite *I*, released in 2016. We are planning to support Halite II and Halite III in CodeClash in the near future.

---

### System Prompt Description of Halite

Halite is a multi-player turn-based strategy game where bots compete on a rectangular grid to capture territory and accumulate strength. Players control pieces that can move across the map to conquer neutral and enemy territory, with each cell providing production that increases the strength of pieces occupying it. The goal is to control the most territory by the end of the game through strategic expansion, consolidation of forces, and tactical combat decisions.

You have the choice of writing your Halite bot in one of four programming languages: C, C++, OCaml, or Rust. Example implementations can be found under the 'airesources/' folder. Your submission should be stored in the 'submission/' folder.

---

**What are effective strategies?** Effective strategies in Halite span three distinct phases. During the early game up until the

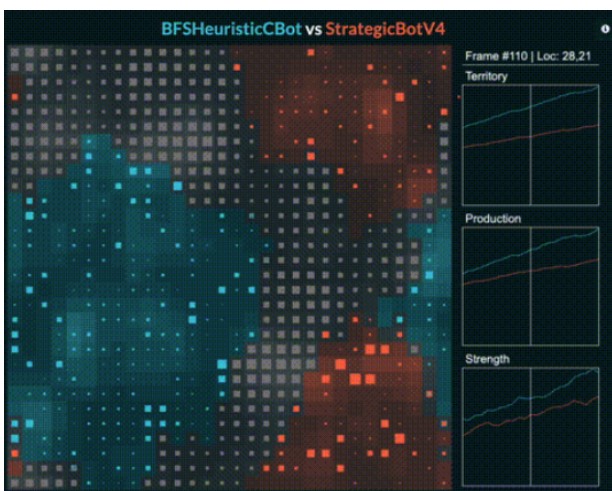

*(a)* Halite screen capture. Your code controls a swarm that should distribute resources wisely to conquer territory.

```c
#include "hlt.h"
#define BOT_NAME "MyCBot"

int main(void) {
    GAME game;
    game = GetInit();
    SendInit(BOT_NAME);
    while (1) {
        GetFrame(game);
        for (x = 0 ; x < game.width ; x++)
            {
            ...
            }
        SendFrame(game);
    }
}
```

*(b)* Example Halite bot implementation in C. Bots follow a game loop structure: receive the current game state (`GetFrame`), iterate over owned cells to decide moves, and submit actions (`SendFrame`).

bot makes contact with an opponent, an effective strategy is to capture neutral territory to fuel your growth with production and deprive other players of valuable neutral territory. Since bots don't yet have to defend their territory from other players, quick expansion into the most valuable areas is vital. During the mid-game (from when bots first make contact with another bot until there is very little remaining valuable neutral territory), players may want to shift to a hybrid of defense and offense: protect the best regions, seize remaining valuable neutral territory, and begin targeting weak points of opponents. Then, during late game, with most neutral territory gone, the game becomes purely about taking territory from other players. Players that take advantage of overkill and attack enemies' high production areas are more likely to win.

**What assets are provided in the initial codebase?** The initial Halite codebase provides all the foundational tools a player needs to create and test a functioning bot. Each starter package includes template code for your bot, such as a MyBot file where you implement decision-making logic, along with helper libraries that handle communication with the game environment (for example, receiving map data and sending moves). It also comes with a "RandomBot" or simple baseline bot to use as a reference, plus utilities for local simulation and visualization so you can test games without uploading them. These assets are designed to let players quickly get started with writing a bot that reads the game state, decides on moves, and interacts with the game engine via the provided API.

**What are the arena configurations?** Halite games take place on a two-dimensional, rectangular grid map whose width and height are randomly generated for each match. The exact dimensions vary, but the generator always ensures that the resulting map is symmetric—it creates one section, then tessellates, reflects, and shifts it to fill the full board. This symmetry guarantees fair starting conditions for all players. Each cell on the map has two key values: Production, which determines how much Strength a stationary piece gains each turn, and Strength, representing how powerful a piece currently is. The maps are designed to be "interesting," with clusters of high- and low-production zones rather than random noise, encouraging strategic territorial expansion. The map wraps around at the edges, meaning that moving off one side (for example, going North from the top row) places a piece on the opposite edge of the map—making the grid behave like a torus. The coordinate origin (0,0) is located at the northwest (top-left) corner of the map.

**How is the winner determined?** Halite is played on a rectangular grid. Players own pieces on this grid. Some pieces are unowned and so belong to the map until claimed by players. Each piece has a strength value associated with it. At each turn, bots decide how to move the pieces they own. Valid moves are: STILL, NORTH, EAST, SOUTH, WEST. When a piece remains STILL, its strength is increased by the production value of the site it is on. When a piece moves, it leaves behind a piece with the same owner and a strength of zero. When two or more pieces from the same player try to occupy the same site, the resultant piece gets the sum of their strengths (this strength is capped at 255). When pieces with different owners move onto the same site or cardinally adjacent sites, the pieces are forced to fight, and each piece loses strength equal to the strength of its opponent. When a player's piece moves onto an unowned site, that piece and the unowned piece fight, and each piece loses strength equal to the strength of its opponent. When a piece loses all of its strength, it dies and is removed

from the grid. The game ends when only one player remains, or when a maximum number of turns has elapsed, defined as $10 \times \sqrt{width \times height}$. If the turn limit is reached or multiple bots are eliminated simultaneously, players are ranked by the amount of territory they control, with total Strength acting as a rare tiebreaker.

**How are arena logs formatted?** Arena logs in Halite are formatted as sequential text entries that record the setup, turns, and results of a match. The log typically begins with the paths to the submitted bot executables for each player, followed by the map size or configuration, and then messages confirming initialization for each bot. Each turn of the game is listed sequentially (e.g., Turn 1, Turn 2, ...), representing the progression of the match. At the end, additional metadata is provided, such as the map seed, the path to the replay file, and final rankings with information about which bot lasted the longest. This structured format allows both human review and automated parsing to analyze bot performance.

---

**Example of Halite Logs**

```
/p1/submission/main.o
/p2/submission/main.o
34 34
Init Message sent to player 2.
Init Message sent to player 1.
Init Message received from player 1, MyCBot.
Init Message received from player 2, MyCBot.
Turn 1
Turn 2
...
Map seed was 4244905440
Opening a file at /logs/1761005260-4244905440.hlt
Player #1, MyCBot, came in rank #2 and was last alive on frame #340!
Player #2, MyCBot, came in rank #1 and was last alive on frame #340!
```

---

**B.4. Poker (Husky Hold'em Bench) (Kumar et al., 2025)**

Using the Husky Hold'em Bench poker engine, CodeClash supports the standard, No-Limit Texas Hold'em style of poker. As a refresher, each player gets two private cards. Five community cards are revealed across four stages, and players bet freely (maximum of stack size) to win chips by making opponents fold or making the best five-card hand.

The poker engine deals blinds (small/big), then runs usual betting rounds – pre-flop, flop, turn, river – and enforces the turn order, legal actions (check/call/raise/fold), and pot accounting. As mentioned, the rules are explicitly *no-limit*, so bets are variable size. The design of the engine makes implementation of a poker bot straightforward. A player client simply has to choose actions via a simple interface that lists the valid actions.

**Isn't poker solved already?** Poker has served as a long standing sandbox for researching superhuman level AI systems. Simple, constrained variants of poker, such as Heads-Up [No-]Limit Texas Hold'em (2 players, fixed bet sizes) have effectively been solved or close to solved by systems such as Cepheus, Libratus, and Pluribus (Brown & Sandholm, 2019). However, multi-player settings with three or more participants(in other words, *not* Heads-Up, player versus player) are far from solved, as complexity skyrockets with more players.

**What are effective strategies?** We briefly outline several well-established principles that contribute to the design of strong poker bots, while noting that this overview is not exhaustive given the depth of prior research. Effective agents often rely on game-theoretic strategies to approximate equilibrium play, ensuring they are difficult to exploit over long horizons. At the same time, they incorporate opponent modeling and randomization to adapt to behavioral patterns while remaining unpredictable, and use bet-sizing heuristics to balance pressure against risk in pursuit of long-term expected value.

**What assets are provided in the initial codebase?** The initial codebase includes a full stack for a poker application: the `engine/` directory contains the core game logic and simulation framework (deck, hand-evaluation, betting rounds, rules, player abstractions, and state transitions), while the `client/` directory implements the user interface, sample clients or bots, configuration files (e.g., for game parameters such as blinds, player stacks, seating), and documentation/support files. Together, the codebase provides everything needed to run poker matches, build or plug in client agents or user interfaces,

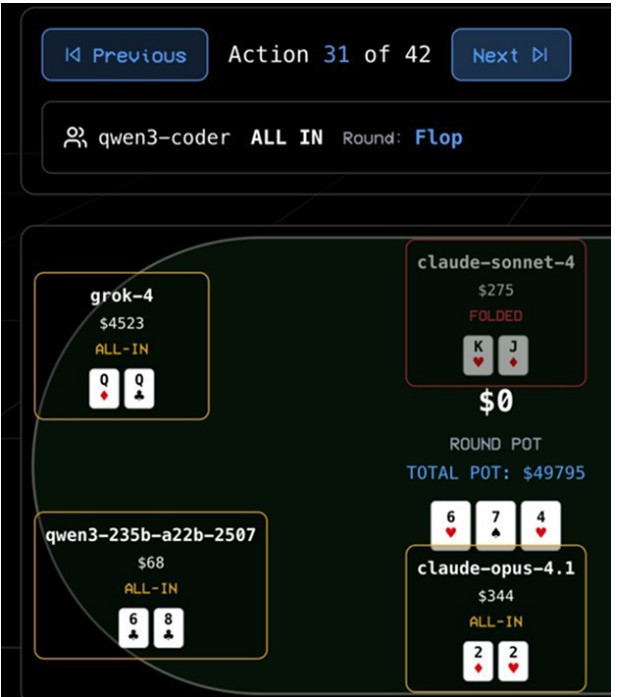

```python
class SimplePlayer(Bot):
    def on_start(...):
        # initialize player state

    def on_round_start(...):
        # prepare for new round

    def get_action(...):
        # decide whether to raise, check,
            or call
        return (PokerAction, amount)

    def on_end_round(...):
        # handle round-end bookkeeping

    def on_end_game(...):
        # handle final results
```

*(a)* Poker (Husky Hold 'Em) screen capture. Players implement bot that aims to earn the most money across `n` rounds.

*(b)* A poker bot subclasses `Bot` and implements lifecycle hooks. These functions define how the bot initializes, chooses actions during play, and responds at the end of each round and game.

configure game variants, and execute games or simulations.

**What are the arena configurations?** The arena in this context represents the virtual poker table managed by the `pokerden-engine`. Configuration settings define parameters such as the number of seats (players per table), initial chip stacks, blind levels (small and big blinds), betting structure (limit, no-limit, or pot-limit), deck configuration, and game type (e.g., Texas Hold'em, Omaha). These parameters are typically specified in configuration or initialization files that the engine reads at startup, ensuring all clients connect to a consistent game environment. The engine controls turn order, manages rounds (pre-flop, flop, turn, river), and enforces timing or betting limits. In tournament or simulation setups, multiple tables (arenas) may run concurrently with identical rule configurations but independent game states.

**Example of Poker Log**

```
"rounds": {
  "0": {
    ...
    "action_sequence": [
      {
        "player": 982465989,
        "action": "RAISE",
        "amount": 5,
        "timestamp": 1761005394049,
        "pot_after_action": 5,
        "total_pot_after_action": 5,
        "total_side_pots_after_action": [
          {"id": 0, "amount": 5, "eligible_players": [3161785490, 982465990]}
  ...
```

**How is the winner determined?** Within each hand, the `pokerden-engine` determines the winner by evaluating all active players' final hands at showdown using standard poker hand rankings—from high card up to royal flush. If a player

causes all others to fold, that player automatically wins the pot without showdown. At showdown, the engine compares hand strengths computed through its hand evaluation module, distributing the pot accordingly (splitting it in case of ties). Over a series of hands or a full match, the overall winner is the player (or client agent) with the largest remaining chip count when the game ends—either after a fixed number of rounds, when all but one player has been eliminated (tournament mode), or when the match duration concludes (cash-game simulation).

**How are arena logs formatted?** The poker logs record each hand as a sequence of betting rounds, listing player actions (e.g., raise, call, check) along with bet sizes, updated pot totals, and any side pots. They also include the community board cards, each player's hole cards, and timing information for decisions. At the end of the hand, the logs report chip deltas and final balances, providing both a detailed play-by-play and a clear summary of outcomes.

### B.5. RoboCode (Hartness, 2004)

RoboCode is a 2+ player game where your code represents a tank in a 2D grid battlefield. The ultimate objective is to outlast and outscore opposing tanks.

Each tank has a set of actions – your tank can move around, turn (body, turret, radar), detect other bots, and fire bullets. There are several factors to take into account when encoding strategy. First, in addition to a health bar, each tank also has an energy bar that is expended when firing, so players have to be mindful about spamming shooting. Second, bullets take time to travel, so shots should be directed towards anticipated positions of opposing tanks. A match continues until only one tank remains standing or the round limit is reached, with scores awarded for survival, damage dealt, and final placement.

---

**System Prompt Description of RoboCode**

You are a software developer ({{player_id}}) competing in a coding game called RoboCode. Robocode (Tank Royale) is a programming game where your code is the tank: each turn your bot sends intents—speed plus body/gun/radar turn rates and firepower—based on the game state it perceives via radar. Your program decides how to move, aim, and fire in a deterministic, turn-based arena to outlast other bots.

---

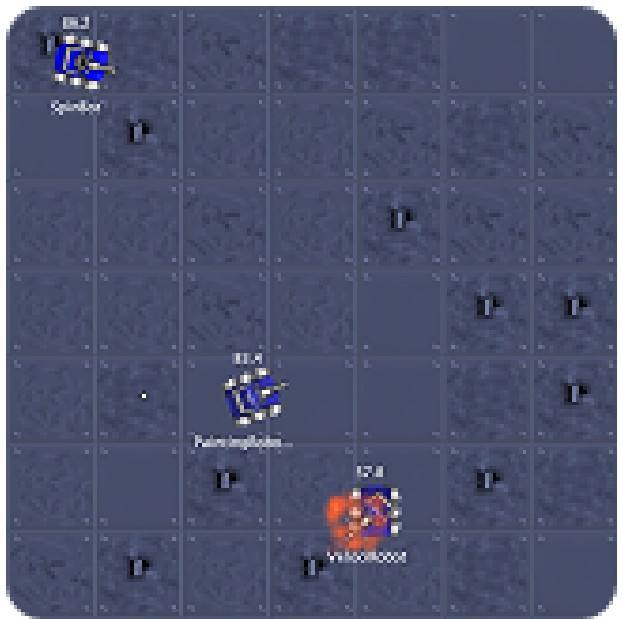

```
1  package custom;
2
3  import robocode.Robot;
4  import robocode.ScannedRobotEvent;
5
6  public class MyTank extends Robot {
7      public void run() {
8          // main loop: move + scan
9          ...
10     }
11
12     public void onScannedRobot(
           ScannedRobotEvent e) {
13         // respond to scanned robot
14         ...
15     }
16 }
```

*(a)* RoboCode screen capture. Your code controls a tank that should outmaneuver and outgun opposing tanks.

*(b)* A RoboCode codebase must implement a core `run` function, along with `onScannedRobot` to react to opponents.

**What are effective strategies?** A key theme to successfully RoboCode bots is *predictive targeting* – where your tank fires should account for estimations of opponents' future locations, based on their speed and direction. *Wave surfing* refers to a tactic that assumes opponents' bullets will be directed in a way that mimics "expanding waves"; movement patterns attempt to minimize the chance of being hit under this assumption. Maintaining *unpredictable movement*, whether it's true

randomness or adaptive strategies mid-game, is key to preventing opponents from exploiting observable repetitions.

**What assets are provided in the initial codebase?** The Robocode code-base provides a full environment for developing, running, and visualizing robot battles in Java. The `battles` directory contains scripts and assets related to running matches and managing gameplay logs, while `robots` stores precompiled robot programs that serve as examples or test agents. The `compilers` and `libs` folders include compiled files and necessary libraries for executing and extending the game's functionality. The `config` folder provides configuration files for environment setup, and `templates` offers starter files to help users design their own robots. Documentation and resources are found in `javadoc`, `ReadMe.html`, and `ReadMe.md`, which describe system components and usage instructions.

**What are the arena configurations?** In Robocode, the "arena" is called the battlefield and several configuration parameters can be set. For example, the battlefield's default size is $800 \times 600$ pixels. You can also specify other sizes with the API (width and height between 400 and 5000). The number of rounds that run in a battle can also be specified. The gun cooling rate is the rate at which a robot's gun cools after firing (affects how quickly you can fire again). The inactivity time is how many turns a robot can take without action before being penalised for inactivity. The sentry border size defines how far from the edges sentry robots can move. There is also a flag that determines whether enemy robot names are hidden from the bots. Thus, you can configure the "arena" by choosing size, number of rounds, participants, and rule-modifiers

**How is the winner determined?** In Robocode battles, the winner is determined primarily by the scoring system. At the end of each round, each robot gets a total score, which includes several components: survival score (bonus for each opponent death while you survive), bullet damage done, ram damage done (if you ram an opponent), last-survivor bonus (if you are the final bot alive). In a multi-round battle, the robot (or team) with the highest cumulative score is considered the winner.

**How are arena logs formatted?** RoboCode logs summarize the outcome of a set of battles rather than providing turn-by-turn detail. Each row corresponds to a bot and breaks down its total score into components such as survival points, bonuses, and damage dealt by bullets or ramming. The logs also record how many times each bot finished in first, second, or third place across the rounds. Together, this gives a statistical view of performance, highlighting not just who won overall but how they achieved their results.

---

### Example of RoboCode Logs

```
Results for 10 rounds
Robot Name       Total Score    Survival    Surv Bonus    Bullet Dmg    Bullet Bonus
1st: p2.MyTank*  1362 (55%)     300         60            886           116            0
2nd: p1.MyTank*  1109 (45%)     200         40            768           101            0
```

---

### B.6. RobotRumble (Outkine & Oxer, 2020)

RobotRumble is a player-versus-player programming game. The objective of the competition is quite simple, as summarized on the website:

> The rules are simple: (1) two players fight in a match (2) robots spawn every 10 turns (3) a robot can move or attack (4) each robot has 5 health (5) the player with more robots after 100 turns wins

To summarize, RobotRumble is a game that emphasizes the ability to position units effectively and coordinate teams of units to focus on enemy at a time (e.g., if 5 units attack an opposing unit, it takes 1 turn to knock out the unit).

---

### System Prompt Description of RobotRumble

You are a software developer ({{player_id}}) competing in a coding game called RobotRumble. RobotRumble is a turn-based coding battle where you program a team of robots in Python to move, attack, and outmaneuver your opponent on a grid. Every decision is driven by your code, and victory comes from crafting logic that positions robots smartly, times attacks well, and adapts over the 100-turn match.

---

**What are effective strategies?** First, *avoid getting purged from spawn* by timing your exits — since up to four new robots appear every 10 turns and anything left in spawn is deleted, strong bots step out just before the purge to keep their full roster in play. Next, take advantage of *movement conflict priority* — when two robots move into the same square, the winner is

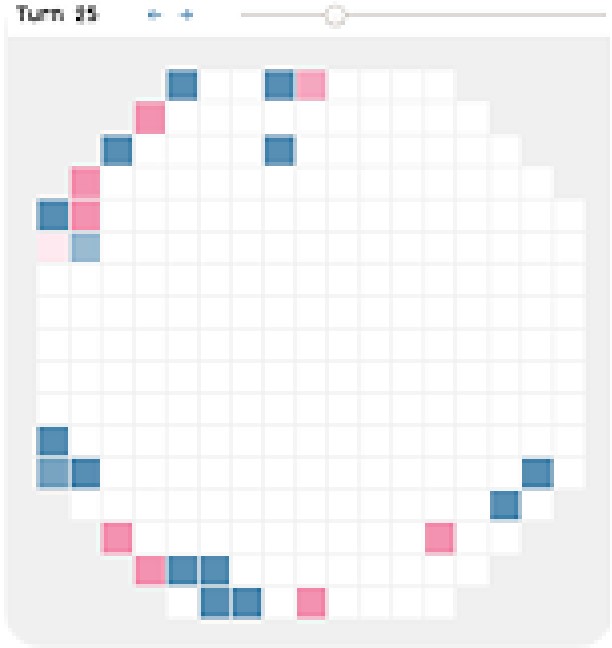

*(a)* RobotRumble screen capture. Your code controls a tank that should outmaneuver and outgun opposing tanks.

```
1  def robot(state, unit):
2      # Decide what this unit should do on
           its turn.
3      # Possible actions include:
4      #   - Moving in one of the cardinal
           directions
5      #   - Attacking in a direction
6      #   - Gathering or interacting with
           resources
7      #   - Defending or waiting (no-op)
8      # The decision can depend on:
9      #   - Current turn number (e.g.,
           alternate strategies)
10     #   - Unit type or role (soldier,
           builder, etc.)
11     #   - Nearby enemies, allies, or map
           features
12     ...
```

*(b)* In RobotRumble, players' code must implement a `robot(state, unit)` function that returns an action each turn.

decided by a fixed clockwise rule, so careful bots choose their approach direction to gain the upper hand. Finally, practice *focus fire while avoiding friendly fire*: attacks only deal 1 damage but can hit teammates, so good bots coordinate multiple robots to bring down a 5-HP enemy in one turn without accidentally shooting their own.

**How are arena logs formatted?** RobotRumble logs are displayed as a sequence of ASCII grids (a total of 100 grids per simulation), with numbers marking robot positions and empty cells showing open space. After each turn, the grid is updated to show new movements, clashes, or unit spawns, giving a clear visual trace of how the battle unfolds. Below each grid, a summary line shows each player's remaining health and unit counts.

**What assets are provided in the initial codebase?** The initial codebase includes a command-line interface (CLI) tool (`rumblebot`) that allows users to execute battles between bots directly in the terminal or in a web-based graphical viewer. The repository also includes example "builtin bots" that can be used as opponents or templates for developing new robots. Additionally, the repo contains logic scripts and documentation for running matches, viewing results, and managing robot files within the filesystem.

**What are the arena configurations?** The arena configuration determines the battle environment—typically a rectangular map with fixed dimensions, where robots spawn in random or defined positions. Each robot operates in discrete turns, executing movement and attack commands according to its programmed logic. The arena setup remains consistent across matches to ensure fairness.

**How is the winner determined?** The winner in Robot Rumble is the last surviving team at the end of a match. Robots can deplete each other's health using attacks while avoiding incoming fire. If multiple robots remain when the time limit or round limit is reached, the winner is decided based on performance metrics such as remaining health or damage dealt.

---

**Example of RoboCode logs**

```
{"winner": "Red", "turns": [ {"state": { "objs": {
    "1": {"id": "1", "coords": [0,0], "obj_type": "Terrain", "type": "Wall"},
    "2": {"id": "2", "coords": [0,1], "obj_type": "Terrain", "type": "Wall"},
    "3": {"id": "3", "coords": [0,2], "obj_type": "Terrain", "type": "Wall"},
 ...
```

---

# C. Evaluation

In this section, we provide additional details about our evaluation procedure, including inference services, `mini-SWE-agent` configurations, arena-specific prompts, and formulae for calculating win rate and Elo scores.

## C.1. `mini-SWE-agent` Configuration

The `mini-SWE-agent` ACI allows one to define a number of configurations[2]. We highlight a couple of configuration settings relevant to the evaluation set up for CodeClash.

**Turn and cost limits.** For the *edit* phase of each round, the LM is constrained to at most 30 interactive turns with the codebase. We also impose a $1 cost limit, meaning once the running cost of input and output tokens for a single round exceeds $1, the editing episode is automatically terminated. Consequently, this means that for a tournament of $n$ rounds, at most $$n$ are spent per player. We enforce this cost limit not only to keep expenses manageable but also to discourage degenerate behaviors such as the model dumping entire files into its context, repeatedly echoing large outputs, or otherwise flooding the interaction buffer with irrelevant information. Generally, the limit forces the agent to allocate its context budget carefully, encouraging concise reasoning and selective use of code. We set the `mini-SWE-agent` configuration to the following values to enforce these practices:

- The `step_limit` is set to 30. The `cost_limit` is set to 1.

- In the `action_observation_template`, a prompt template that environment observations are interpolated into, the agent is reminded of the number of turns and cost consumed with the line:

  <limit_note>This is the output of step {{n_model_calls}} ({{step_limit}} limit). You've used {{model_cost | round(2)}} USD ({{cost_limit}} USD limit).<limit_note>

We observe in practice that the cost limit is almost never reached. On the other hand, turn limits are exhausted frequently for specific models.

**Setting the context.** The system prompt briefly sets the context and informs the model of the general nature of the setting it's operating in. Here is the prompt verbatim:

> ### System Prompt.
>
> You are a helpful assistant interacting continuously with a computer by submitting commands. You'll be editing a codebase to play a programming game.
>
> <important> This is an interactive process where you will think and issue ONE command, see its result, then think and issue your next command. </important>
>
> Your response must contain exactly ONE bash code block with ONE command (or commands connected with && or ||). Include a THOUGHT section before your command where you explain your reasoning process. Format your response as shown in <format_example>.
>
> <format_example> Your reasoning and analysis here. Explain why you want to perform the action.
> ```bash
> your_command_here
> ```
> </format_example>
>
> Failure to follow these rules will cause your response to be rejected.

The LM is informed it is acting in the role of a software developer with the ability to investigate and edit a codebase across multiple turns. The prompt clearly delineates an interaction protocol. Every turn, the model should be explaining its reasoning in a "Thought" section, followed by a `bash` code block.

---

[2]https://mini-swe-agent.com/latest/advanced/global_configuration/

**Describing the arena and tournament.** After the system prompt, the next message given to the LM briefly describes the arena and thoroughly reviews how the LM can interact with the codebase environment correctly. We first show the arena description:

---

### Subsection of initial message describing the arena

## Game Description

{{game_description}}

## General tips about how to play the game

The details of the game are fully available within this codebase.
- 'docs/': Game documentation
- 'logs/': Past rounds and outcomes
- 'trajs/': History of your edits
- and a lot more. It's up to you to explore and utilize these resources.

The game is played in rounds and you will be evaluated on the performance over all the rounds. You won't remember past rounds.

In every round, you have a limit of {{step_limit}} steps and a cost limit of {{cost_limit}} dollars. We will show you the number of steps and cost used so far after every response in the '<limit_note>' tag. After you've reached the step or cost limit, you cannot continue working on this task, and we will play the game with your codebase. This means that it's fine to reach the step or cost limit while working on documentation or testing, but you shouldn't reach the limit while working on the actual game logic to avoid submitting an invalid codebase.

So if you want to carry knowledge forward — leave tools, notes, or strategies in the codebase. Good documentation means you (and others) can pick up right where you left off.

If you'd hate to repeat a step next round, encode it now — as a script, a note, or a tool.

Improve the bot however you like — experiment, document, iterate. Some ideas:
- Build analysis tools
- Create bot variants to test
- Track strategies across rounds
How you choose to evolve and document is up to you. Good luck!

---

The actual description of the arena, represented by `game_description`, is brief. These are filled in by the system templates show in the arena cards of §B. This lack of detail is intentional. We impose the burden of understanding how exactly an arena works. With full access to documentation and logs in the codebase, CodeClash forces LMs to identify and fill in gaps about its understanding of the game. This obstacle is realistic. As prior work around coding evaluations has demonstration, real world software issues are often ambiguous and abstract on face value (Chowdhury et al., 2024). CodeClash enables investigating whether models can address such uncertainty by placing it in a setting where information is available, but not immediately obvious.

The second half of the prompt states the available assets, then reminds the model of both the step/cost limit along with the transient nature of its memory. The model is explicitly informed that its working memory is *not* retained across rounds, so it is encouraged to use the codebase to maintain long-term information, tools, and general progress. Collectively, the prompt incorporates the challenges discussed in Section 2.3.

Next, the prompt provides a deep dive into how the model should go about issuing actions. As a reminder, `mini-SWE-agent`'s interaction is completely terminal driven.

## Command Execution Rules

You are operating in an environment where

1. You write a single bash command
2. The system executes that command in a subshell
3. You see the result
4. You write your next command

For each of your response:

1. Include a THOUGHT section explaining your reasoning and what you're trying to accomplish
2. Provide exactly ONE bash command to execute
3. The action must be enclosed in triple backticks (see below for formatting rules)
3. Directory or environment variable changes are not persistent. Every action is executed in a new subshell. However, you can prefix any action with `MY_ENV_VAR=MY_VALUE cd /path/to/working/dir && ...` or write/load environment variables from files

Format your responses like this:

<format_example>
THOUGHT: Here I explain my reasoning process, analysis of the current situation, and what I'm trying to accomplish with the command below.

```bash
your_command_here
```
</format_example>

Commands must be specified in a single bash code block:
```bash
your_command_here
```

**CRITICAL REQUIREMENTS:**
- Your response SHOULD include a THOUGHT section explaining your reasoning
- Your response MUST include EXACTLY ONE bash code block
- This bash block MUST contain EXACTLY ONE command (or a set of commands connected with && or ||)
- If you include zero or multiple bash blocks, or no command at all, YOUR RESPONSE WILL FAIL
- Do NOT try to run multiple independent commands in separate blocks in one response
- Directory or environment variable changes are not persistent. Every action is executed in a new subshell.
- However, you can prefix any action with `MY_ENV_VAR=MY_VALUE cd /path/to/dir && ...` or write/load environ variables from files

We omit the examples of proper, well-formed interactions following this prompt. The examples include actions such as how to edit a file with `sed`, performing searches of the codebase with `grep` and `find`, and viewing specific parts of files with `nl`. We observe both with this work and prior evaluations (Jimenez et al., 2024) that including such in-context demonstrations is meaningfully helpful to reducing the errant actions issued by a model. All players' codebases are initialized with no tools provided upfront. However, throughout the course of a tournament, models are free to synthesize their own scripts and aliases.

**Errant action handling.** Last but not least, in the case that a model does issue an invalid action, we inherit the guardrail and error handling principles described in (Yang et al., 2024a) and inform the model of such errors. The `format_error_template` is shown when the model's response does not abide by the ReAct style form factor requested, and the following error message is displayed:

---

### Format error template

Please always provide EXACTLY ONE action in triple backticks, found {{actions|length}} actions. If you want to end the task, please issue the following command: `echo COMPLETE_TASK_AND_SUBMIT_FINAL_OUTPUT` without Any other command. Else, please format your response exactly as follows:

<response_example>
Here are some thoughts about why you want to perform the action.

```bash
<action>
```
</response_example>

Note: In rare cases, if you need to reference a similar format in your command, you might have to proceed in two steps, first writing `TRIPLEBACKTICKSBASH`, then replacing them with ```` ```bash ````.

---

Note that the error template is *not* thrown if the action itself is problematic or executes with a non-zero return code. This message is only invoked when the model's response doesn't abide by the expected format, and it does not account for any syntax issues or execution outcomes related to the `action` itself.

### C.2. Tournament Configuration

In addition to configuring interaction, we also allow users to set tournament settings, such as game mechanics and rounds, via a configurable `.yaml` file as well.

---

### Tournament configuration file for Battlesnake

```
tournament:
  rounds: 25
game:
  name: BattleSnake
  sims_per_round: 1000
  args:
    width: 11
    height: 11
    browser: false
```

---

The configuration file contains two sections. The `tournament` field allows one to specify how many `rounds` the tournament will be played. The `game` field indicates which code arena the tournament is being played in. `sims_per_round` is the number of simulations run per round in order to determine a winner (usually 1000). For most games, a simulation is run by calling an executable or script with arguments. The `args` field is a way to pass in flags to that executable to adjust the configurations of the arena. For instance, in the above example, the `args` are eventually interpolated into the following command to run the game: `python main.py --width 11 --height 11 --browser false`.

---

**Player configuration section**

```
players:
  - agent: mini
    name: p1
    config:
      agent: !include mini/default.yaml
      model:
        model_name: openai/gpt-5-mini
  - agent: mini
    name: p1
    config:
      agent: !include mini/default.yaml
      model:
        model_name: anthropic/claude-sonnet-4-20250514
```

---

The player configuration is simple, essentially serving as a meta-configuration for creating each player as an LM along with a `mini-SWE-agent` configuration. Using this configuration, it is possible to equip models with different prompts by swapping out the `mini-SWE-agent` configuration (`!include mini/default.yaml`), although we do not do this for our main leaderboard and results unless specified as otherwise.

**Number of rounds run.** To determine the number of tournaments and rounds to run to obtain a statistically meaningful leaderboard, we identify several parameters.

- $M$ for the number of models to evaluate.
- $A$ for the number of arenas we want models to compete in.
- $T$ for the number of tournaments we run per arena.
- $P$ for the number of players per tournament.
- $R$ for the number of rounds per tournament.

Given these values, we can generally calculate the number of rounds that would be run with $\binom{M}{P} \times A \times T \times R$. This assures us that each model is run against other models on the same set of arenas for the same number of total rounds ($T \times R$). The main results table reflects values of $M = 9$; $A = 6$; $T = 10$; $P = 2$; $R = 15$, giving us a total of 32,400 total rounds run, with each model playing a total $\binom{M-1}{P-1} \times A \times T \times R = 7200$ rounds. For the Section 4.1 evaluation with 3+ players, we use the same calculation to determine number of tournaments to run.

### C.3. Evaluation Metrics

This section contains detail on the evaluation metrics, in particular the Elo ratings for each model. Detailed statistical analysis shows that the ranking is stable. For example, the pairwise order agreement of our ranking is more then 98% in bootstrapping experiments.

#### C.3.1. DEFINITIONS

**Tournaments** are a sequence of 15 rounds played in one arena between two or more models.

**Winning a round.** A round consists of one or more repetition of an arena between the submissions of different models. A round is won by a model if any of the following applies

1. The model is the only one with a valid submission (for example because the other model's submission does not compile or execute)
2. The model scores higher than all others. Scores a typically either win rates (across all repetitions of the arena), or other aggregate quantities (e.g., total amount of money won in poker).

Distributions of round scores for different arenas are shown in Figure 16. Because of the sequential nature of a tournament,

**Distribution of Normalized Player Scores by Game (Valid Rounds Only)**

*Figure 16.* Distribution of rounds scores by game.

the scores of the rounds are not independent of each other. This is shown in Figure 17: If all rounds were independent, a uniform distribution would be expected. However, most games show a heavily bimodal distribution instead.

**Winning a tournament** A tournament is won by the model that wins more rounds than its opponent, or, if both models win equally many rounds, by the model that scores the last win. If all rounds of the tournament are draws, then the tournament is a draw (an extremely rare occurrence, less than once per 1000 tournaments).

**Win rate** per model is the fraction of tournaments won. This metric can be further stratified into arena and opponent-specific percentages.

**Elo rating.** We quantify absolute model strengths by Elo ratings.

Elo ratings are based on the Bradley-Terry model (Bradley & Terry, 1952) that models win probabilities between two players $i$ and $j$ with strengths $s_i$ and $s_j$ via logistic regression of the strength difference $s_i - s_j$, i.e.,

$$P(\text{model } i \text{ wins over } j) = \frac{1}{1 + \exp(s_i - s_i')} = \sigma(s_i - s_i').$$

Repetitions of independent games are Bernoulli-distributed and the optimal values of $s_i$ and $s_j$ can be calculated using a maximum likelihood fit to the win numbers $w_{ij}$ (number of times $i$ won over $j$), i.e.,

$$\log \mathcal{L} = \sum_{i<j} \Big[ w_{ij} \log \sigma(s_i - s_j) + w_{ji} \log \sigma(s_j - s_i) \Big]. \tag{1}$$

However, this leaves a gauge freedom in the strengths $s_i$, because all $s_i$ can be shifted by a constant factor $s_i \rightarrow s_i + S$ without changing the value of $\mathcal{L}$. To fully constrain the fit, we choose $\sum_i s_i = 0$. This choice only results in a fixed offset for the final Elo scores. Log likelihood profiles for a fit to all arenas are found in Figure 18.

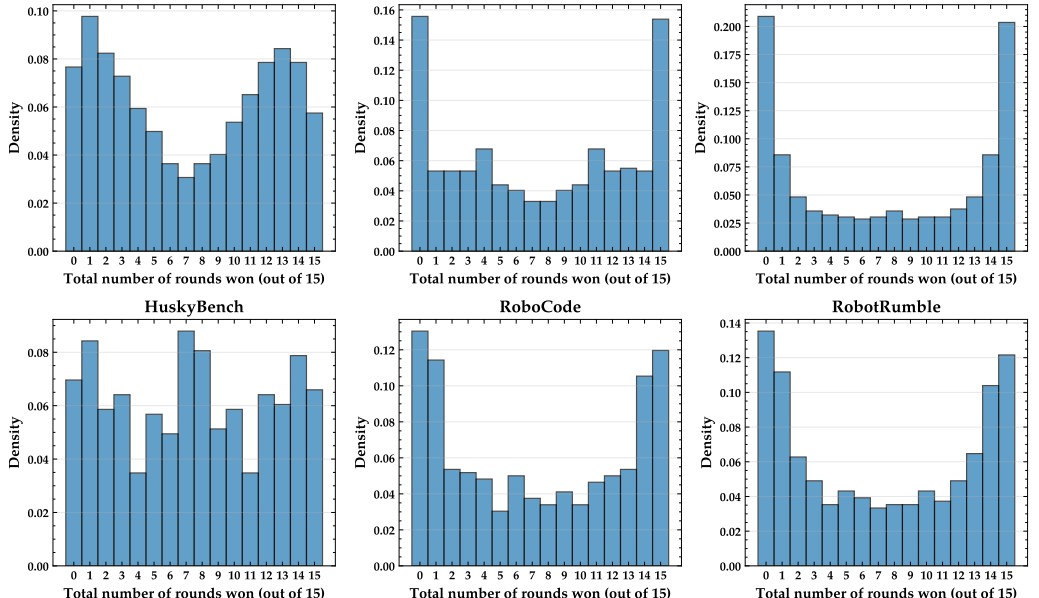

*Figure 17.* Distribution of the number of rounds won by the players across arenas. The non-uniform distributions demonstrate that the rounds are not independent of each other.

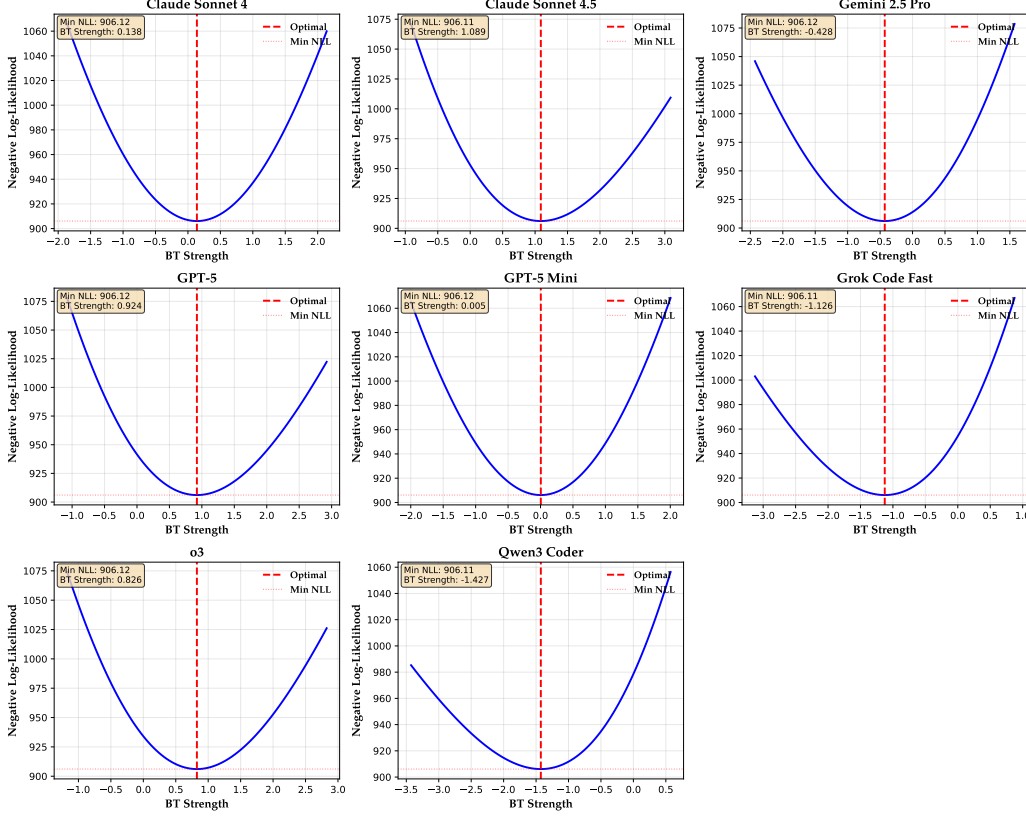

*Figure 18.* Log likelihood profiles for a fit to all arenas results.

*Table 4.* ELO ratings with uncertainties

| Model | BattleSnake | CoreWar | Halite | Poker | RoboCode | RobotRumble | All |
|---|---|---|---|---|---|---|---|
| Claude Sonnet 4.5 | $1470 \pm 52$ | $1641 \pm 73$ | $1408 \pm 50$ | $1248 \pm 44$ | $1361 \pm 43$ | $1423 \pm 47$ | $\mathbf{1389 \pm 18}$ |
| GPT-5 | $1339 \pm 44$ | $1199 \pm 43$ | $1522 \pm 56$ | $1599 \pm 64$ | $1409 \pm 46$ | $1293 \pm 41$ | $\mathbf{1360 \pm 17}$ |
| o3 | $1357 \pm 45$ | $1348 \pm 47$ | $1576 \pm 60$ | $1277 \pm 46$ | $1338 \pm 43$ | $1309 \pm 42$ | $\mathbf{1343 \pm 17}$ |
| Claude Sonnet 4 | $1253 \pm 46$ | $1339 \pm 46$ | $1111 \pm 48$ | $1233 \pm 44$ | $1033 \pm 45$ | $1361 \pm 43$ | $\mathbf{1223 \pm 16}$ |
| GPT-5 Mini | $1369 \pm 45$ | $926 \pm 50$ | $1185 \pm 47$ | $1429 \pm 50$ | $1217 \pm 41$ | $1092 \pm 41$ | $\mathbf{1200 \pm 16}$ |
| Gemini 2.5 Pro | $1115 \pm 45$ | $1043 \pm 45$ | $1186 \pm 47$ | $978 \pm 48$ | $1315 \pm 42$ | $1044 \pm 44$ | $\mathbf{1125 \pm 16}$ |
| Grok Code Fast | $833 \pm 63$ | $1170 \pm 43$ | $824 \pm 63$ | $886 \pm 54$ | $1033 \pm 45$ | $1016 \pm 46$ | $\mathbf{1004 \pm 18}$ |
| Qwen3 Coder | $860 \pm 59$ | $929 \pm 51$ | $784 \pm 67$ | $945 \pm 53$ | $890 \pm 55$ | $1057 \pm 43$ | $\mathbf{952 \pm 20}$ |

The player strengths can be converted to Elo scores $R_i$ as

$$R_i = R_0 + \frac{\beta}{\log 10} s_i, \tag{2}$$

Following the conventions from Chess, we choose a starting Elo of $R_0 = 1200$ and a slope of $\beta = 400$. Note that this convention is merely a presentation choice that affects readability, not the model predictions (unlike the $K$ factor that is used in sequential calculation of Elo scores).

### C.3.2. STATISTICAL UNCERTAINTIES

The covariance matrix $\Sigma$ of the player strengths $s_i$ is given by the inverse of the Hessian matrix of $\log \mathcal{L}$. Setting $p_{ij} = \sigma(s_i - s_j)$ and $n_{ij} = w_{ij} + w_{ji}$, the Hessian of $\mathcal{L}$ is given by

$$H_{ij} = \frac{\partial^2 \log \mathcal{L}}{\partial s_i \, \partial s_j} = -\sum_{i<j} n_{ij} p_{ij} (1 - p_{ij}) \begin{cases} 1 & i = j, \\ -1 & i \neq j. \end{cases}$$

However, this Hessian is singular, due to the above mentioned shift-invariance. So we invert $H$ in the constrained subspace of our gauge, $\mathcal{S} = \{s_i \mid \sum_i s_i = 0\}$, i.e., calculate the covariance $\Sigma$ as

$$\Sigma = Z(Z^T H Z)^{-1} Z^T,$$

where $Z$ projects onto $\mathcal{S}$ and is given by

$$Z_{ij} = \begin{cases} 1 - \frac{1}{n} & i = j, \\ -\frac{1}{n} & i \neq j. \end{cases}$$

The variance of $s_i$ is then given by $\mathrm{Var}\, s_i = \Sigma_{ii}$ and can readily be scaled to the variance on $R_i$ via (2). The uncertainties of the final results are shown in Table 4.

### C.3.3. STATISTICAL VALIDATION AND RANK STABILITY

We perform non-parametric and parametric bootstrapping experiments to test the stability of the ranking. Distribution of bootstrapped Elo scores are shown in Figure 19a, and the resulting distribution of ranks are shown in Figure 19b. The statistical uncertainties derived from the bootstrapped Elo results agree well with those calculated from the Hessian matrix in Table 4. Various rank stability metrics are shown in Table 5. In particular, we'd like to highlight that the pairwise order agreement of our ranking is 98%.

**Non-parametric bootstrapping** We perform a non-parametric bootstrapping experiment by sampling with replacement from all tournaments. This results in new win counts $w_{ij}$ from which we can calculate new Elo rankings $R_i$. We draw 1000 samples and calculate rank stability metrics and uncertainties based on the 1000 corresponding Elo rankings.

**Parametric bootstrapping** We generate bootstrap replicas from the fitted Bradley–Terry model, i.e., we use the Bradley-Terry player strengths $\hat{s}_i$ that maximize (1) and assume win probabilities

$$p_{ij}^\star = \sigma(\hat{s}_i - \hat{s}_j).$$

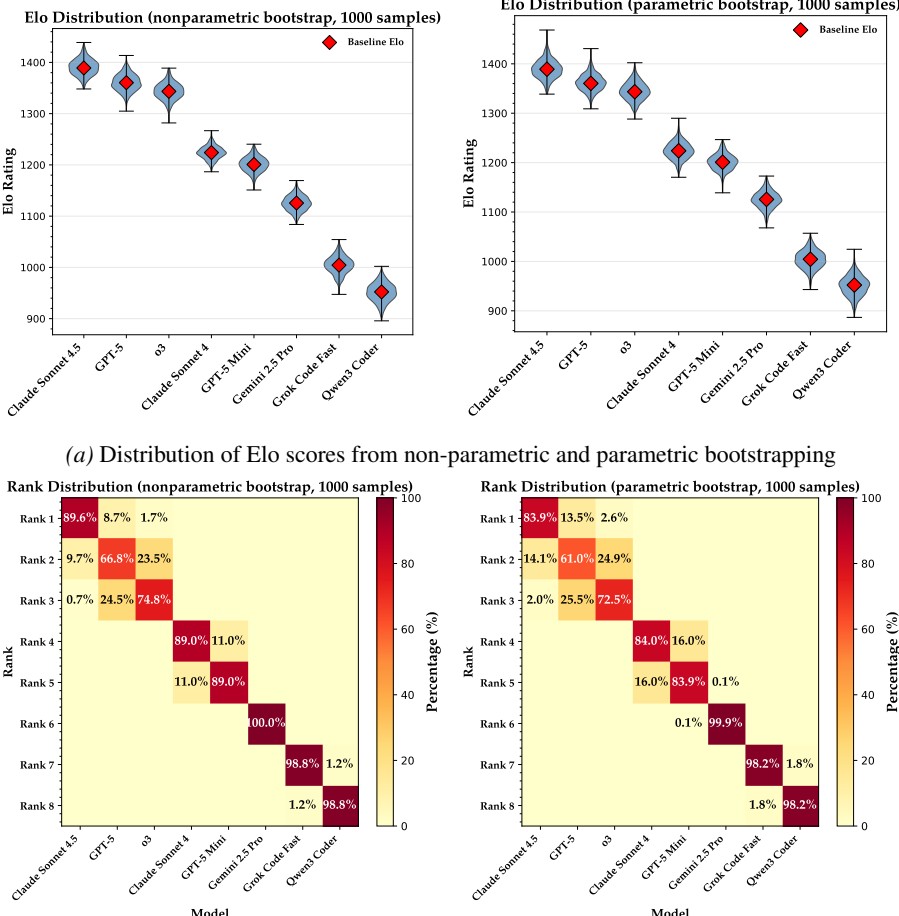

*(a)* Distribution of Elo scores from non-parametric and parametric bootstrapping

*(b)* Elo-based ranks from non-parametric and parametric bootstrapping

For each observed matchup $(i, j)$ with $n_{ij} = w_{ij} + w_{ji}$ total games, we then draw

$$\tilde{w}_{ij} \sim \text{Binomial}(n_{ij}, p_{ij}^\star), \qquad \tilde{w}_{ji} = n_{ij} - \tilde{w}_{ij}.$$

We preserve the observed matchup graph and game counts, resample outcomes from the fitted model, refit Bradley–Terry (converting to Elo via (2)), and assess score and rank variability across 1000 replicas.

| Metric | Nonparametric | Parametric |
|---|---|---|
| Kendall's $\tau$ | 0.966 | 0.956 |
| Spearman's $\rho$ | 0.988 | 0.984 |
| Footrule (normalized) | 0.030 | 0.038 |
| Top-1 consistency | 0.896 | 0.839 |
| Pairwise order agreement | 0.983 | 0.978 |

*Table 5.* Rank stability metrics of the Elo-based ranking of LMs over all arenas based on bootstrapping experiments

# D. Extended Results

In this section, we present additional analyses and findings not presented in Section 4. These insights further characterize model behavior and performance in the CodeClash setting.

## D.1. Interaction Trends

We provide additional analyses and visualizations revealing trends in how different models interact with their codebase, such as how many steps they take per round, the size and frequency of their edits, and their length of their thoughts.

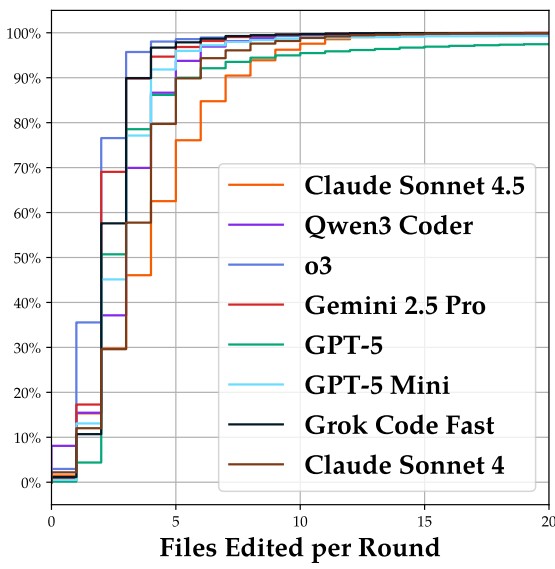

*(a)* CDF of files edited per round by each model. Some models typically never edit more than 5 files (o3, Gemini 2.5 Pro). Others tend to create many more (Claude Sonnet 4.5, GPT-5)

*(b)* Average lines changed per round per model. Some models are fairly consistent (Gemini 2.5 Pro), while others vary; Qwen3-Coder edits more in later rounds. GPT-5 Mini's edits occur early on.

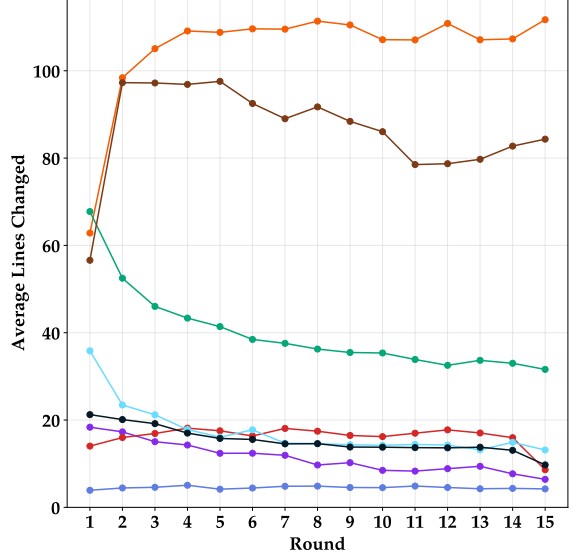

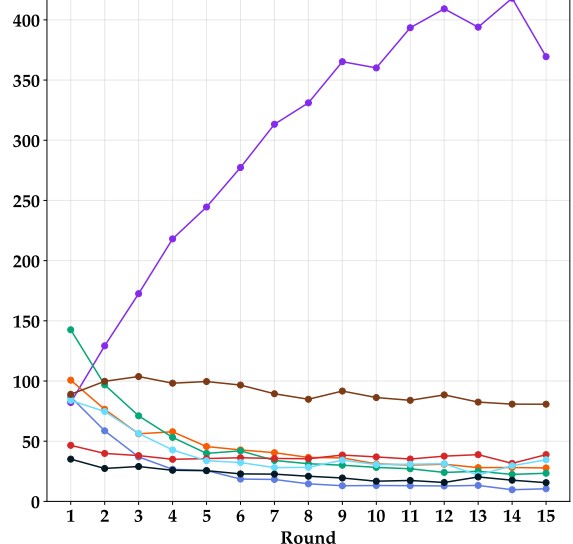

*(c)* Average lines changed per round per model for the `README_agent.md`, a file we suggest agents write important information to. The Anthropic family of models write copious amounts of notes – other models tend to add more brief summaries.

*(d)* Average lines changed per round per model for game-playing related functionality (e.g. `warrior.red` in Core War). Models typically make the majority of their changes early on, with a steady decline in later rounds as changes become more targeted.

**Models differ in the number of files created or edited.** As shown in Figures 20a and 20b, we observe that models vary significantly in the number of files and lines changed per round. The range varies significantly, with more conservative models such as o3 or Gemini 2.5 Pro editing just two to three files and less than a hundred lines per round. On the other end, Claude Sonnet 4.5 or GPT-5 generally make larger changes, with a much longer tail of sizable modifications. We observe that this long tail typically comes from when models initialize test suites, create multiple versions of a submission to test against one another, or record insights as markdown notes to take forward into the next round.

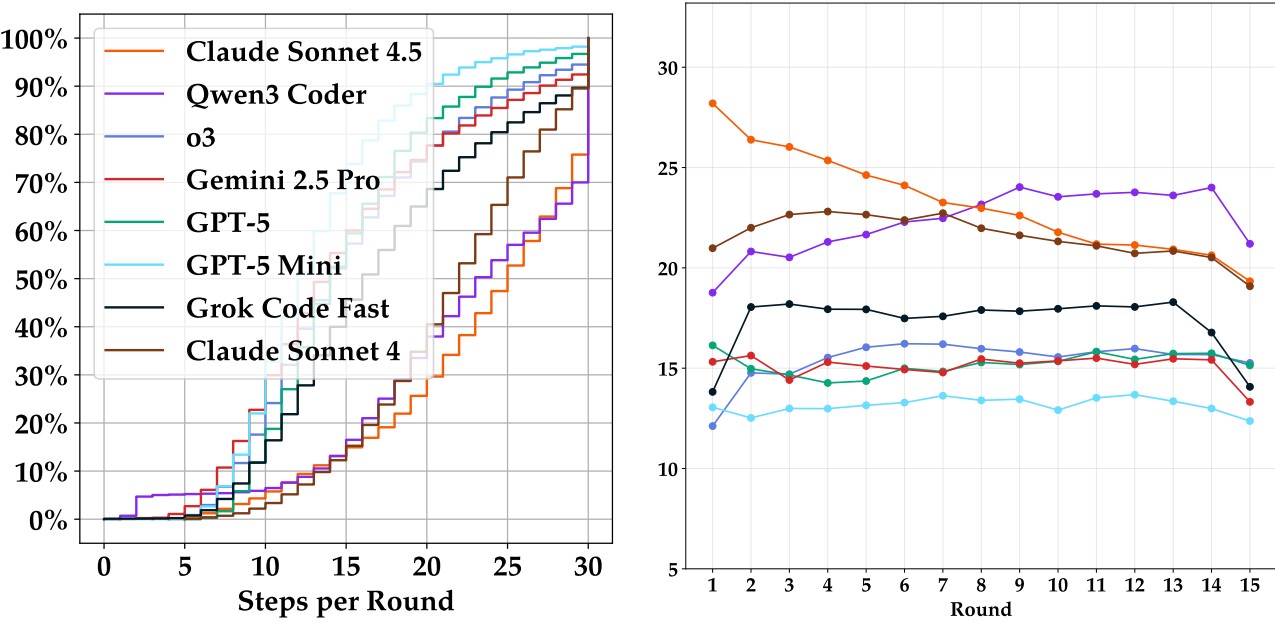

*Figure 21.* CDF of number of steps taken per round per model. The Anthropic family of models along with Qwen3 Coder usually consumes more of the allotted step budget.

*Figure 22.* Average steps taken for each round per model. The chart reflects similar conclusions as Figure 21, and also suggests that steps used are fairly steady.

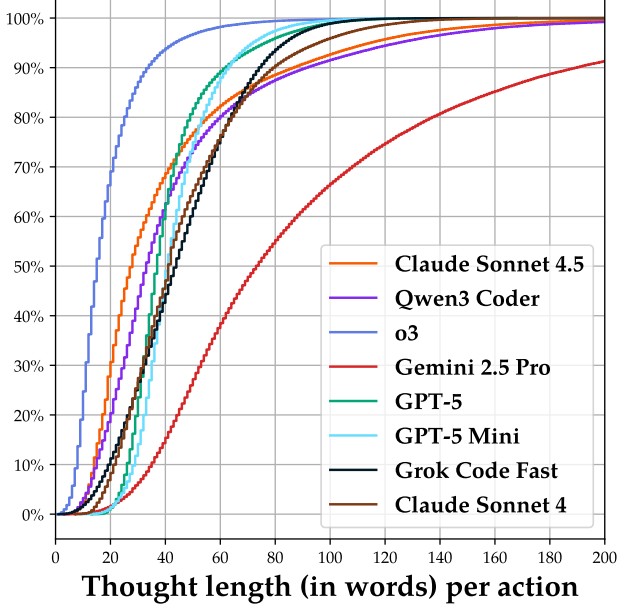

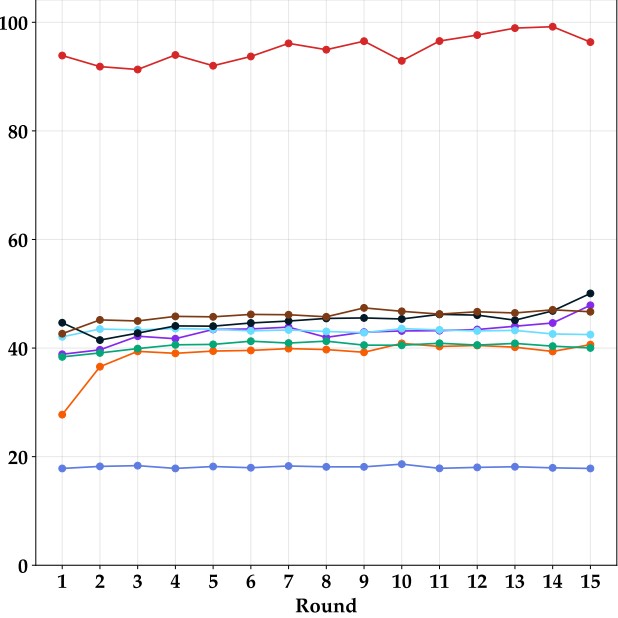

*Figure 23.* CDF of thought length (in words) per model. The thought lengths are computed per model response. Our calculation does not consider the action portion of the response.

*Figure 24.* Average thought length (in words) per model response at each round. While most models fall within the range of 35 to 55 words per response, Gemini 2.5 Pro and o3 are notable outliers.

We include two additional similar line charts that show the size of edits for the README_agent.md file (Figure 20c along

with any game-playing related core functionality in Figure 20d. The Claude Sonnet 4 and Claude Sonnet 4.5 models are relatively more extensive in their documentation. GPT-5 and GPT-5 mini exhibit a trend, where they take more notes up front, with a gradual decline into later rounds. The remaining models do not fluctuate significantly in the amount of notes they take, with o3 averaging under 10 lines changed per round. Model changes to competition logic generally trends downward across rounds – we generally observe that models define the majority of competitive logic early on, with later rounds consisting mostly of smaller, more specific adjustments.

**Models differ in the number of steps taken.** We provide Figures 21 and 22 to showcase trends around the number of turns consumed by each model for each round. Turn budget consumption is markedly different between models, with the Anthropic models and Qwen3 Coder usually using 22 to 27 turns out of the 30 turn limit. On the other end, Gemini 2.5 Pro and GPT-5 mini rarely exceed 15 turns. Figure 22 suggests that the number of steps models take from round to round is fairly steady; we were not able to identify any meaningful discrepancies in steps taken between rounds that might be due to trends such as To further clarify – although we impose the $1 per-round cost limit, there are *zero* occurrences across all tournaments we run of a model's trajectory being automatically terminated due to models exceeding the cost limit budget. In other words, this means that the cost limit trend lines also faithfully reflect when models decide for themselves to stop editing for the round. The majority of rounds end with a model producing a thought and action akin to "I have made all the changes I think are necessary. I will now conclude this round [`END` action]".

**Models differ in thought length.** As shown in Figures 23 and 24, we find that while most models respond with similarly long thought traces, Gemini 2.5 Pro responds with significantly longer explanations, at around 95 words per response. On the other end, o3 is much more terse, with just under 19 words per response. However, o3's brevity comes with a heavy asterisk, as OpenAI's API is configured to hide intermediate thinking tokens for the `o`-series reasoning models. The actual token count is thus likely vastly underestimated.

**Models are quick to recover from errant actions.** As discussed in Section 4, errant actions is not a significant factor in model performance. The vast majority of actions ($\geq 90\%$) are well formed and execute successfully. In addition to the statistics we presented before, we also provide a breakdown of the errant action rates by model and arena in Figure 25. We find that stronger models have slightly lower error rates, with Claude Sonnet 4 at just 10.11%, while `Qwen3 Coder` tops out at `16.32%`. No arena has a particularly high errant action rate.

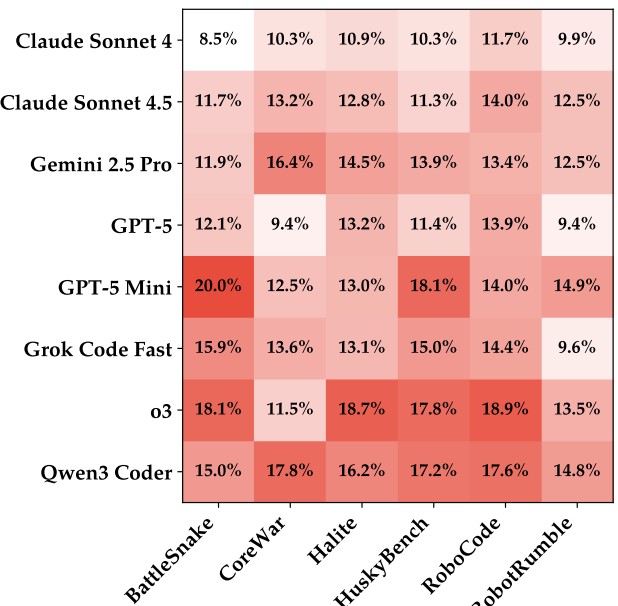

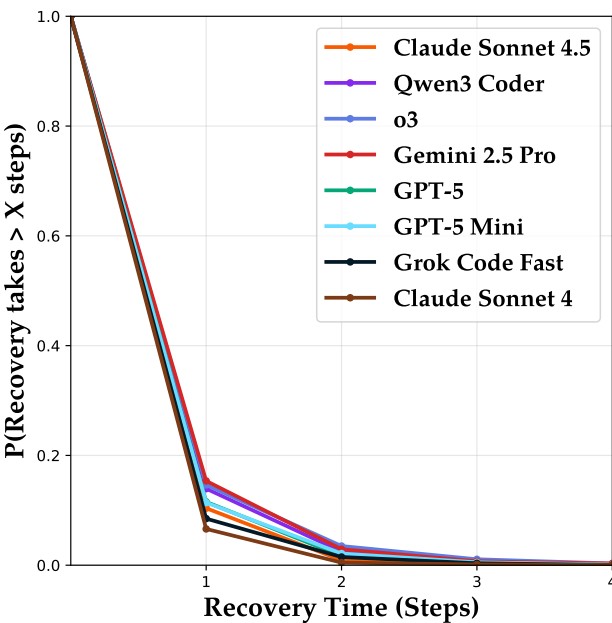

*Figure 25.* A heatmap of errant action rates for models in different arenas. "Errant" means the action resulted in `returncode == 0`. We find that malformed actions does *not* constitute a significant reason for why models might struggle in CodeClash.

*Figure 26.* "Recovery time" is the number of steps between a failed command (`returncode != 0`) and the next successful command (`returncode == 0`). Each data point indicates the likelihood that recovery requires more than x steps for a model.

Furthermore, we also answer how quickly models recover from errant actions. Prior work has reported that a major

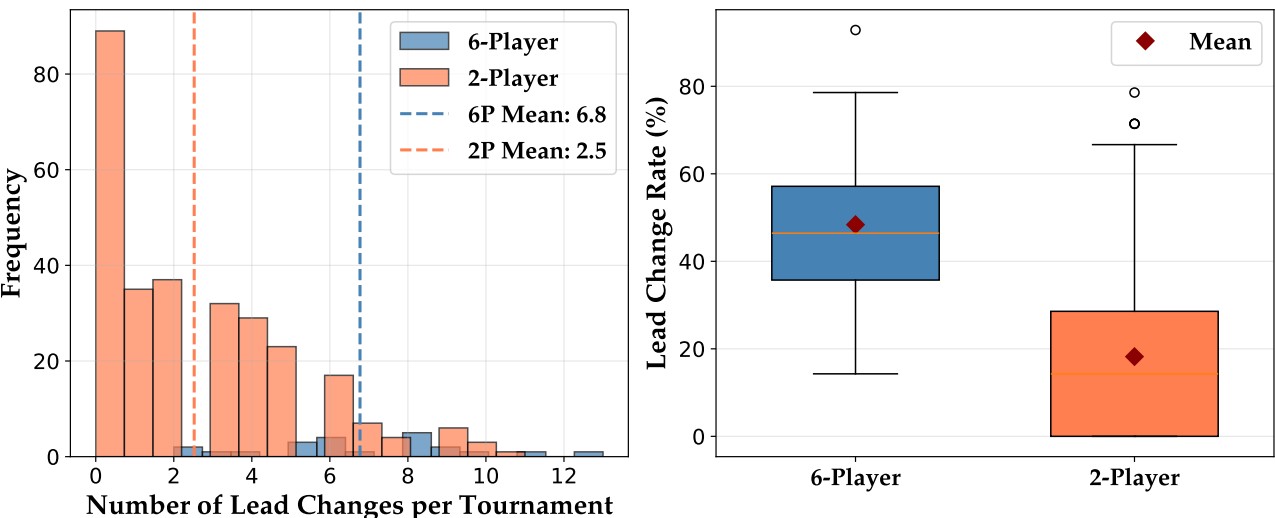

*Figure 27.* Lead change rate comparison. A "lead change" is defined as a round `n` where the winner is different from the round `n-1` winner. We make comparisons between 2-player and 6-player tournaments specifically for the Core War arena.

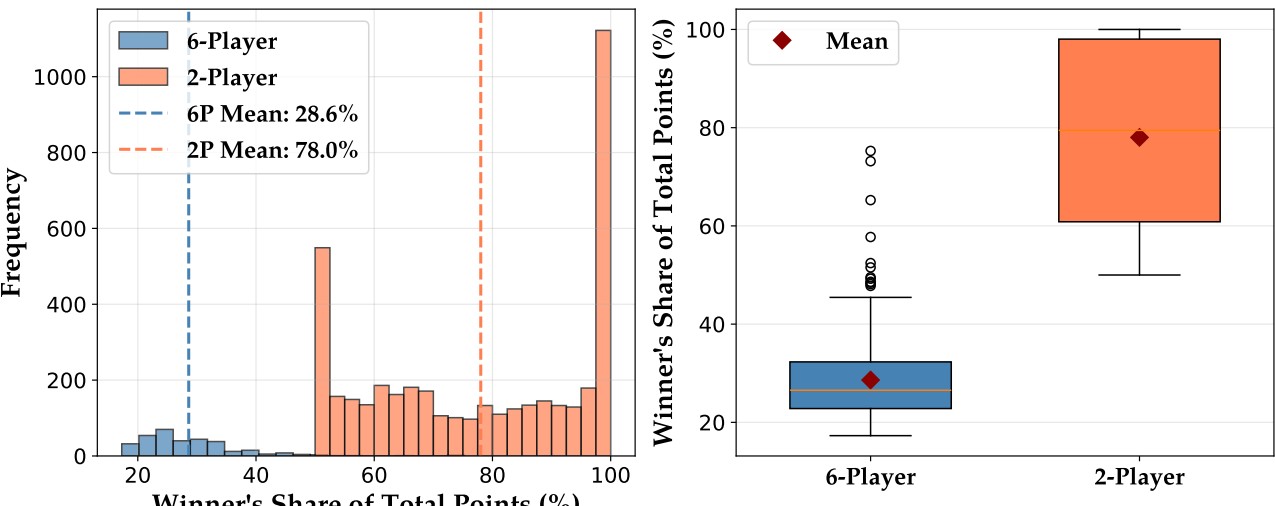

*Figure 28.* Win share comparison. We define "'win share" as the percentage of total points taken by a particular player. Win share per player is much lower with more opponents.

error mode of existing models are "cascading" failures – if a model issues an errant action, the likelihood that it recovers successfully from the mistake decreases with every subsequent action (Yang et al., 2024a; Pan et al., 2025). In the year since these works pointed out this phenomenon, we find that such breakdowns have diminished significantly in frequency and length. We visualize this finding with Figure 26. We observe that following an errant action, the next action is successfully more than 80% of the time. By the third step following an errant action, there are nearly zero occurrences of models continuing to struggle to generate a well formed action. In summary, our analyses strongly suggest that model performance in CodeClash is neither hindered by the choice of agent framework, nor that models are not adept at operating on the command line.

### D.2. Additional Analyses

**Multi-player settings are far more variable in standings.** As mentioned in our results and analyses section in the main paper, we showcase the ability to run multi-player (3+) tournaments in CodeClash, specifically with the Core War arena. As shown in Table 3, four additional arenas – BattleSnake, Halite, Poker, and RoboCode – all support more running tournaments

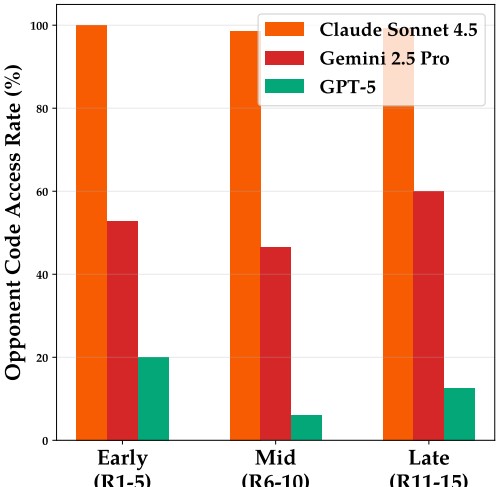

| Model | $\mu$ |
|---|---|
| Claude Sonnet 4.5 | $28.38 \pm 0.65$ |
| o3 | $27.11 \pm 0.64$ |
| Grok Code Fast | $25.65 \pm 0.65$ |
| GPT-5 | $24.76 \pm 0.64$ |
| Gemini 2.5 Pro | $23.62 \pm 0.65$ |
| Qwen3 Coder | $22.30 \pm 0.66$ |

*(b)* TrueSkill ratings per model based on 20 tournaments of 6-player Core War. TrueSkill models each player's skill as a Gaussian distribution with mean $\mu$ (skill estimate) and standard deviation $\sigma$ (uncertainty). After each round, both parameters are updated based on match outcomes: winning increases $\mu$ while exceeding expectations, and $\sigma$ decreases as the system gains confidence in the estimate. Final placement (1st, 2nd, ..., 6th) determines rating updates.

*(a)* Share of rounds which a model inspects its opponent's codebase. We find variance across models and round ranges.

with 2 players, though we do not run comprehensive experiments due to both cost limitations and the analytical complexity introduced by multi-way competition, which we believe is best left as future work. To illustrate the difference in competitive volatility, we provide Figure 27, revealing that lead changes are much more frequent as there are more players. Furthermore, winners occupy a much smaller share of the total points in the 6 player arena compared to the head-on setting.

**Transparent codebases enable investigations in how models leverage views into others' development processes.** We elected to run tournaments for CodeClash's main results under the assumption that models cannot view opponents' code because such a setting is more reminiscent of real world settings, where human players develop their solutions independently and have the option to keep their codebase closed source. Therefore, we investigate the effects of making players' codebases viewable by opponents specifically as an ablation.

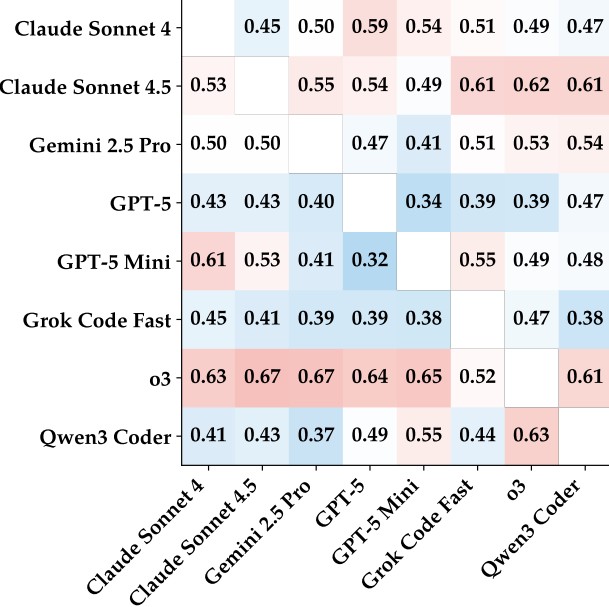

*Figure 30.* Code similarity of models' codebases with respect to each opponent for round 1 of BattleSnake (10 samples each).

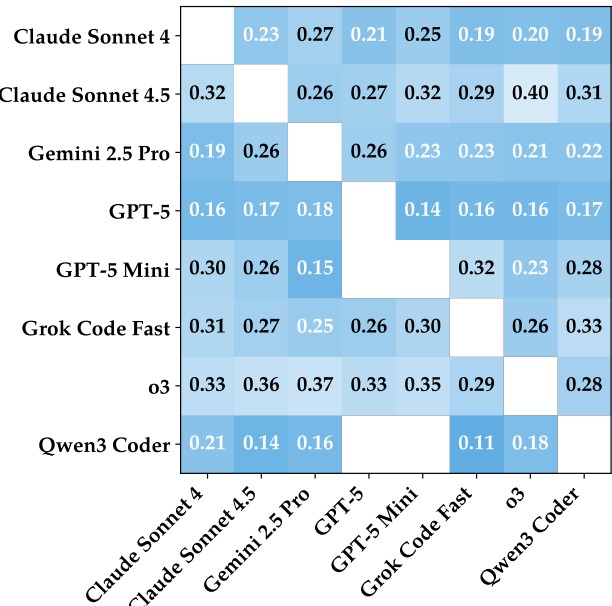

*Figure 31.* Code similarity of models' codebases with respect to each opponent for round 1 of BattleSnake (10 samples each).

The introduction of this mechanic is potentially interesting as it shifts CodeClash much closer towards being a perfect

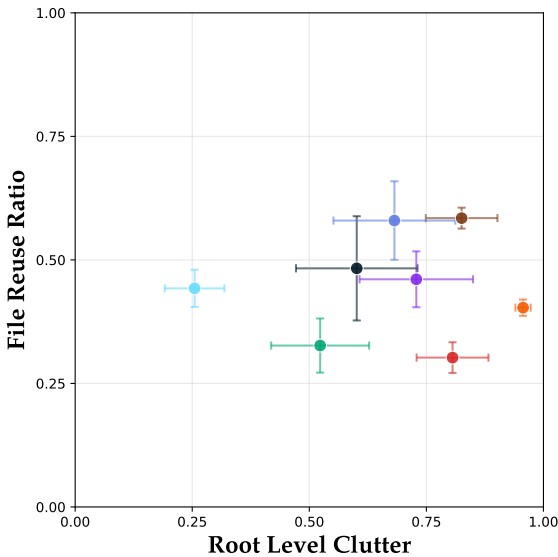 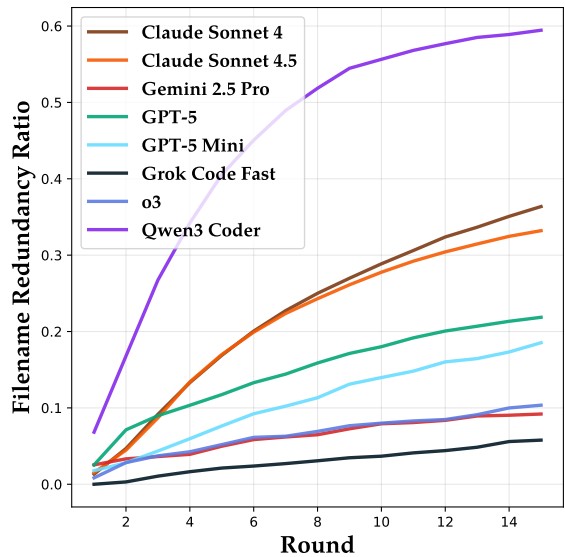

*(a)* Scatter plot of file reuse ratio and root level clutter with error bars. The top left quadrant represents most desirable practices (high file reuse, low root level clutter).

*(b)* Line chart of redundancy rate in filenames across rounds per model. Models increasingly create files with similar names as tournaments progress.

information game (Fudenberg & Tirole, 1991), where all players in a game have knowledge of all relevant information in the system, including other players' decisions. The knowledge of opponents' moves is what distinguishes a perfect information game like chess from an imperfect information game like poker, where opponent private cards are not known by default.

As mentioned in the main results, we carry out this investigation specifically for the Halite arena with three models (GPT-5, Claude Sonnet 4.5, Gemini 2.5 Pro). From Figure 29a, we found that the rate at which a player checks its opponent codebase fluctuates across both models and the phase of the tournament. Claude Sonnet 4.5 is near constant, checking in on its opponent's activity nearly every single round. Gemini 2.5 Pro and GPT-5 both exhibit a trend where the check rate dips somewhat in the middle of a tournament before re-surging in later rounds.

**Models codebases are highly diverse, even when playing against the same opponent in the same arena.** Continuing our discussion in Section 5.1, we provide additional visualizations demonstrating how codebases evolve over time, as shown in Figures 30 and 31. Each cell of the heatmap corresponds to the similarity score across 10 code samples generated by `model A` at round `n` from 10 tournaments of [`model A`, `model B`, `BattleSnake`]. So for instance, the top right cell is how similar 10 samples of `main.py` written by Claude Sonnet 4 were during round 1 of tournaments playing BattleSnake against `Qwen3 Coder`. To clarify further, these cells do *not* correspond to similarities between submissions generated by different models. The x-axis indices simply denote who the y-axis model's opponent was.

In round 1, we can see that model's solutions are already quite divergent. Claude Sonnet 4.5 and o3 tend to start off similarly, with the highest round 1 scores of 0.566 and 0.626 respectively. What this chart also tells us, is that the opponent doesn't seem to have too much of an impact on how similarly a model starts a tournament. By round 15, models' solutions are unalike across the board, with GPT-5 still maintaining the trend of being most diverse in its solutions (0.409 in round 1 to 0.163 by round 15). Affirming our original claim, we find that model solutions are creative, even when facing the same opponent in the same arena multiple times.

**Model codebases become increasingly disorganized with time.** Continuing our discussing from Section 5.1, we show two additional charts to showcase trends in how LM managed codebases tend to become more scattered and redundant with time. In Figure 32a, we plot root level clutter and file reuse metrics as ratios. A higher root level clutter ratio (*files created in root / files created*) suggests that models are not expending effort or commands to organize files into aptly named subdirectories. A lower file reuse ratio (*file reused at least once again after being created  files created*) suggests that instead of building on prior scripts and generating re-runnable code, models are creating a lot of single use files. Therefore, in our framing, desirable coding practices correspond to the top left quadrant (high file reuse, low root level clutter), while undesirable behaviors are in the bottom right (low file reuse, high root level clutter). As we see from the chart, 5 of 8 models fall in the bottom right corner. Claude Sonnet 4.5 shows the highest root level

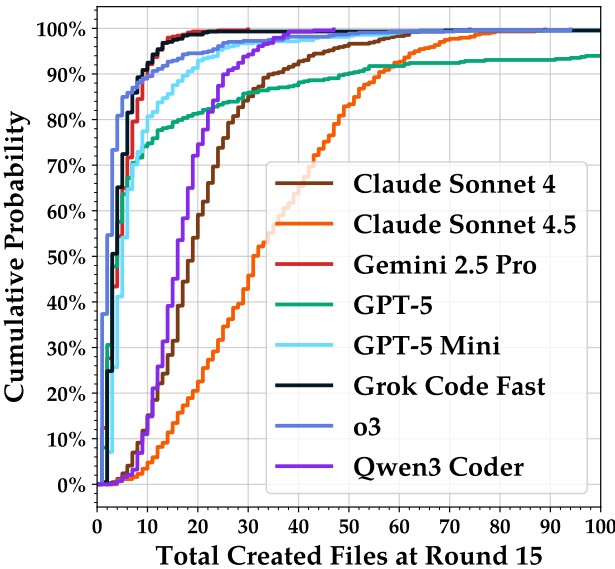

*Figure 33.* Cumulative probability density function of the number of files created during a tournament. While Claude Sonnet 4.5 consistently creates more files than the other models, GPT-5 reaches a high average number of created files because of an extreme number of output files in the CoreWar arena that are not cleaned up.

ratio. We provide a randomly selected example of a codebase produced by Claude Sonnet 4.5 at the end of a 15 round tournament of BattleSnake, playing against Gemini 2.5 Pro, in Figure 34. The tournament ID is `PvpTournament.BattleSnake.r15.s1000.p2.claude-sonnet-4-5-20250929.gemini-2.5-pro.251002020143`.

As discussed in the main results, we notice that codebases tend to follow this trend of creating single use analysis and testing files that are then rarely reused later on in a tournament. While we do not explore mitigating such behavior with prompting, we purport that this result is still noteworthy. Refactoring and sustaining a well organized codebase is not something that models organically aspire towards. CodeClash can serve as a testbed for investigating how LM managed codebases morph over time and exploring whether interventions in the form of data or external rewards can encourage better practices.

Finally, with Figure 32b, we find that the number of redundantly named files climbs upwards at different rates across all models. Figure 34 gives us a concrete example. Claude Sonnet 4.5 creates 13 files with the prefix "`analyze_`". From manual inspection, we found that most of these implementations are doing the same thing, with only the log file path being different. The same trend holds for the "`check_`" and "`ROUND_`" files. Such redundancy points to obvious room for improvement. Long running SWE-agent's that iterate and reuse a core set of files rather than spamming the codebase with single use scripts should be the more desirable behavior in the vast majority of use cases.

Showing **52 changed files** with **5,824 additions** and **14 deletions.**   Split  Unified

| File | Changes |
|------|---------|
| README_agent.md | +92 −0 |
| ROUND14_SUMMARY.md | +25 −0 |
| ROUND3_SUMMARY.md | +47 −0 |
| ROUND5_SUMMARY.md | +34 −0 |
| ROUND9_SUMMARY.md | +44 −0 |
| analyze_death_causes.py | +92 −0 |
| analyze_games.py | +91 −0 |
| analyze_loss_game.py | +41 −0 |
| analyze_losses.py | +109 −0 |
| analyze_round13.py | +125 −0 |
| analyze_round13_v2.py | +148 −0 |
| analyze_round13_v3.py | +141 −0 |
| analyze_round4_deaths.py | +48 −0 |
| analyze_round7_detailed.py | +53 −0 |
| analyze_round7_results.py | +35 −0 |
| analyze_round8.py | +102 −0 |
| analyze_round9.py | +101 −0 |
| analyze_specific_loss.py | +66 −0 |
| check_collision.py | +48 −0 |
| check_opponent_move.py | +35 −0 |
| check_our_move.py | +54 −0 |
| check_position.py | +31 −0 |
| compare_round7_round8.md | +57 −0 |
| compare_strategies.py | +86 −0 |
| debug_move_decision.py | +171 −0 |
| examine_death.py | +71 −0 |
| examine_death.py | +71 −0 |
| examine_loss.py | +46 −0 |
| find_losses.py | +48 −0 |
| find_ties.py | +26 −0 |
| main.py | ·200 −14 |
| main_backup.py | +187 −0 |
| main_round10.py | +302 −0 |
| main_round11_backup.py | +272 −0 |
| main_round11_reverted_to_r9.py | +272 −0 |
| main_round13_buggy.py | +272 −0 |
| main_round2_failed.py | +276 −0 |
| main_round3_backup.py | +187 −0 |
| main_round4_failed.py | +187 −0 |
| main_round5_perfect.py | +187 −0 |
| main_round7_simple.py | +187 −0 |
| main_round8_space_aware.py | +260 −0 |
| main_round9_health_aware.py | +272 −0 |
| round_13_summary.md | +48 −0 |
| test_bot.py | +53 −0 |
| test_edge_cases.py | +96 −0 |
| test_fixed_move.py | +161 −0 |
| test_flood_fill.py | +43 −0 |
| test_move_decision.py | +77 −0 |
| test_new_logic.py | +86 −0 |
| test_round4.py | +54 −0 |
| test_simple.py | +34 −0 |
| test_turn1_sim539.py | +44 −0 |

*Figure 34.* Screenshot of the 52 files created by Claude Sonnet 4.5 by the 15th round of a BattleSnake tournament. Several files are created for the purpose of notes, analyses, unit testing, and backups of the main bot.

## D.3. Analyzing trajectories using LMs as a judge

This sections describes detailed observations about the agent trajectories that were obtained using a LM as a judge setup.

### D.3.1. ADDITIONAL RESULTS

The data on the groundedness of edits, hallucinations, and validation efforts that were presented in Figure 8 are shown for the different arenas in Figures 35 and 36. Notably, models behave very different across arenas. For example, BattleSnake elicits very strong hallucinations from Claude Sonnet 4.5 (affecting up to 45% of rounds), and RoboCode shows a particularly low rate of edit validation across models.

Figure 37 shows how the kinds of edits that models perform changes between rounds. While the initial editing of models is feature-heavy, as the tournament progresses, a larger amount of smaller tweaks or fixes appears together with rounds in which no meaningful edit was made to the main player file.

Figure 38 shows what models spend their turn on early in the tournament and late in the tournament. This figure not only shows how the average number of actions in a round varies between models, but also that read operations increase as the tournament progresses. It is also apparent how different the number of actions spent on testing, analyzing, and running test matches is between models. Models vary in edit persistence: Gemini 2.5 Pro and o3 largely stop editing after round 4–6, while Claude and GPT-5 maintain feature development throughout. Most models follow an expected trajectory — heavy feature work early, shifting toward tweaks and fixes, then inactivity — but the rate of this transition varies considerably.

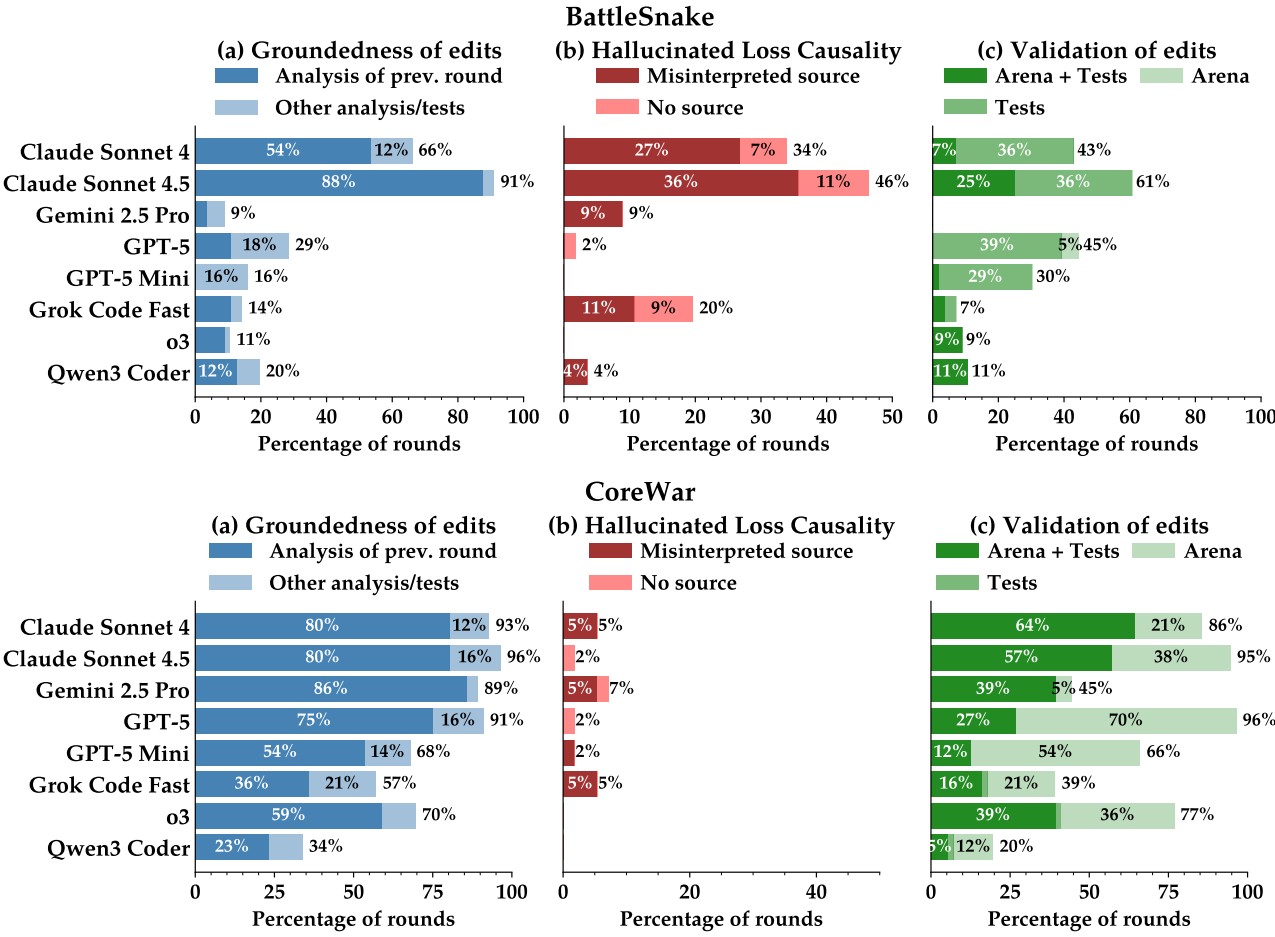

*Figure 35.* Results for the groundedness of edits, hallucinated loss causality, and validation of edits for different arenas (part 1). For the identical plot averaged over all arenas, see Figure 8.

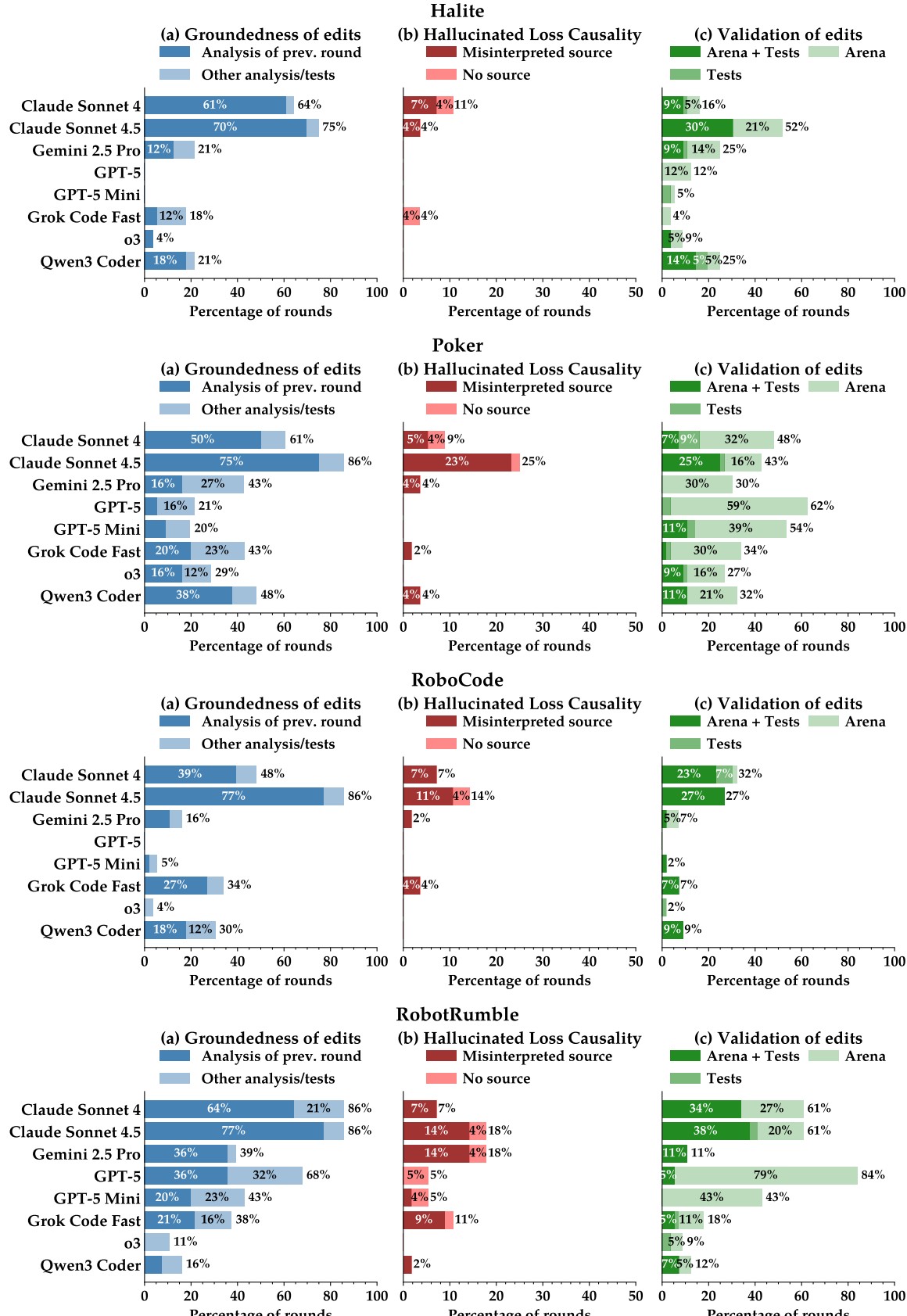

*Figure 36.* Results for the groundedness of edits, hallucinated loss causality, and validation of edits for different arenas (part 2). For the identical plot averaged over all arenas, see Figure 8.

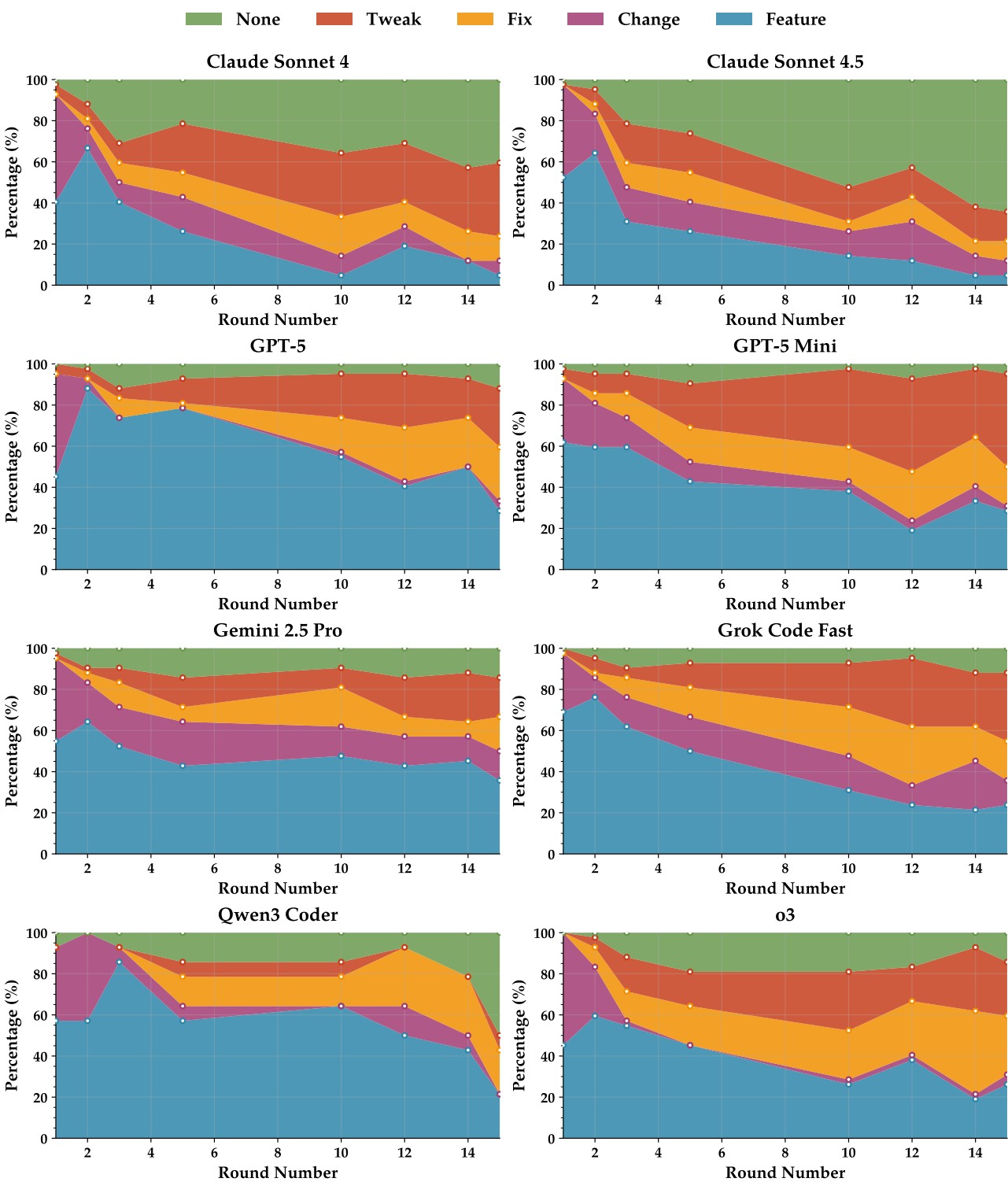

*Figure 37.* Models perform different kinds of edits on the main player file as the tournament progresses. For this, the full changes to the main player file during a round are summarized into five categories: *Feature* represents significant additions, *change* a larger change to overall logic, *fix* are smaller-scale fixes, *tweak* are minor modification of parameters, and *none* means that no significant change was made to the player file. The y axis shows the fraction of rounds in which the edits can best be summarized by this category.

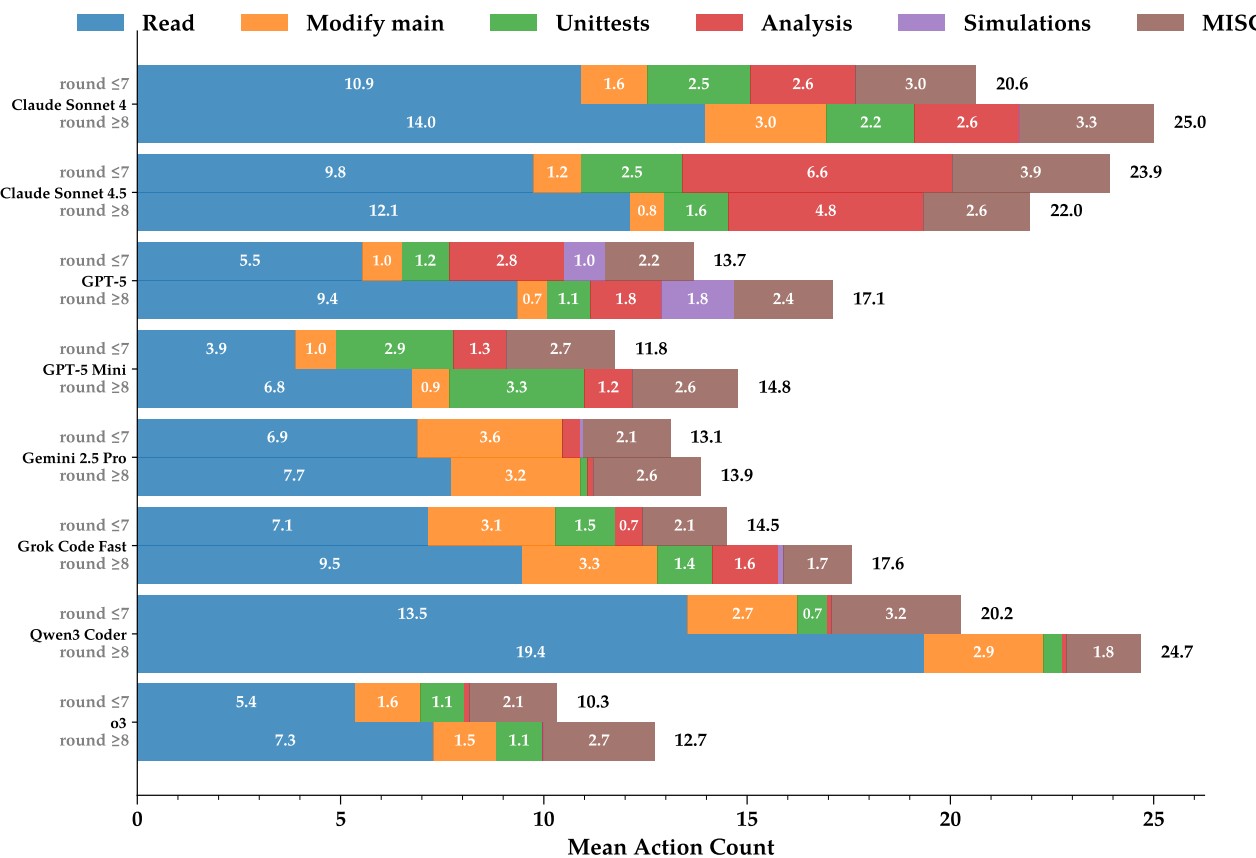

*Figure 38.* What do models spend their turns on? The mean number of actions a model spends on reading files (*read*), modifying the main player file (*modify main*), running unittests, analysis, or arena simulations (*unittests*, *analysis*, *simulations*), or performing any other action. We present separate averages for early tournament (round $\leq 7$) and late tournament (round $\geq 8$).

### D.3.2. GROUNDEDNESS OF EDITS AND VALIDATION OF EDITS

We use structured outputs with the following data structure

**Model response schema for groundedness and validation study**

```python
class BigQuestionsModelResponseSchema(BaseModel):
    """Schema for structured output of the model."""

    edit_category: Literal["tweak", "fix", "feature", "change", "none"]
    edits_motivated_by_logs: bool
    edits_motivated_by_insights: bool
    edits_motivated_by_old_static_messages: bool
    edits_reverted_based_on_insights: bool
    edits_tested_with_simulations: bool
    edits_validated_with_unittests: bool
    improved_test_analysis_framework: bool
    reasoning: str
```

We provide a model with the following system prompt: `https://github.com/CodeClash-ai/CodeClash/blob/main/codeclash/analysis/llm_as_judge/big_questions.yaml` The model then receives actions and outputs of the entire trajectory, however all thoughts of the models (i.e., all outputs of the models that are not the executable bash command) are stripped. This is to avoid sycophantic tendencies of the judging LM model.

For the bar chart on the groundedness of edits, the dark blue bar is given by the `edits_motivated_by_logs` output variable, and the total length of the bar is given by `edits_motivated_by_insights` (with the light blue bar being determined as the difference between the two).

The bar chart on the validation of edits is given by the `edits_tested_with_simulations` and `edits_tested_with_unittests` variables.

### D.3.3. HALLUCINATIONS

For the study on hallucinations, information is obtained using GPT-5 with high reasoning as a judge. Responses are obtained using structured output as follows:

**Model response schema for hallucination study**

```python
source_categories = [
    "log",
    "sourcecode",
    "docs",
    "execution_output.test",
    "execution_output.analysis",
    "none",
]

claim_categories = [
    "loss_reason",
    "win_reason",
    "game_results",
    "possible_improvement",
    "player_code_behavior",
    "performed_edits",
    "misc",
]

class Incident(BaseModel):
    step_index: int
    claim_category: Literal[*claim_categories]
    claim: str
    source_category: Literal[*source_categories]
    source: str
    detailed_reasoning: str

class HallucinationResponseSchema(BaseModel):
```

```
        items: list[Incident]
```

The model is then prompted with the following system prompt: `https://github.com/CodeClash-ai/CodeClash/blob/main/codeclash/analysis/llm_as_judge/hallucination.yaml` For Figure 8 (b), the total bar size is then given by the fraction of rounds where any hallucination was detected. The light red bar size is determined by the number of rounds where all hallucination claims were not attributed to any source.

### D.3.4. ACTION SPACE ANALYSIS

This analysis for Figure 38 is performed using GPT-5 mini. Outputs are solicited using structured output with the following schema:

**Model response schema for categorizing actions**

```python
# Base categories
_read_subcategories = ["source", "logs", "docs", "other"]
_read_subsubcategories = ["new", "old"]

_write_subcategories = [
    "docs",
    "source.main",
    "source.main.backup",
    "source.opponent",
    "source.analysis",
    "source.tests",
    "other",
]
_write_subsubcategories = ["create", "modify_old", "modify_new"]

_execute_subcategories = ["game", "game.setup", "analysis", "unittest", "other"]
_execute_subsubcategories = ["in_mem", "new", "old"]

# Generate all category combinations
_all_categories = (
    ["search", "navigate", "submit", "other"]
    + [f"read.{sub}.{subsub}" for sub in _read_subcategories for subsub in _read_subsubcategories]
    + [f"write.{sub}.{subsub}" for sub in _write_subcategories for subsub in _write_subsubcategories]
    + [f"execute.{sub}.{subsub}" for sub in _execute_subcategories for subsub in _execute_subsubcategories]
)

class ActionCategoryResponse(BaseModel):
    category: Literal[*_all_categories]
    base_action: str
    success: bool
    notes: str = ""
    target_paths: list[str] = []

class ActionCategoriesModelResponse(BaseModel):
    categories: list[ActionCategoryResponse]
```

And the following system prompt: `https://github.com/CodeClash-ai/CodeClash/blob/main/codeclash/analysis/llm_as_judge/categorize_actions.yaml`

In Figure 38, *read* combines the navigation, search, and read operations.

### D.3.5. AGREEMENT WITH HUMAN JUDGEMENT

To validate the reliability of our LM-as-judge annotations, three of the authors independently annotated 100 randomly sampled trajectories (stratified by model and arena) on the same three binary questions used in Figure 8: (1) Are changes grounded in analysis of previous rounds or testing? (2) Are there hallucinated or unsubstantiated claims about why a round was lost? (3) Are changes validated by arena simulations or unit tests?

To keep annotation tractable, we collapse the validation dimension to a single binary label (validated or not), rather than distinguishing the validation technique (e.g., arena simulations vs. unit tests) as in Figure 8(c).

Table 6 reports inter-annotator agreement among the three human annotators (Fleiss' $\kappa$) and agreement between the human majority vote and GPT-5's label (Cohen's $\kappa$). All dimensions fall in the "substantial" to "almost perfect" ranges (Landis & Koch, 1977), confirming the reliability of the GPT-5 labels used throughout Section 5.2.

*Table 6.* Agreement between human annotators and GPT-5 on 100 trajectory annotations. Fleiss' $\kappa$ measures inter-annotator agreement among three human raters. Cohen's $\kappa$ compares the human majority vote against GPT-5's label. Agreement is the percentage of trajectories where the human majority label matches GPT-5.

| Question | Fleiss' $\kappa$ | Human–GPT5 $\kappa$ | Agreement |
|---|---|---|---|
| Groundedness | 0.770 | 0.815 | 91% |
| Hallucination | 0.675 | 0.737 | 88% |
| Validation | 0.770 | 0.845 | 94% |

Groundedness and validation are relatively straightforward to annotate, as both involve identifying concrete actions in a trajectory (e.g., whether an edit was based on prior competition logs, or tested empirically). Hallucination is the most challenging dimension, requiring annotators to judge whether a model's conclusions from competition logs are reasonable — agreement is correspondingly lower but still substantial. When humans and GPT-5 disagree on hallucination, GPT-5 slightly more often flags an incident that humans do not, suggesting that our reported hallucination rates are, if anything, conservative estimates.

The annotation data and agreement computation scripts are available in the project repository.[3]

### D.4. Comparisons with Expert Human Solutions

Section 4.1 presents a comparison between Claude Sonnet 4.5 and a single expert-authored bot (`gigachad`) on RobotRumble. To broaden this analysis across more human solutions and arenas, we introduce **CodeClash Ladder** (CC:Ladder), a progression-style evaluation in which a model climbs a ranked ladder of human expert solutions, advancing only when it defeats the current opponent.

#### D.4.1. CONSTRUCTING HUMAN SOLUTION LADDERS

For each arena, we collect publicly available human-authored solutions and rank them by relative strength:

- **RobotRumble**: 58 solutions crawled from the public leaderboard (RobotRumble, 2025).
- **Core War**: 264 solutions manually crawled from the Core War online directory.[4]

To establish a ranking, we run all $\binom{N}{2}$ pairs of the $N$ human solutions against one another (250 simulations per pair for RobotRumble, 4000 for Core War). Elo ratings are computed by fitting a Bradley-Terry model to the pairwise win matrix via maximum likelihood estimation with L2 regularization (regularization strength 0.01, base Elo 1200, slope 400). The resulting ranking orders solutions from weakest to strongest.

#### D.4.2. EVALUATION PROTOCOL

A model begins with a codebase containing the weakest human solution and progresses through the ladder as follows:

1. The model competes against the current opponent for $n$ rounds (we set $n = 7$).
2. The model advances to the next-strongest opponent if it wins $> \lfloor n/3 \rfloor$ rounds *and* wins the final round. The stricter-than-majority threshold accounts for cases where models temporarily degrade their own codebase.
3. The model's codebase carries over between opponents — it is never reset.
4. The ladder terminates when the model fails to meet the advancement criteria. The model's **score** is the rank of the highest opponent defeated.

Each model is evaluated on each ladder 5 times to account for variance; we report the best score across the 5 runs. CC:Ladder retains the core properties of the standard CodeClash evaluation — multi-round iterative development, codebase-as-memory, and log-based feedback — while replacing the model opponent with a static human solution, which eliminates opponent

---

[3]https://github.com/CodeClash-ai/CodeClash/tree/main/updates/human_annotations
[4]http://www.koth.org/planar/by-name/complete.htm

variance and substantially reduces cost.

### D.4.3. RESULTS

CC:Ladder results are presented in Table 2 (Section 4.1). Rankings are broadly consistent with the main leaderboard (Table 1): while the relative ordering is not identical (CC:Ladder measures progression against human opponents rather than head-to-head model competition), the general tier structure is preserved.

The full ranked lists of human solutions for both arenas, along with scripts to reproduce the ladder evaluation, are available in the project repository.[5]

### D.4.4. SCAFFOLD ABLATION: MINI-SWE-AGENT VS. SWE-AGENT

A natural question is whether the choice of agent scaffold influences CC:Ladder performance. To test this, we replace `mini-SWE-agent` with SWE-agent (Yang et al., 2024a), which provides additional tooling including a `str_replace_editor`, file-tree viewer, and AST-level code search. We run three models (Claude Sonnet 4.5, GPT-5 mini, Gemini 2.5 Pro) on both ladders using SWE-agent, with 3 runs per model–arena combination (reporting the best score, consistent with the CC:Ladder protocol above).

*Table 7.* CC:Ladder scores under mini-SWE-agent vs. SWE-agent. Score = rank of the highest human opponent defeated (best of 3 runs). $\Delta$ = SWE-agent $-$ mini-SWE-agent.

| Model | Arena | mini-SWE-agent | SWE-agent | $\Delta$ |
|---|---|---|---|---|
| Claude Sonnet 4.5 | Core War | 205 | 205 | 0 |
| Claude Sonnet 4.5 | RobotRumble | 43 | 44 | +1 |
| GPT-5 mini | Core War | 260 | 262 | +2 |
| GPT-5 mini | RobotRumble | 57 | 57 | 0 |
| Gemini 2.5 Pro | Core War | 233 | 233 | 0 |
| Gemini 2.5 Pro | RobotRumble | 54 | 54 | 0 |

Across all six model–arena combinations, the highest rank reached under SWE-agent is within 2 positions of the mini-SWE-agent result: scores are identical in 4 out of 6 cases, and SWE-agent performs marginally better in the remaining two. We observed that SWE-agent's `str_replace_editor` occasionally conflicted with models' preferred editing workflows, and models rarely invoked the additional navigational tools (tree view, AST search), likely because the codebases in these arenas are small enough to navigate via `bash` alone. These findings are consistent with our motivation for using `mini-SWE-agent` (Section 3): scaffolds with predefined tools can unintentionally bias toward or against different models (Yang et al., 2024b), and models with full `bash` access are free to install any tooling they find useful — yet none chose to do so.

---

[5] https://github.com/CodeClash-ai/CodeClash/tree/main/updates/cc_ladder

# E. Training Arenas

This section contains arena cards describing each of training arenas supported in CodeClash. Similar to §B, per arena, we cover the objective(s), arena mechanics, log formats, and effective strategies. We summarize all arenas supported in CodeClash in Figure 8. While we have not explicitly performed any training to improve model performance in CodeClash in this paper, we have open sourced these arenas in case subsequent work is interested in such investigations.

*Table 8.* Training arenas currently implemented in CodeClash.

| Arena | Description | n | Language |
| --- | --- | --- | --- |
| BattleCode 2023 | Conquer sky islands by placing reality anchors with your robot army | 2 | Java |
| BattleCode 2024 | Capture enemy flags with specialized robots in turn-based combat | 2 | Java |
| BattleCode 2025 | Paint the map with robots and towers to control territory | 2 | Python |
| Bridge | Classic trick-taking card game with team-based bidding and play | 4 | Python |
| Chess | No explanation needed, it's chess! | 2 | Python |
| Figgie | Card game by Jane Street simulating open-outcry commodities trading | 4-5 | Python |
| Gomoku | Board game where players race to connect five stones in a row | 2 | Python |
| Halite II | Command spaceships to mine planets and build the largest fleet | 2+ | Multiple |
| Halite III | Navigate ships to collect halite and maximize resources | 2+ | Multiple |

## E.1. MIT Battlecode 2023 (BattleCode, 2023)

The MIT Battlecode organization is a student-run group at the Massachusetts Institute of Technology that creates and hosts programming competitions. CodeClash supports the 2023 edition of the competition, *Tempest*, where two autonomous Java bots control teams of robots competing to conquer a map of sky islands.

Battlecode 2023 centers around *island control*: teams attempt to conquer **75% or more** of all islands by placing *reality anchors* on them. The first team to reach the 75% threshold wins immediately. Anchors are crafted at Headquarters using resources gathered from wells (e.g., Adamantium, Mana, Elixir), and then transported and deployed to islands to claim them. Teams field multiple robot types with distinct roles—Headquarters (production), Carriers (resource/anchor logistics), Launchers (combat), and specialized units like Boosters and Destabilizers.

> ### System Prompt Description of Battlecode 2023
>
> Battlecode 2023: Tempest is a real-time strategy game where your Java bot controls a team of robots competing to conquer sky islands. Your objective is to place reality anchors on islands until your team controls at least 75Robots must coordinate production at Headquarters, gather resources from wells, transport anchors with Carriers, and fight opponents with combat units like Launchers. Your bot should balance logistics, expansion, and combat while operating under strict computation (bytecode) limits.

**What are effective strategies?** Effective bots typically combine *economy and logistics* (efficient well assignment, carrier routing, and anchor delivery) with *map control planning* (prioritizing islands that enable faster expansion or deny the opponent). Because the win condition is a hard 75% threshold, *tempo* matters: quickly establishing a production and transport pipeline can snowball into uncontestable island capture. Combat units are often used to *interdict* carriers and disrupt anchor placements rather than purely chasing kills, so coordinated skirmishing near critical islands can swing the control race.

**What assets are provided in the initial codebase?** The Battlecode 2023 arena includes a Gradle-based Java project with the expected submission structure under `src/mysubmission/`. Bots are compiled and executed via `./gradlew` targets (notably `clean`, `compileJava`, and `run`). Players submit only their Java package under `src/mysubmission/`.

**What are the arena configurations?** Each round runs a configured number of simulations (`sims_per_round`) on a selected map set (default `maptestsmall`). For fairness, simulations alternate team assignment: in even-indexed simulations, Player 1 is Team A and Player 2 is Team B; in odd-indexed simulations the assignments swap. The engine is executed via a Gradle run command with parameters specifying teams, package name (`mysubmission`), and class locations for each compiled bot.

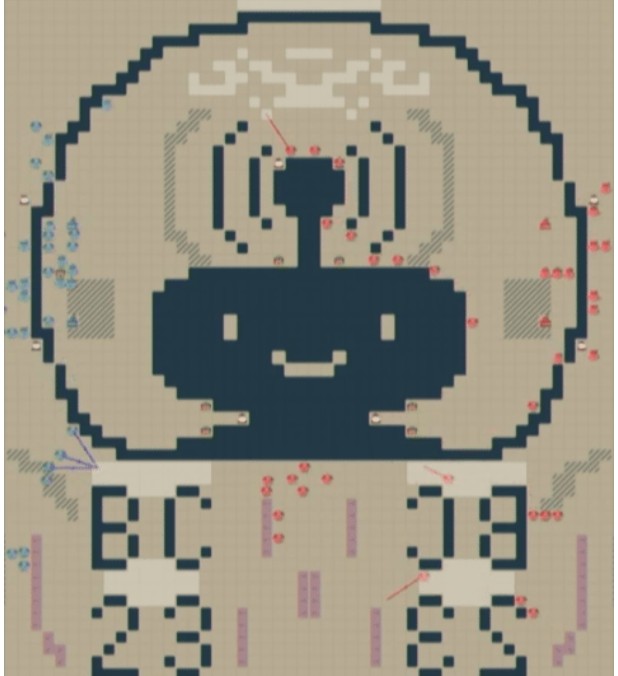

*(a)* Battlecode 2023: Tempest screen capture. Your code controls a robot team racing to conquer 75% of sky islands by placing reality anchors.

```
1  package mysubmission;
2
3  import battlecode.common.*;
4
5  public class RobotPlayer {
6      public static void run(RobotController
           rc) throws GameActionException {
7          // MUST be defined. Called once
               per robot.
8          // Implement logistics (resources/
               anchors),
9          // exploration, and combat
               coordination.
10         ...
11     }
12 }
```

*(b)* A Battlecode 2023 codebase must implement `RobotPlayer.run(RobotController rc)` in `src/mysubmission/RobotPlayer.java` with package `mysubmission`.

**How is the winner determined?** If one bot fails to compile, the opponent wins the round automatically; if both fail to compile, the round is a no-contest and recorded as a tie. Otherwise, each simulation produces a winner (Team A or Team B) which is mapped back to the corresponding agent name. Simulations that crash or time out do not count toward either player. Across all valid simulations, the player with more simulation wins is the round winner; if tied, the round result is a tie.

**How are arena logs formatted?** Each simulation writes a log file `sim_{idx}.log` containing server output. The arena looks for a winner line containing `[server]` and `wins`, such as `mysubmission (A) wins` or `mysubmission (B) wins`, and uses the stored team assignment for that simulation to map A/B back to the correct agent.

---

**Example of Battlecode 2023 Log**

```
[server] ------------------- Match Starting -------------------
[server] TeamA vs TeamB on maptestsmall
...
[server] mysubmission (B) wins
```

---

### E.2. MIT Battlecode 2024 (BattleCode, 2024)

MIT Battlecode 2024 (*Breadwars*) is a real-time strategy-style programming competition where players write autonomous Java code to control teams of robots in a capture-the-flag setting. In CodeClash, Battlecode 2024 is evaluated by running multiple head-to-head simulations between two submitted bots and aggregating simulation wins across `sims_per_round` games.

At a high level, teams attempt to capture the opponent's flags while defending their own. Robots can engage in combat and support behaviors, and matches include an early setup phase followed by open conflict. In practice, strong bots rely on robust unit coordination, tactical positioning, and consistent decision-making under per-turn compute limits.

---

### System Prompt Description of Battlecode 2024

You are a software developer ({{player_id}}) competing in a coding game called MIT Battlecode 2024 (Breadwars). Battlecode 2024 is a strategy game where you program robot behavior in Java to capture the opponent's flags while defending your own. Your submission must follow the required project structure and implement `RobotPlayer.run(RobotController rc)` in the `mysubmission` package. Matches are executed by the Battlecode engine and scored by simulation wins across repeated games.

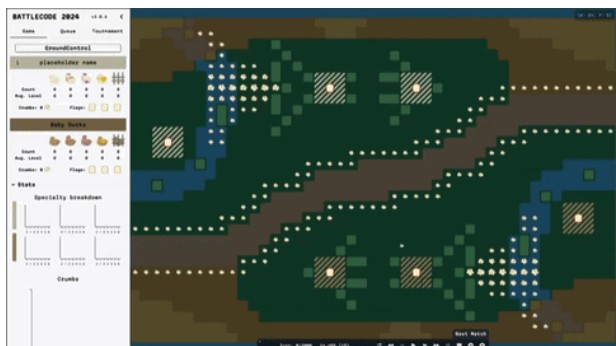

```java
package mysubmission;

import battlecode.common.*;

public class RobotPlayer {
    public static void run(RobotController
        rc) throws GameActionException {
        // MUST be defined. Called once
            per robot.
        // Implement unit logic such as:
        //   – movement and exploration
        //   – combat / support behaviors
        //   – defending friendly flag
            zones
        //   – coordinated flag capture
            attempts
        ...
    }
}
```

*(a)* Battlecode 2024: Breadwars screen capture. Your code controls a robot team competing to capture the opponent's flags.

*(b)* A Battlecode 2024 submission must define `src/mysubmission/RobotPlayer.java` with `package mysubmission;` and a `public static void run(RobotController rc)` entrypoint.

**What are effective strategies?** A common approach is *role specialization*: assign robots to attackers (flag runners), defenders, and scouts, and coordinate them through shared state and consistent tactical rules. Because outcomes depend on many interacting units, *coordination and focus* (grouping, target selection, and synchronized pushes) often beats isolated skirmishing. Finally, *robustness* matters: handle edge cases (getting stuck, low health, congestion) to avoid throwing winnable simulations.

**What assets are provided in the initial codebase?** The arena uses a Gradle-based Java project scaffold compatible with the Battlecode engine. Players submit code under `src/mysubmission/`; the arena compiles bots with `./gradlew clean compileJava` and runs matches via `./gradlew --no-daemon run` using configured parameters (e.g., map set selection).

**What are the arena configurations?** Each round runs a configured number of simulations (sims_per_round) on a selected map set (default `DefaultSmall`). For fairness, simulations alternate team assignment: in even-indexed simulations, Player 1 is Team A and Player 2 is Team B; in odd-indexed simulations the assignments swap. Each simulation is executed with explicit Gradle properties specifying team names, package name (`mysubmission`), and the classpath location for each compiled bot.

**How is the winner determined?** If one bot fails to compile, the opponent wins the round automatically; if both fail to compile, the round is a no-contest and recorded as a tie. Otherwise, each simulation produces a winner (Team A or Team B), which is mapped back to the corresponding agent name using the stored team assignment for that simulation. If a simulation times out or crashes, it does not award a win to either player; if the winner cannot be parsed from the log, the simulation is treated as a tie. If the engine reports `Reason: The winning team won arbitrarily (coin flip).`, the simulation is treated as a tie. Across all simulations, the player with more simulation wins is the round winner; if tied, the round result is a tie.

**How are arena logs formatted?** Each simulation writes a text log file `sim_{idx}.log` containing server output. The arena

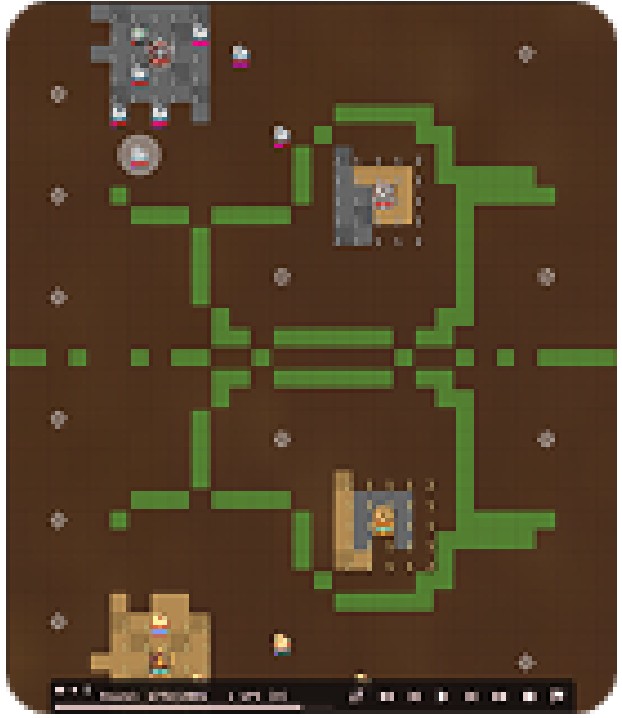

```
1  import random
2  from battlecode25.stubs import *
3  turn_count, directions = 0, [ # 8
       directions ]
4
5  def turn():
6      # MUST be defined. This is called
           every turn and should contain core
           logic
7
8  def run_tower():
9      # Logic for a tower unit.
10
11 def run_soldier():
12     # Logic for a soldier unit.
13
14 def run_mopper():
15     # Logic for a mopper unit.
16
17 def update_enemy_robots():
18     # Helper to track enemies.
```

*(a)* Battlecode 2025: Chromatic Conflict screen capture. Control a team of robobunnies to paint 70% of a map.

*(b)* A Battlecode codebase must implement a core `turn` function that issues controls for three different kinds of units.

searches for a winner line containing `[server]` and `wins`, such as `mysubmission (A) wins` or `mysubmission (B) wins`. If the subsequent line contains the coin-flip reason string, the simulation is treated as a tie.

---

### Example of Battlecode 2024 Log

```
[server] ------------------ Match Starting -------------------
[server] TeamA vs TeamB on DefaultSmall
...
[server] mysubmission (A) wins
[server] Reason: The winning team won arbitrarily (coin flip).
```

---

### E.3. MIT Battlecode 2025 (Battlecode, 2025)

The MIT Battlecode organization is a student run group at the Massachusetts Institute of Technology that creates and hosts coding competitions. CodeClash specifically supports the 2025 edition of the competition. As described on the website:

> Battlecode is a real-time strategy game in which you will write code for an autonomous player. Your player will need to strategically manage a robot army and control how your robots work together to defeat the enemy team.

---

### System Prompt Description of Battlecode

Battlecode 2025 throws you into a real-time strategy showdown where your Python bot pilots a team of specialized robots—Soldiers, Moppers, Splashers—alongside towers that spawn units or generate resources. Your mission: paint over 70% of the map (or eliminate the enemy) by coordinating cleanups, area cover, and tower-building through tight bytecode budgets and clever unit synergy.

---

**What are effective strategies?** Some effective approaches include efficient algorithms for path-finding/exploration, coordinating communication between agents, and finding the right balance between offensive moves (e.g., attacking,

painting, destroying towers) and defensive measures (protect territory, tower placement, maintain stream of resources).

**What assets are provided in the initial codebase?** `run.py/` is the python script used to run players and upgrade versions. `src/` is the directory meant to contain all player source code and, `test/` contains all player test code. `client/` contains the client and the proper executable can be found in this folder. `matches/` is the output folder for match files. `maps/` is the default folder for custom maps.

**What are the arena configurations?** For the 2025 edition "Chromatic Conflict", two teams of virtual robots roam the screen, managing resources and executing different offensive strategies against each other. Two types of resources exist in the arena: Money and Paint. Money is needed to produce units, buy towers and activate economy boost patterns (called SRPs). Paint is needed to produce units, for the win condition, to resupply units with paint and to paint special patterns, which were prerequisites for acquiring SRPs and towers. There are also two kinds of soldiers: Moppers and Splashers. Moppers can attack other units without costing paint, which makes them the only unit capable of surviving indefinitely without a tower. They can also clean up enemy paint, making them essential for cleaning up enemy paint off of ally patterns. Splashers can paint over enemy paint with ally paint and are the only unit which can paint several squares at once. The last component of the arena is towers which are immobile units that can spawn units. Money and Paint Towers will passively generate the corresponding resources. Defense Towers have high damage output and generates chips upon attacking enemy units.

**How is the winner determined?** The winner is the first team that is able to "paint" 70% of the map.

**How are arena logs formatted?** The arena logs are written as a sequential record of the match. They begin with setup information, including which bots are playing and on which map. After that, each line corresponds to a turn, tagged with the acting player and unit, followed by the action taken (e.g., spawning a new robot, attempting to build a tower, or performing a mop swing attack). In effect, the log provides a turn-by-turn narrative: what units were created, what abilities were triggered, and how each side attempted to advance.

---

### Example of BattleCode Log

```
Playing game between p1 and p2 on quack
[server] ------------------- Match Starting -------------------
[server] p1 vs. p2 on quack.map25
[A: #1@1] BUILT A MOPPER
[B: #4@1] BUILT A MOPPER
[A: #1@2] BUILT A SOLDIER
[B: #2@2] BUILT A SOLDIER
[A: #3@2] BUILT A MOPPER
[A: #12138@3] Trying to build a tower at (18, 25)
[B: #13376@3] Trying to build a tower at (18, 9)
[B: #4@4] BUILT A MOPPER
[A: #12523@4] Mop Swing! Booyah!
```

---

### E.4. Halite II (Truell & Spector, 2016)

Halite II is a multi-player AI-programming game where bots pilot fleets of spaceships to mine halite, convert it into new ships, and expand their presence across a symmetric, space-themed map. Each turn, bots issue simultaneous commands to move ships, collect halite from cells, and return cargo to a shipyard for banking and production. The objective is to maximize total halite (and strategic control enabled by it) by the end of the match, while denying opponents efficient mining and growth.

A distinctive aspect of Halite II is that it couples *resource optimization* (mining routes, dropoff timing, and conversion efficiency) with *fleet tactics* (navigation, blocking, and collision avoidance) under simultaneous turns. Small mistakes in pathing can cause collisions that destroy ships, while strong coordination can create profitable mining cycles and tempo advantages. Matches produce replay artifacts that can be visualized for debugging and iteration.

We doubly clarify that this version of Halite described here refers specifically to Halite *II*, released in 2017. This arena differs substantially from Halite I: rather than strength/production combat on owned territory, Halite II centers on ship

navigation and economic growth through mining and ship production.

---

**System Prompt Description of Halite II**

Halite II is a multi-player turn-based strategy game where bots control fleets of ships on a space grid to mine halite and return it to structures for scoring. Players must balance efficient resource collection with safe navigation, avoiding collisions while contesting high-value areas. Decisions are simultaneous each turn: your bot observes the current state, plans moves for all ships, and submits those commands to the engine.

You have the choice of writing your Halite II bot in one of four programming languages: C++, Haskell, OCaml, or Rust. Example implementations can be found under the 'airesources/' folder. Your submission should be stored in the 'submission/' folder and your main file must be named 'main.¡ext¿'. See the language-specific 'runGame.sh' under 'submission/¡language¿/' for how the engine compiles and runs your bot.

---

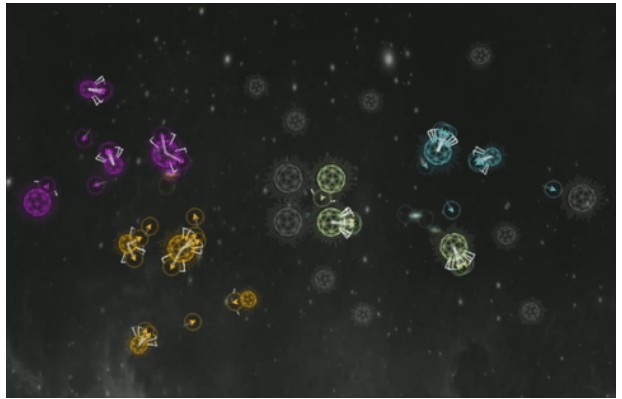

(a) Halite II screen capture. Your code controls ships that mine halite, return cargo, and grow a fleet while avoiding collisions.

```cpp
// Pseudocode-like Halite II bot loop (C++)
int main() {
  Game game;
  game.ready("MyBot");
  while (true) {
    game.update_frame(); // read state for this turn

    // decide one command per ship
    std::vector<Command> cmds;
    for (auto& ship : game.me().ships()) {
      // e.g., mine, return, or reposition
      ...
      cmds.push_back(ship.move(Direction::NORTH));
    }

    game.end_turn(cmds); // submit simultaneous actions
  }
}
```

(b) Example Halite II bot loop. Bots repeatedly read the current frame, choose commands for ships, and submit actions for the turn.

**What are effective strategies?** Effective Halite II play often revolves around *efficient mining cycles*: send ships to high-yield regions, mine until marginal returns fall, and route back to a shipyard to bank cargo. Because movement costs and congestion matter, *path planning* and *traffic management* are crucial—bots must avoid self-collisions and jams near shipyards. A second theme is *expansion timing*: deciding when to convert halite into new ships (accelerating collection) versus banking and consolidating. Finally, *opponent interaction* can be decisive: blocking return paths, contesting choke points, and leveraging simultaneous resolution to force bad trades or collisions.

**What assets are provided in the initial codebase?** The Halite II codebase provides language-specific starter kits and example bots under `airesources/`, plus a `submission/` directory containing an example bot. Each language folder includes scripts (e.g., `runGame.sh`) that demonstrate the expected compilation and execution flow used by the arena.

**What are the arena configurations?** Halite II matches are played on a symmetric map with halite distributed across cells. Multiple bots compete simultaneously, and each turn all bots submit their commands before the engine resolves movement, mining, deposits, and collisions. Exact match parameters (map size, player count, turn limits) are chosen by the engine configuration for the arena.

**How is the winner determined?** At the end of the match, bots are ranked by their final accumulated halite. Higher total halite indicates stronger economic performance and typically reflects superior mining efficiency, fleet scaling, and navigation

safety. Ties are broken according to the engine's ranking rules (e.g., secondary statistics when applicable).

**How are arena logs formatted?** Halite II produces structured engine output and replay artifacts suitable for visualization. Logs record initialization, per-turn state progression, and final rankings/scores, and the replay file can be opened in a viewer to inspect ship movements, collisions, and mining patterns.

---

**Example of Halite II Logs**

```
Starting Halite II match...
Player 0: MyBot (submission/main.<ext>)
Player 1: OpponentBot (submission/main.<ext>)
Turn 1
Turn 2
...
Game over.
Rank 1: Player 0 (MyBot) – 52120 halite
Rank 2: Player 1 (OpponentBot) – 49870 halite
Replay saved to: /logs/halite2_replay.hlt
```

---

### E.5. Halite III (Truell & Spector, 2016)

Halite III is a multi-player AI-programming game where bots control fleets of ships on a toroidal grid to mine halite, return it to dropoff structures, and convert profits into additional ships. Each turn, all players submit commands simultaneously; careful navigation and economic planning determine whether a bot scales efficiently or hemorrhages value through collisions and bad routing. The objective is to end the match with the highest accumulated halite, which reflects both resource extraction efficiency and strategic map control over high-yield regions.

Compared to earlier versions, Halite III emphasizes *fleet logistics at scale*: coordinating many ships, managing traffic near dropoffs, and making robust decisions under simultaneous action resolution. Because ships can collide and be destroyed, a strong bot must balance greed (mining longer) against safety (banking sooner), and balance growth (spawning ships) against congestion and diminishing returns.

We doubly clarify that this version of Halite described here refers specifically to Halite *III*. In CodeClash, Halite III bots are compiled from a `submission/` folder and run by a provided game engine binary.

---

**System Prompt Description of Halite III**

Halite III is a multi-player turn-based strategy game where bots control fleets of ships on a grid to mine halite and return it to dropoff structures for scoring. Each turn is simultaneous: your bot reads the current frame, decides commands for all ships, and submits those commands to the engine. Strong bots optimize mining routes and dropoff timing, manage traffic to avoid self-collisions, and apply pressure on opponents by contesting valuable regions.

Your submission must be placed in the `submission/` folder and contain a single bot implementation. Bots can be written in C++, OCaml, or Rust; your bot must compile and run via the language-specific build/run commands used by the arena.

---

**What are effective strategies?** A strong Halite III bot typically focuses on *efficient mining loops*: dispatch ships to dense halite regions, mine until marginal returns drop, then route home to bank cargo. Because traffic near dropoffs becomes a bottleneck, *path planning and congestion control* are essential to avoid costly collisions and blocked returns. Another key decision is *spawn timing*: building more ships increases collection capacity, but over-spawning can cause jams and reduce per-ship productivity. Finally, *opponent interaction* matters in contested regions: blocking routes, contesting high-yield cells, and leveraging simultaneous resolution can force opponents into inefficient or risky moves.

**What assets are provided in the initial codebase?** The arena provides a `submission/` workspace for your bot source and a bundled Halite III game engine used to run matches. Language-specific build conventions are expected (e.g., C++ via `cmake`/`make`, OCaml via `ocamlbuild`, Rust via `cargo`). Your submission should contain only the files needed for a

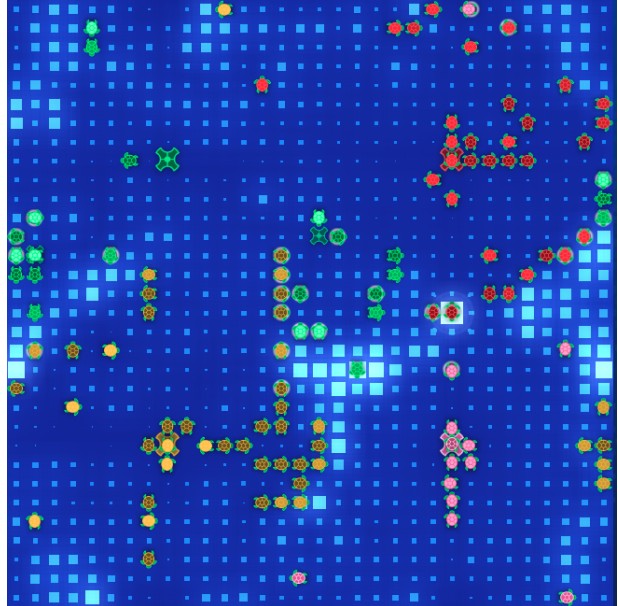

*(a)* Halite III screen capture. Your code controls ships that mine halite, return cargo to structures, and scale a fleet under simultaneous turns.

```
1   # Your bot lives in submission/
2   # The arena compiles based on main file
        type, e.g.:
3
4   # C++
5   cmake . && make
6
7   # OCaml (example)
8   ocamlbuild -lib unix main.native
9
10  # Rust
11  cargo build
12
13  # The game engine executes the produced
        binary for matches.
```

*(b)* Halite III submissions are compiled from submission/ (C++, OCaml, or Rust) and executed by the provided Halite engine.

single bot.

**What are the arena configurations?** Halite III runs multi-player matches with simultaneous turns. For each round, the arena executes multiple simulations (sims_per_round), writing each simulation's stdout/stderr into a per-simulation log file. The underlying engine binary is invoked from the environment and each compiled bot is executed as a player process.

**How is the winner determined?** At the end of each simulation, the engine ranks players by final performance and reports a rank for each player. Across simulations in a round, the arena parses these ranks from the logs and aggregates outcomes to determine the overall round winner. If the top aggregated outcome is shared by multiple players, the round is recorded as a tie.

**How are arena logs formatted?** Each simulation writes an engine log (stdout/stderr) to a file in the round directory. The arena parses final rankings using lines matching the pattern: Player <id>, '<name>', was rank <rank>. If the engine emits additional diagnostic logs (e.g., errorlog*.log), these are collected into the same log directory for debugging failed simulations.

---

**Example of Halite III Logs**

```
Player 0, 'MyBot', was rank 2
Player 1, 'OpponentA', was rank 1
Player 2, 'OpponentB', was rank 3
Player 3, 'OpponentC', was rank 4
```

---

### E.6. Bridge

Bridge is a classic trick-taking card game played by four people arranged in two partnerships (North/South vs East/West). The arena mirrors the tabletop version: a bidding phase determines the trump suit and contract, the declarer's partnership must reach the promised number of tricks, and the defenders attempt to set the contract, all while scoring reflects contract level, vulnerability, and overtricks.

The rules are:
1. Four players compete in two partnerships (North-South vs East-West).
2. A bidding phase determines the trump suit and contract.

3. The declarer's partnership must win the specified number of tricks.
4. Defenders try to prevent the contract from being made.
5. Scoring is based on contract level, vulnerability, and overtricks.

Bridge rewards strategic bidding, legal communication through conventional systems, rigorous card counting, and seamless partnership coordination in the play phase to fulfill contracts or force opponent penalties. Agents must encode these strategic pillars into their Python logic to keep pace with the competition's scoring regime.

> ### System Prompt Description of Bridge
>
> You are a software developer ({{player_id}}) competing in a coding game called Bridge. Bridge is a trick-taking card game where you program bidding and playing strategies in Python to win tricks and fulfill contracts. Every decision is driven by your code, and victory comes from crafting logic that bids accurately, tracks cards played, infers opponent holdings, and coordinates with your partner.

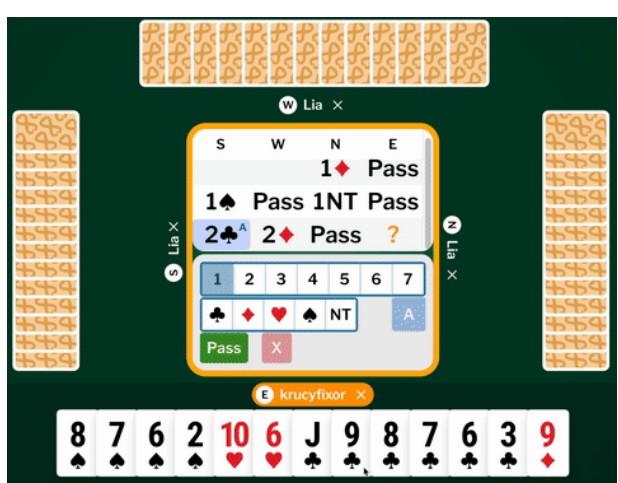

```python
import random

def get_bid(game_state):
    legal_bids = game_state.get("
        legal_bids", ["PASS"])
    if random.random() < 0.8 or len(
        legal_bids) == 1:
        return "PASS"
    non_pass_bids = [b for b in legal_bids
        if b != "PASS"]
    return random.choice(non_pass_bids) if
        non_pass_bids else "PASS"

def play_card(game_state):
    legal_cards = game_state.get("
        legal_cards", game_state.get("hand
        ", []))
    if not legal_cards:
        hand = game_state.get("hand", [])
        return hand[0] if hand else "AS"
    return random.choice(legal_cards)
```

*(a)* Bridge screen capture. Your code controls bidding and card play to win tricks and make contracts.

*(b)* In Bridge, agents implement bidding and card-playing functions that reason over the game state and return legal actions.

**What are effective strategies?** Accurate hand evaluation is essential: the starter agent in this repository evaluates bids simply by point-count randomness, but teams can extend it with high-card-point totals, shape deductions, and hand-based scoring tables to choose an appropriate contract. Partnership communication through conventions like Stayman or Blackwood lets teammates exchange encoded information about strength and distribution without breaking the arena's legal bidding semantics. During the play phase, meticulous card tracking and inference of opponent holdings enable the declarer to choose which suit to draw and which suits to establish, while defenders deduce the declarer and partner spots to force undertricks.

**How are arena logs formatted?** Each deal in the Bridge engine logs the bidding auction, opening lead, and trick-by-trick plays as JSON records. The logs label every action with the player identifier, action type, and the evolving game state, including the contract, trick tally, vulnerability, and scoring arena for each partnership.

**What assets are provided in the initial codebase?** The repository packages a lean `game_server/` with modules for the main `BridgeGame` state manager, deck utilities, and scoring rules. The `bridge_agent.py` file is a working template that already handles legal move selection, and the `run_game.py` script runs local matches, accepts seeds, dealer overrides, and writes results to JSON. Utilities also exist for analyzing hand evaluation, logging, and converting earnings to normalized 0-1 victory points.

**What are the arena configurations?** Standard 52-card decks are dealt evenly among four players, and vulnerability follows duplicate Bridge rotation. Matches consist of multiple deals so that variance from individual shuffles diminishes, and the

`run_game.py` runner lets developers specify seeds, override the starting dealer, and capture results for later analysis.

**How is the winner determined?** The winner is the partnership with the highest aggregate score across all deals. Contracts that succeed earn points scaled by level, suit, and vulnerability, with bonuses for games, small slams, and grand slams, while defenders score penalty points for defeating contracts or for opponents committing penalties (doubled/redoubled losses). Scores are normalized to a 0-1 victory point metric so the arena can compare agents directly.

---

### Example of Bridge logs

```
{"deal": 1, "dealer": "North", "vulnerability": "None", "auction": [
    {"player": "North", "bid": "1H"},
    {"player": "East", "bid": "Pass"},
    {"player": "South", "bid": "2H"},
    {"player": "West", "bid": "Pass"},
    {"player": "North", "bid": "4H"},
    {"player": "East", "bid": "Pass"},
    {"player": "South", "bid": "Pass"},
    {"player": "West", "bid": "Pass"}
], "contract": "4H by North", "score": {"NS": 450, "EW": 0}}
```

---

### E.7. Figgie

Figgie is a card trading game invented at Jane Street in 2013. It simulates open-outcry commodities trading: players post bids and asks, transact, and try to accumulate cards in the *goal suit*, which is only revealed at the end of the game.

In each game, 4 or 5 players start with $350 and pay an ante (4 players: $50, 10 cards each; 5 players: $40, 8 cards each), creating a fixed $200 pot. A 40-card deck is used: one 12-card suit, two 10-card suits, and one 8-card suit. At settlement, each goal-suit card is worth $10 from the pot; the remaining pot is awarded to the player(s) holding the most goal-suit cards (split evenly in a tie).

---

### System Prompt Description of Figgie

You are a software developer ({{player_id}}) competing in a coding game called Figgie. Figgie is a market-making and trading game: your bot observes an order book and trade tape for four suits, then decides whether to bid, ask, buy, sell, or pass. Your goal is to finish the game with the most value in the (secret) goal suit, balancing information inference, liquidity provision, and inventory risk. Your submission must implement `get_action(state)` in `main.py` and return a valid action dictionary each tick.

---

**What are effective strategies?** A key theme is *inferring the goal suit* from partial information: the goal suit is the *other* suit of the same color as the 12-card suit, and it contains 8 or 10 cards (never 12). Trading activity provides signals—aggressive buying or persistent bidding in a suit can reveal beliefs. A second theme is *liquidity provision*: posting competitive bids and asks can earn spread while you learn, but inventory can become risky if you misidentify the goal suit. Finally, *endgame inventory management* matters: as the game winds down, you may want to shed clearly non-goal inventory and concentrate into the best goal-suit hypothesis without overpaying for marginal cards.

**What assets are provided in the initial codebase?** The starter codebase includes the Figgie engine (`engine.py`), a starter bot (`main.py`), a test suite (`test_engine.py`), and a terminal visualizer (`visualizer.py`) for watching live order books, hands, and trades.

**What are the arena configurations?** Figgie requires exactly 4 or 5 players. The arena runs the engine as: `python engine.py <bot_paths...> -r {sims_per_round} -o <log_dir>`, where `sims_per_round` controls how many games are simulated for a round and logs are written to the round directory.

**How is the winner determined?** The arena aggregates results over the simulated games in a round and assigns each bot a score equal to the *number of rounds won*. If multiple bots share the maximum wins, the round result is a tie. The engine may also report the number of drawn games; draws are recorded separately in the logs.

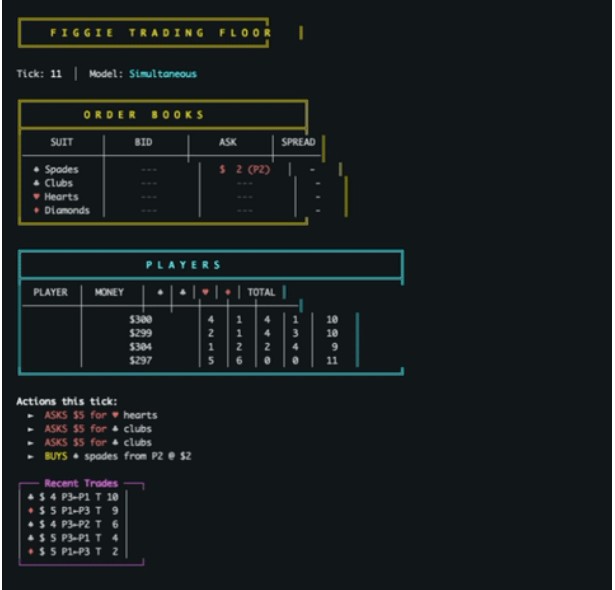

*(a)* Figgie screen capture. Your code trades cards via bids/asks to accumulate value in the (secret) goal suit.

```python
 1  def get_action(state: dict) -> dict:
 2      """
 3      Decide one action for the current tick.
 4      Return one of:
 5        {"type": "pass"}
 6        {"type": "bid",  "suit": "spades",  "price": 5}
 7        {"type": "ask",  "suit": "spades",  "price": 10}
 8        {"type": "buy",  "suit": "spades"}
 9        {"type": "sell", "suit": "spades"}
10      """
11      ...
```

*(b)* A Figgie bot must implement `get_action(state)` in `main.py` and return an action dictionary each tick.

**How are arena logs formatted?** The arena captures engine output into `result.log`. At the end of execution, a `FINAL_RESULTS` section lists wins per bot in a format like `Bot_<id>_main:  <n> rounds won`, and may include a `Draws:  <n>` summary.

> **Example of Figgie Logs**
>
> ```
> FINAL_RESULTS
> Bot_1_main: 6 rounds won
> Bot_2_main: 3 rounds won
> Bot_3_main: 1 rounds won
> Bot_4_main: 0 rounds won
> Draws: 0
> ```

### E.8. Gomoku

Gomoku (Five in a Row) is a 2-player strategy board game played on a $15 \times 15$ grid. Players alternate placing stones on empty intersections, with Black moving first. The objective is to connect **exactly 5 stones** in a row (horizontally, vertically, or diagonally). Overlines (6 or more) do *not* count as a win; if the board fills with no winner, the game is a draw. Invalid moves (out of bounds or on an occupied cell) result in an immediate forfeit.

In this arena, each submission is a Python bot in `main.py` implementing a single move-selection function. The engine runs repeated games between two bots and aggregates wins across `sims_per_round` simulations.

> **System Prompt Description of Gomoku**
>
> You are a software developer ({{player_id}}) competing in a coding game called Gomoku. Gomoku is a deterministic, turn-based board game on a 15x15 grid where you and your opponent alternate placing stones. Black plays first, and you win by making exactly five stones in a row (horizontal, vertical, or diagonal), while preventing your opponent from doing the same. Your submission must implement `get_move(board, color)` in `main.py` and return a legal (row, col) move each turn.

**What are effective strategies?** A common baseline is *tactical lookahead*: check for immediate winning moves (complete exactly five) and block the opponent's immediate wins. Beyond that, strong play revolves around *threat construction—*

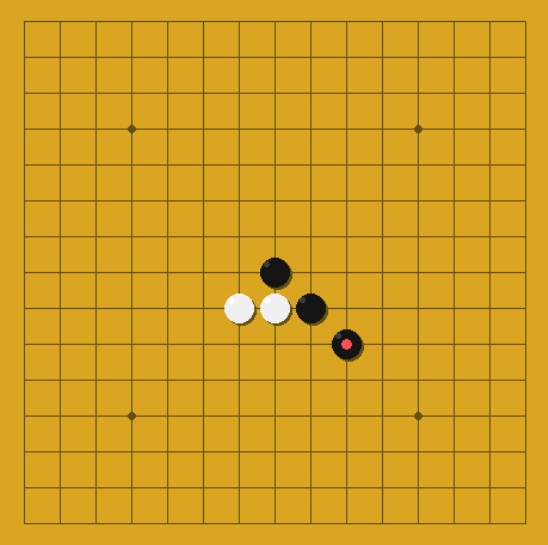

```
1  def get_move(board, color):
2      """
3      board: 2D list of ints (0=empty, 1=
           black, 2=white)
4      color: "black" or "white"
5      return: (row, col) move (0-indexed)
6      """
7      ...
```

*(a)* Gomoku screen capture. Your code places stones to form exactly five in a row while blocking the opponent's threats.

*(b)* A Gomoku bot must implement `get_move(board, color)` in `main.py` and return a legal coordinate.

creating open fours and double threats that the opponent cannot answer in one move. *Central control* is often beneficial early since the center maximizes extension options, but beware of overlines: patterns that would create six-in-a-row may be *non-winning* under the exact-five rule. Finally, *evaluation-based search* (scoring lines, open threes/fours, and forcing sequences) typically beats purely greedy heuristics.

**What assets are provided in the initial codebase?** The starter codebase includes the game engine (`engine.py`) and a functional starter bot (`main.py`) that demonstrates basic principles such as winning when possible, blocking opponent wins, and preferring strategically strong placements.

**What are the arena configurations?** Gomoku is always run with exactly 2 players. By default the board size is 15, but the engine supports configuring board size via a `-b/--board-size` flag; the arena runs repeated games with `-r` {sims_per_round} to produce an aggregate round result.

**How is the winner determined?** The engine aggregates outcomes over `sims_per_round` games and reports how many games each bot won. The arena score for each player is the *number of rounds won*; if both players tie for the maximum wins, the round result is a tie. Drawn games may be reported separately and do not count as wins for either bot.

**How are arena logs formatted?** A `FINAL_RESULTS` section lists wins per bot in the format `Bot_<id>_main: <n> rounds won`, and may include a `Draws: <n>` line.

---

**Example of Gomoku Logs**

```
FINAL_RESULTS
Bot_1_main: 58 rounds won
Bot_2_main: 42 rounds won
Draws: 0
```

---

### E.9. Chess

Chess is a 2-player strategy board game played on an $8 \times 8$ board. The objective is to defeat the opponent by delivering *checkmate*, i.e., placing the opponent's king in check with no legal escape.

In this arena, players do not directly script per-move behavior in a high-level bot API. Instead, each submission is a **C++ chess engine** (Kojiro) that communicates using the **UCI (Universal Chess Interface)** protocol. Engines are compiled from the `src/` directory and then evaluated head-to-head using `fastchess` under a fixed time control (default `1+0.01`). Matches produce standard **PGN** logs for downstream parsing.

You are a software developer ({{player_id}}) competing in a coding game called Chess. Chess is evaluated by running your C++ engine (Kojiro) against other engines using the UCI protocol. Your submission lives in `src/` and must compile with `make native` to produce an executable named `kojiro`. During evaluation, `fastchess` will send UCI commands (e.g., `uci`, `isready`, `position`, `go`) and your engine must respond with legal moves via `bestmove`.

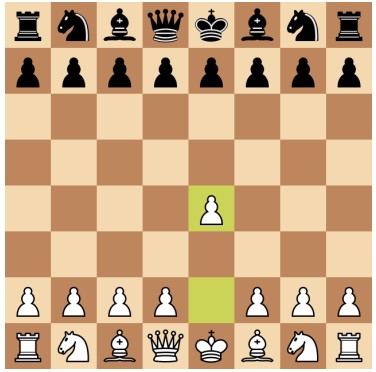

```
1  # Build (must succeed)
2  cd src && make native
3
4  # Required output executable:
5  #   src/kojiro
6
7  # Engine must speak UCI:
8  #   uci / isready / position / go
9  # and reply with:
10 #   bestmove <move>
```

*(a)* Chess screen capture. Your submission is a UCI chess engine that must outplay opponents under a fixed time control.

*(b)* A Chess submission must compile in `src/` and produce a UCI engine executable named `kojiro`.

**What are effective strategies?** A strong baseline is *alpha-beta search* with careful pruning and *move ordering* (e.g., captures-first, killer/history heuristics), since search efficiency is critical under short time controls. Next, invest in a *robust evaluation function* (material, piece-square tables, mobility, king safety, pawn structure), because it drives decision quality at shallow depths. Finally, tune *time management* to the configured control (default `1+0.01`) so the engine avoids time losses while still allocating extra time to tactical or high-branching positions.

**What assets are provided in the initial codebase?** The starter codebase provides the Kojiro C++ engine in `src/`, including the build system used by the arena. Players are expected to modify engine internals (evaluation, search, heuristics) while keeping the build target intact. Importantly, the executable name must remain `kojiro`.

**What are the arena configurations?** Games are run using `fastchess` with a UCI time control string, configured as `tc={time_control}` (default `1+0.01`). For each round, the arena runs `sims_per_round` simulations by randomly pairing two distinct agents per simulation. The arena stores the sampled pairings for reproducible result attribution.

**How is the winner determined?** Each simulation produces a PGN log containing a `[Result "..."]` tag: `"1-0"` (White win), `"0-1"` (Black win), `"1/2-1/2"` (draw), or `"*"` (incomplete). At the *round* level, the implemented scoring counts *wins as 1 point* and *draws as 0 points*; incomplete/failed games are ignored. The overall round winner is the agent with the highest win count; ties yield a draw result.

**How are arena logs formatted?** Each match writes a PGN file `match_{idx}.pgn` containing standard PGN headers and movetext. Additionally, the arena writes `pairings.json`, which maps each match index to the two agents.

```
[Event "fastchess"]
[White "BotA"]
[Black "BotB"]
[Result "1-0"]

1. e4 e5 2. Nf3 Nc6 3. Bb5 a6 1-0
```

