*(b)* RobotRumble leaderboard screen capture as of October 31, 2025. We evaluate Claude Sonnet 4.5 against the top open-source submission `gigachad` by `entropicdrifter`

ratio. We provide a randomly selected example of a codebase produced by Claude Sonnet 4.5 at the end of a 15 round tournament of BattleSnake, playing against Gemini 2.5 Pro, in Figure 34. The tournament ID is `PvpTournament.BattleSnake.r15.s1000.p2.claude-sonnet-4-5-20250929.gemini-2.5-pro.251002020143`.

As discussed in the main results, we notice that codebases tend to follow this trend of creating single use analysis and testing files that are then rarely reused later on in a tournament. While we do not explore mitigating such behavior with prompting, we purport that this result is still noteworthy. Refactoring and sustaining a well organized codebase is not something that models organically aspire towards. CodeClash can serve as a testbed for investigating how LM managed codebases morph over time and exploring whether interventions in the form of data or external rewards can encourage better practices.

Finally, with Figure 32b, we find that the number of redundantly named files climbs upwards at different rates across all models. Figure 34 gives us a concrete example. Claude Sonnet 4.5 creates 13 files with the prefix "`analyze_`". From manual inspection, we found that most of these implementations are doing the same thing, with only the log file path being different. The same trend holds for the "`check_`" and "`ROUND_`" files. Such redundancy points to obvious room for improvement. Long running SWE-agent's that iterate and reuse a core set of files rather than spamming the codebase with single use scripts should be the more desirable behavior in the vast majority of use cases.

**Claude Sonnet 4.5 loses to a static solution written by a human expert.** As discussed in Section 4.1, we run 10 tournaments of Claude Sonnet 4.5, the top model on the RobotRumble arena, against the top open-source submission we found on RobotRumble's online leaderboard (`gigachad` by `entropicdrifter`).

Beyond the setup discussed in the main paper, we point out several additional details:

- The top open source submission we use is ranked fourth overall (1554 Elo) on the leaderboard. Three additional, closed source submissions rank above, as shown in Figure 33b, with the top submission ranking nearly 700 Elo points higher.
- While our main RobotRumble results ask models to write their bots in JavaScript, since the human submission is implemented in Python, for fairness, we ask Claude Sonnet 4.5 to implement its bot in Python as well.

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

```