# OpenReview forum: "CodeClash: Benchmarking Goal-Oriented Software Engineering"
_ICML.cc/2026/Conference — ICML 2026 regular_

### Official Review · Reviewer_NxFK · 2026-02-22

**Soundness:** 4
**Presentation:** 4
**Significance:** 3
**Originality:** 3
**Overall Recommendation:** 5
**Confidence:** 4

**Summary:**

This paper introduces a novel benchmark designed to evaluate whether LM agents can iteratively develop code to achieve open-ended objectives without explicit guidance. In this benchmark, LM agents continuously edit their code, which then acts as a proxy to compete head-to-head across six distinct game environments.

After testing eight frontier LMs, the authors found that these models exhibit fundamental limitations in both strategic reasoning and long-term codebase maintenance, often resulting in increasingly messy and redundant repositories. Furthermore, the study highlights that top-performing LMs losing every single round against human expert.

**Compliance With Llm Reviewing Policy:**

Affirmed.

**Final Justification:**

I thank the authors for their rebuttal and will maintain my positive score.

**Key Questions For Authors:**

Aside from RobotRumble, other arenas lack quantitative comparisons with human performance. Could CodeClash measure the gap between human experts and LM agents by introducing a large number of human-crafted solutions in these arenas? Does the evaluation cost increase linearly as more candidates are added?

**Limitations:**

Yes.

**Strengths And Weaknesses:**

**Strengths**
- **Novel Task Formulation**: The benchmark shifts the code generation focus from the static unit-test passing to open-ended objective fulfillment. It contains diverse and well-defined game environments, which has plenty of human solutions serving as the strong baseline to evaluate the performance of LM agents.
- **Unique Memory & Feedback Loop**: By implementing the “Codebase-as-memory” and “Log-based feedback” constraints, CodeClash effectively isolates the model's intrinsic engineering capabilities from the interference from agent frameworks.
- **Extensive and Sound Evaluation**: The scale of the experiments is impressive, involving 1,680 tournaments of 8 LMs across 6 diverse environments, providing a statistically rigorous model ranking.
- **Extensibility**: The CodeClash framework acts as a general scaffold where new environments can be integrated with minimal overhead, showing the potiential to be an extendable testbed for future work.

**Weaknesses**
- **Coarse-Grained Attribution:** While the benchmark provides a clear overall ranking for model candidates, the multi-faceted nature of the task makes it difficult to decouple specific agent failures. It remains hard to attribute the final model performance to certain abilities, such as strategic reasoning, code generation, or parse massive logs.
- **Reliance on Proprietary Models:** The evaluation is heavily weighted toward closed-source frontier models. Including a broader range of open-source models beyond Qwen3 Coder, would enhance the benchmark's utility for the wider research community.

---

> ### Author Rebuttal · Authors · 2026-03-31
>
> # Response to Reviewer NxFK
>
> Our thanks to Reviewer `NxFK` for the detailed and supportive review, and in particular for highlighting CodeClash's extensibility as a general scaffold where new environments can be integrated with minimal overhead.
>
> Several concerns were shared across reviewers. To reduce redundancy and detail new experiments thoroughly, we provide three [General Responses](https://github.com/emagedoc/CodeClash/tree/main/updates) that address these common points. Where relevant, we reference them below.
>
> ---
>
> **> W1: Coarse-Grained Attribution — the multi-faceted nature of the task makes it difficult to decouple specific agent failures (strategic reasoning vs. code generation vs. log parsing).**
>
> We firmly concur with the reviewer's observation, as it also highlights why CodeClash's premise of shifting from task-oriented to goal-oriented evaluations is well-founded.
>
> Benchmarks like SWE-bench or Terminal-bench often result in models following a fairly "fixed" problem solving pattern (e.g., For SWE-bench: 1. reproduce the bug, 2. find the buggy file/code, 3. apply fix, 4. run the reproduction script to validate the fix).
> On the other hand, as the reviewer aptly points out, the open-endedness of CodeClash arenas exposes much richer model behaviors, and in turn, a wealth of potential failure modes.
>
> **Section 5.2**'s decomposed analysis is our best effort at disentangling such factors.
> By annotating a trajectory along multiple dimensions (groundedness, hallucination, validation), we aimed to identify *what* models do wrong or poorly.
> Such an approach is inspired by tools such as [Docent](https://transluce.org/introducing-docent) and [StringSight](https://blog.stringsight.com/).
> That said, we recognize the limitations of this strategy; for instance, our annotations are not capable of attributing specific, concrete actions or thoughts directly to a win/loss.
>
> To this end, we argue that effective analyses are a promising direction for future work.
> The agent and analysis strategies we use are heavily representative of the tooling used for existing, popular LM agent tasks like SWE-bench and Terminal-bench.
> The desire for better understanding of model behaviors in the CodeClash settings motivates better toolkits for understanding model capabilities for carrying out goals across longer horizons.
>
> ---
>
> **> W2: Reliance on Proprietary Models — the evaluation is heavily weighted toward closed-source frontier models.**
>
> We agree. However, while the work was in progress, we empirically found that Qwen 3 Coder was the only model that showed any meaningful foothold in the task.
> Sub 70B models and other OSS ones lost nearly every round against the 8 models.
> That said, it's quite likely the landscape is different now.
>
> Within the amount of time and budget available for this rebuttal period, we spent more of our effort on addressing more immediate concerns (comparison to human experts, validating GPT 5 labels with human annotations).
>
> We aim to add more models to the leaderboard ourselves in the coming months, but have also invested immense effort in reproducibility and ease of use.
> CodeClash is entirely open source, and any interested members of the community should be able to run an OSS model of their choice with ease.
>
> ---
>
> **> Q1: Could CodeClash measure the gap between human experts and LM agents across more arenas? Does evaluation cost increase linearly?**
>
> Thanks for pointing this out along with fellow reviewers.
> This is a meaningful shortcoming we hope we've addressed adequately with [General Response #1](https://github.com/emagedoc/CodeClash/blob/main/updates/general_response_1.md).
>
> * We extended the human vs. AI competition format introduced in Section 4.1 ablations and built [CodeClash Ladders](https://github.com/emagedoc/CodeClash/blob/main/updates/cc_ladder/report.md) exactly for this purpose: models climb a ranked ladder of human expert solutions, advancing only when they defeat the current opponent.
> * We construct ladders for RobotRumble (58 human solutions) and Core War (264 human solutions), ranked by Elo from all-pairs matchups.
> * The strongest performer, Claude Sonnet 4.6, only reaches rank 39/58 on RobotRumble and 177/264 on Core War, confirming a substantial human-AI gap.
> * Cost scales linearly with the number of model candidates (not quadratically as in head-to-head tournaments), making it practical to extend to additional arenas.

---

> > ### Author Rebuttal · Reviewer_NxFK · 2026-04-03
> >
> > I thank the authors for their rebuttal and will maintain my positive score.

---

> > > ### Author Response · Authors · 2026-04-05
> > >
> > > We sincerely appreciate you taking time to review our work, thank you again!

---

### Official Review · Reviewer_kanR · 2026-03-04

**Soundness:** 2
**Presentation:** 3
**Significance:** 2
**Originality:** 3
**Overall Recommendation:** 3
**Confidence:** 3

**Summary:**

This paper introduces a novel benchmark called CodeClash, designed to evaluate LLMs in goal-oriented software engineering tasks. Unlike traditional benchmarks that focus on fixing single bugs, CodeClash has multiple LLM agents compete against each other in a multi-round tournament framework. The paper evaluated 8 foundation models across 1680 tournaments. It points out that while models exhibit some diversity in code development strategies, they have severe systematic flaws in long-term strategic reasoning, log analysis, and codebase maintenance.

**Compliance With Llm Reviewing Policy:**

Affirmed.

**Final Justification:**

While the authors have made commendable efforts during the rebuttal to validate their metrics, the foundational setup of the benchmark, specifically the restrictive tooling and the artificial memory constraints, continues to limit the real-world applicability of the findings. This is a paper with clear merits, but also some weaknesses, which overall outweigh the merits. Because the paper requires revisions to its core evaluation framework before it can be meaningfully built upon by others, I maintain my original overall recommendation.

**Key Questions For Authors:**

1. What is a calculated agreement score between the GPT-5 judge's results and human expert annotations when evaluating Section 5.2? Without a human-validated baseline, how can we trust the absolute values of the hallucination rates for each model presented in Figure 8?
2. How to prove that the models' failures stem from reasoning bottlenecks rather than the implementation of the mini-SWE-agent?
3. How to justify the top models lose every round against expert human programmers?

**Limitations:**

The authors do not acknowledge the severe limitations of their evaluation setup. To improve this section, they must explicitly discuss the inherent biases of using an LLM as a judge to evaluate the strategic reasoning and hallucination rates of other models without any human validation baseline.

**Strengths And Weaknesses:**

Strengths:
1. Shifting LLM evaluation from static, single-round code fixing (like SWE-bench) to a dynamic, multi-round competitive environment
2. Massive Experimental Scale

Weaknesses:
1. A core conclusion of the paper is that "top models lose every round against expert human programmers". However, this conclusion is solely based on a limited comparison in a single arena: RobotRumble. The authors also admit in the text that other arenas (Core War, RoboCode, Halite, BattleSnake, Poker) either do not have leaderboards or lack readily available open-source, ranked submissions.
2. When analyzing the limitations of models' strategic reasoning and log analysis capabilities in Section 5.2 (i.e., whether models hallucinate or perform adequate validation), the authors entirely rely on GPT-5 as a judge for automated annotation and evaluation. Using an LLM to evaluate other LLMs carries inherent biases well-recognized in the academic community. The paper provides no human-annotation overlap metrics to demonstrate the objectivity and accuracy of the GPT-5 judge in evaluating such complex long-term planning tasks.
3. The paper intentionally excludes popular frameworks with comprehensive toolchains, such as SWE-agent or OpenHands, and instead chooses a highly restricted, minimalist interface, mini-SWE-agent. This restriction limits models to purely bash command interactions. Consequently, the observed phenomena of models performing poorly in long-term code maintenance and generating redundant files might be due to the deprivation of standard linting, AST parsing, or code tree visualization tools found in advanced agent frameworks, rather than a direct failure of the foundation models' reasoning capabilities.
4. The design of CodeClash mandates that agents have no explicit memory of actions from previous rounds. The agents' only source of information is whatever they choose to record in their own codebases. This "codebase-as-memory" setup is severely disconnected from the reality of software development, where developers have version control systems, issue tracker histories, and continuous contextual memory. This introduces an unreasonable artificial hurdle in the evaluation, leading to an incorrect underestimation of model capabilities.

---

> ### Author Rebuttal · Authors · 2026-03-31
>
> # Response to Reviewer kanR
>
> We appreciate Reviewer `kanR` for recognizing CodeClash's aim to shift from task- to goal-oriented evaluations.
> The feedback encouraged us to strengthen the paper's empirical foundations to the best of our abilities.
>
> To reduce redundancy and detail new experiments thoroughly, we provide three [General Responses](https://github.com/emagedoc/CodeClash/tree/main/updates) shared concerns expressed by multiple reviewers. Where relevant, we reference them below.
>
> ---
>
> **> W1 + Q3: "Top models lose every round" is based on one arena. How to justify this claim?**
>
> We understand the concern.
> [General Response #1](https://github.com/emagedoc/CodeClash/blob/main/updates/general_response_1.md) addresses this directly:
>
> * We extended human vs AI studies into [CodeClash Ladders](https://github.com/emagedoc/CodeClash/blob/main/updates/cc_ladder/report.md), where models climb a ranked ladder of human expert solutions (58 for RobotRumble, 264 for Core War), advancing only when they defeat the current opponent.
> * The strongest performer, Claude Sonnet 4.6, only reaches rank 39/58 on RobotRumble and 177/264 on Core War, confirming a substantial gap between models and human experts.
>
> On the second part of the critique: the sentence in the paper does not mean open source submissions are non-existent. Crawling, cleaning, and putting them in CodeClash requires substantial human effort. With Core War and RobotRumble, we constructed leaderboards from online sources. For Halite, BattleSnake, and RoboCode, human solutions exist online but unifying them into a leaderboard is time consuming and an ongoing effort.
>
> ---
>
> **> W2 + Q1 + Limitations: Reliance on GPT-5 as judge with no human-annotation overlap. How can we trust hallucination rates?**
>
> We agree that LM-as-judge analysis should be validated with human annotations.
> [General Response #2](https://github.com/emagedoc/CodeClash/blob/main/updates/general_response_2.md) presents the human annotation study the reviewer asked for: three authors independently labeled 100 trajectories across all three dimensions (groundedness, hallucination, validation).
> Inter-annotator agreement (Fleiss' kappa 0.67–0.77) and human-vs-GPT5 agreement (Cohen's kappa 0.74–0.85) both fall in the "substantial" to "almost perfect" ranges.
>
> On hallucination rates specifically: this dimension has the lowest agreement of the three (as expected). However, when humans and GPT-5 do disagree, it is predominantly GPT-5 *over*-flagging hallucinations, meaning our reported rates are conservative estimates.
>
> ---
>
> **> W3 + Q2: mini-SWE-agent's bash-only interface may cause failures rather than reasoning limitations. How to disentangle?**
>
> [General Response #3](https://github.com/emagedoc/CodeClash/blob/main/updates/general_response_3.md) discusses this in depth. The key points:
>
> First, models are proficient with the bash interface. Section 5.2 (Figure 7) reports 85%+ bash command success rates with rapid error recovery. The dominant failure modes in Figure 8 are *strategic*: a linter would not help a model that misinterprets why it lost a round; AST parsing would not help a model that deploys changes without testing.
>
> Second, scaffold complexity does not reliably proxy agent capability:
>
> * On the SWE-bench leaderboard ([swebench.com](https://swebench.com)), models with mini-SWE-agent sometimes outperform those using heavier frameworks. SWE-agent has been officially deprecated in favor of mini-SWE-agent by its developers, citing similar-if-not-better performance.
> * Scaffolds that prescribe specific tools may accidentally bias for/against certain models ([Yang et al., 2024](https://arxiv.org/abs/2410.03859)). Work like Live-SWE-agent ([Xia et al., 2025](https://arxiv.org/abs/2511.13646)) demonstrates how mini-SWE-agent's simplicity enhances, *not* restricts, a model's ability to operate.
> * The SWE-agent creators recommend mini-SWE-agent instead of SWE-agent (see [swe-agent.com](swe-agent.com))
>
> Finally, mini-SWE-agent's minimalist design is intentional (Section 3.1): it isolates the foundation model's capabilities from scaffold-specific advantages. A richer toolchain would confound attribution.
>
> ---
>
> **> W4: The "codebase-as-memory" setup is disconnected from reality.**
>
> We'd argue the opposite: codebases *are* memory in software development.
> READMEs, inline comments, commit messages, and agent-facing docs like `CLAUDE.md` all exist to encode knowledge so any new contributor can onboard from the codebase alone.
> The tools the reviewer mentions are either available (`git` is installed) or can be constructed by the model itself (e.g., inline notes, `README_agent.md`).
>
> This design choice also produces some of the paper's most novel findings: Section 5.1 reveals that models spontaneously develop memory strategies such as writing analysis scripts, leaving notes to themselves, and creating throwaway files. These emergent behaviors could not be studied if we provided an external memory system.

---

> > ### Author Rebuttal · Reviewer_kanR · 2026-04-03
> >
> > Thank you for the thorough rebuttal and the new human-annotation experiments, which nicely address my concerns regarding the LM-as-judge bias. However, regarding the restricted mini-SWE-agent interface (W3), while I understand your point that advanced tools wouldn't solve strategic failures (like misinterpreting losses), it remains unclear whether the specific issues of long-term codebase clutter and redundant file generation could have been mitigated if the models had access to standard code-tree visualization or AST parsing tools.

---

> > > ### Author Response · Authors · 2026-04-07
> > >
> > > Thanks so much reviewer `kanR`, glad the response helped with some questions.
> > >
> > > Regarding your follow up point: We agree and think you raise a valid conceptual point.
> > >
> > > * In large codebases with deep directory hierarchies, lots of interdependent modules, and high code reuse, such tools may help with navigation and localization.
> > > * We noticed a model accumulates throwaway files and duplicated code over many rounds; a file-tree viewer or AST-level code inspection might be beneficial.
> > >
> > > To test this directly, we swap out mini-SWE-agent for SWE-agent and run 3 models (Claude Sonnet 4.5, GPT-5 mini, Gemini 2.5 Pro) on the **CodeClash Ladders** (*CC:Ladder*) evaluation introduced in the rebuttal to benchmark models against multiple human solutions (**[full report](https://github.com/emagedoc/CodeClash/blob/main/updates/general_response_1.md)**).
> > > As a reminder, CC:Ladder is a progression-style evaluation where models start against the weakest ranked human solution and must win a majority of rounds (best-of-n) to advance to the next opponent.
> > >
> > > We use CC:Ladder instead of the default setting because it provides a head-to-head scaffold comparison without the opponent model being a confounding factor.
> > > Cost was also a consideration:
> > >
> > > * The original CodeClash setting requires re-running all pairwise tournaments with SWE-agent (~$25k total, same as original experiments)
> > >    * We could reduce some factors (fewer models/arenas/tournaments per pair). But each reduction either limits generalizability across the model roster or game types, or amplifies noise — CodeClash tournaments already have inherent variance from model-vs-model dynamics, so stable estimates require sufficient repetitions.
> > > * With CC:Ladder, using static human solutions eliminate variance from opponents. 3 models × 2 arenas × 3 runs = 18 ladder runs, costing ~$250 total.
> > >
> > > **Results**
> > > | Model | Arena | mini-SWE-agent (rank) | SWE-agent (rank) | Δ |
> > > |---|---|---|---|---|
> > > | Claude Sonnet 4.5 | Core War | 205 / 264 | 205 / 264 | 0 |
> > > | Claude Sonnet 4.5 | RobotRumble | 43 / 58 | 44 / 58 | -1 |
> > > | GPT-5 mini | Core War | 260 / 264 | 262 / 264 | -2 |
> > > | GPT-5 mini | RobotRumble | 57 / 58 | 57 / 58 | 0 |
> > > | Gemini 2.5 Pro | Core War | 233 / 264 | 233 / 264 | 0 |
> > > | Gemini 2.5 Pro | RobotRumble | 54 / 58 | 54 / 58 | 0 |
> > >
> > > "rank" = highest rank reached across 3 runs, consistent with our reporting for the original CC:Ladder report.
> > >
> > > Across all six model–arena combinations (spanning three providers), the highest rank reached under SWE-agent is within 2 positions of the mini-SWE-agent result.
> > >
> > > * In 4/6 cases, the rank is identical
> > > * In the other 2, SWE-agent performs *slightly worse*. We observed that SWE-agent's `str_replace_editor` occasionally conflicts with the model's preferred editing workflow, and models rarely invoke the navigational tools (tree view, AST search) given the small codebase size.
> > >    * This is consistent with our original motivation for using mini-SWE-agent (Section 4): scaffolds with predefined tools can unintentionally bias toward or against different models ([Yang et al., 2024](https://arxiv.org/abs/2410.03859)).
> > >    * Worth noting: mini-SWE-agent provides full bash access, so models are free to install and use AST parsers, tree viewers, or linters at any point. None of the models chose to do so.
> > >
> > > *tl;dr* We understand and agree, to an extent, with reviewer `kanR`'s sound feedback.
> > > However, our experiments show that at least for two arenas, switching to the SWE-agent scaffold doesn't change things too much.
> > >
> > > CodeClash codebases are small - this could limit the impact of otherwise useful tools like a linter or directory/file viewer.
> > > Beyond this work, we agree the impact of tools on model performance is certainly an interesting topic and could serve as strong motivation for extending CodeClash to more complex arenas that require models to manage larger codebases.

---

### Official Review · Reviewer_y7Xj · 2026-03-13

**Soundness:** 3
**Presentation:** 4
**Significance:** 3
**Originality:** 4
**Overall Recommendation:** 5
**Confidence:** 4

**Summary:**

In this study, the authors evaluate 8 different language models across 6 competitive arenas. This is done through 1,680 tournaments each of which consist of 15 rounds. In every round the models autonomously modify their codebases in order to improve their chances of winning the future rounds. The models are not allowed to access information from the previous rounds but they are allowed to store information from the previous rounds in the repository. The model performance is then compared using Elo based rankings.

**Compliance With Llm Reviewing Policy:**

Affirmed.

**Key Questions For Authors:**

N/A

**Limitations:**

yes

**Strengths And Weaknesses:**

Strengths:
+ Novel and well-motivated problem formulation addressing a key gap in current coding benchmarks
+ distinctive and thoughtfully designed benchmark
+ diverse arenas that improve the robustness of evaluation
+ large-scale and convincing experimental evaluation
+ Insightful analysis beyond simple leaderboard comparisons
+ high potential for community impact if released.

Weaknesses:
- The comparison with human-written solutions is limited to a single arena (RobotRumble)
- The paper does not explore how results change under varying budgets and prompts.

The paper presents a well-motivated problem formulation that targets an important limitation of existing coding benchmarks. While most prior evaluations focus on short-term, well-specified tasks such as bug fixing or function implementation, the authors argue that real-world software development is often driven by higher-level goals and requires iterative improvements over time. CodeClash evaluates whether models can iteratively evolve a codebase across multiple rounds to achieve competitive objectives. The benchmark design is distinctive in that it combines iterative code editing, competition-based feedback, and persistent codebases, encouraging models to reason about longer-term strategies rather than producing one-shot solutions.

Another strength of the paper is the diversity and scale of the evaluation. The benchmark includes six arenas with different objectives and interaction dynamics, which reduces the risk that results are overly dependent on a single task environment. In addition, the experimental study is relatively large-scale, evaluating eight frontier models across 1,680 tournaments. This setup allows for more robust comparisons between models and provides a clearer view of their behavior in goal-driven development settings, increasing the credibility and potential usefulness of the benchmark for studying autonomous software engineering systems.




The paper introduces a new way of evaluating a model’s behaviour by including how models adapt across rounds and manage their codebases. This makes the benchmark more informative. The release of CodeClash as an open-source benchmark increases its potential impact on the research community. A publicly available benchmark of this kind could serve as a valuable testbed for studying autonomous software engineering agents


A limitation of the paper is that the human comparison is restricted only to a single arena, RobotRumble, which makes it difficult to judge how broadly the human vs model gap generalizes. In addition, the paper does not study how results change under different prompt settings or editing budgets, so the robustness of the reported rankings across configurations is not clear.

---

> ### Author Rebuttal · Authors · 2026-03-31
>
> # Response to Reviewer y7Xj
>
> Thank you Reviewer `y7Xj` for the thoughtful and encouraging review, and especially for recognizing CodeClash's potential for community impact as an open-source benchmark for studying autonomous software engineering agents.
>
> Several concerns were shared across reviewers. To reduce redundancy and detail new experiments thoroughly, we provide three [General Responses](https://github.com/emagedoc/CodeClash/tree/main/updates) that address these common points. Where relevant, we reference them below.
>
> ---
>
> **> W1: The comparison with human-written solutions is limited to a single arena (RobotRumble).**
>
> [General Response #1](https://github.com/emagedoc/CodeClash/blob/main/updates/general_response_1.md) addresses this concern directly.
>
> * Following our initial human vs AI match up to include more human expert solutions, we formalize such comparisons at scale with [CodeClash Ladders](https://github.com/emagedoc/CodeClash/blob/main/updates/cc_ladder/report.md), a new evaluation format where models climb a ranked ladder of human expert solutions, advancing only when they defeat the current opponent.
> * We construct ladders for RobotRumble (58 human solutions) and Core War (264 human solutions), ranked by Elo from all-pairs matchups.
> * The strongest performer, Claude Sonnet 4.6, only reaches rank 39 out of 58 on RobotRumble and rank 177 out of 264 on Core War, indicating that models still have a substantial gap to close against human experts.
>
> ---
>
> **> W2: The paper does not explore how results change under varying budgets and prompts.**
>
> [General Response #3](https://github.com/emagedoc/CodeClash/blob/main/updates/general_response_3.md) addresses this concern.
>
> * We provide evidence from multiple paper sections that the observed performance gaps stem from strategic reasoning limitations, not scaffold or budget choices.
> * Models achieve 85%+ bash command success rates (Section 5.2, Figure 7), and the dominant failure modes (hallucinated loss causality, untested deployments) would not be mitigated by richer tooling.
> * On budget sensitivity, model rankings remain consistent between the main evaluation (15 rounds) and CodeClash Ladders (7 rounds per opponent), and the multi-agent study in Section 4.2 (6-player Core War with TrueSkill) also produces consistent rankings despite very different competitive dynamics.

---

### Official Review · Reviewer_5t8L · 2026-03-13

**Soundness:** 3
**Presentation:** 3
**Significance:** 3
**Originality:** 4
**Overall Recommendation:** 4
**Confidence:** 4

**Summary:**

The CodeClash is a benchmark that aids goal-directed software engineering by allowing LM agents to iteratively edit persistent code bases and compete in multi-round programming tournaments based on their edited code bases. The CodeClash consists of 6 arenas with heterogeneous objectives and languages, and 8 vastly different models evaluated as benchmarks in those arenas. This paper explores not only how to rank each model but also how the models adapt over multiple rounds of the tournament, code diversity from each round of the tournament, proof of codebase maintenance, codebase recovery from failure to produce winning code, and whether the edit was based on a log or received pre-deployment validation. The CodeClash study reveals that: Claude Sonnet 4.5 is ranked number 1 overall, no model outperforms every arena, models do not recover well from failing to produce the best edited code, all codebases become increasingly cluttered over time, and many of the edits were either weakly based on logs or weakly validated before deployment.

**Compliance With Llm Reviewing Policy:**

Affirmed.

**Ethical Review Concerns:**

No paper-specific ethics flag from the current draft

**Final Justification:**

Based on initial comments and rebuttal.

**Key Questions For Authors:**

1. How stable are rankings under different edit budgets or interaction scaffolds?
2. Do you plan to add larger, less game-like repositories or stronger human baselines?

**Limitations:**

The paper discusses technical limitations, including limitations of the arenas being smaller/self-contained than full systems, and there being only text as feedback. There is little discussion of the overall impact on society.

**Strengths And Weaknesses:**

Strengths
- CodeClash provides a variety of ways to evaluate long-horizon and open-ended improvements to software via competition in established codebases; this provides more than just static correctness or fixing issues as benchmarks.
- The setup allows for evaluating adaptation, strategic planning, and ongoing maintenance of repositories, as opposed to just a one-shot generation of code.
- An extensive amount of evaluation has been performed, and the appendix contains information to clarify many important implementation details, including arenas, agent configurations, method limitations on interaction, tournament design, and rank estimation.
- The appendix also clarifies the details regarding setup for the bash-only mini-SWE agents, editing a maximum of $1 per step for 30 steps, applying the approach to fitting, and measuring bootstrap-based rank stability.
- The paper examines various aspects related to recovery from failures, diversity of solutions, clutter of repositories, and validating/grounding edits, providing scientific data to add a level of scientific validity to the benchmark.

Weaknesses
- While the benchmark provides an LM judge for groundedness, validation, and hallucination analysis, there is insufficient evidence provided for the reliability of those assessments, such as human validation, agreement analysis, and robustness across judges.
- The benchmarks are still relatively small and self-contained arenas and log-driven compared to real-world software engineering environments; therefore, the benchmark is still somewhat stylized.
- The appendix clearly specifies the fixed interactions, but there is little evidence from the paper that the results will be stable across varying budgets or prompting, or different action interfaces.
- The RobotRumble comparison is good, but is limited because the appendix makes the comparison fairer than the paper, and only provides information on one arena and only one expert bot created by a static expert.

---

> ### Author Rebuttal · Authors · 2026-03-31
>
> # Response to Reviewer 5t8L
>
> We thank Reviewer `5t8L` for the thorough review, and especially for appreciating that CodeClash evaluates adaptation, strategic planning, and ongoing maintenance, not just one-off tasks for issue resolution.
>
> Several concerns were shared across reviewers. To reduce redundancy and detail new experiments thoroughly, we provide three [General Responses](https://github.com/emagedoc/CodeClash/tree/main/updates) that address these common points. Where relevant, we reference them below.
>
> ---
>
> **> W1: Insufficient evidence for the reliability of the LM judge assessments (groundedness, validation, hallucination).**
>
> We agree and aim to address this with [General Response #2](https://github.com/emagedoc/CodeClash/blob/main/updates/general_response_2.md).
> Three authors annotated 100 trajectories with labels corresponding to the three questions presented in Figure 8 (Section 5.2).
> Inter-annotator agreement (Fleiss' kappa 0.67–0.77) and human-vs-GPT5 agreement (Cohen's kappa 0.74–0.85) both fall in the "substantial" to "almost perfect" ranges, validating the reliability of these assessments.
>
> ---
>
> **> W2: Arenas are still relatively small and self-contained compared to real-world software engineering environments.**
>
> This is a fair point. CodeClash arenas are programming/board games, smaller in scale than a popular GitHub repository.
>
> That said, we offer several points on CodeClash's extensibility:
>
> * In addition to the 6 test arenas, Appendix G includes 9 training arenas, highlighting extensibility to more settings.
> * Beyond games, simulators for cybersecurity ([CybORG/CAGE](https://github.com/cage-challenge/CybORG)), supply chain ([ANAC SCML](https://scml.cs.brown.edu/)), and financial markets ([ABIDES](https://github.com/abides-sim/abides)) could be adapted into CodeClash. Such settings have multiple agents competing for a high-level objective while mirroring more realistic environments.
>     * For this paper, we opted for programming games for their simplicity, as more involved settings might complicate the comprehensibility of our findings.
> * CodeClash tests an orthogonal and arguably more challenging dimension of using code. Although SWE-bench presents larger codebases, the fixes are more well specified. Despite smaller codebases, CodeClash's open-endedness makes the evaluation much more challenging than bug fixing.
>
> ---
>
> **> W3 + Q1: Little evidence that results will be stable across varying budgets, prompting, or different action interfaces. How stable are rankings under different edit budgets or interaction scaffolds?**
>
> [General Response #3](https://github.com/emagedoc/CodeClash/blob/main/updates/general_response_3.md) addresses this concern with evidence from multiple paper sections.
>
> * In brief: model rankings remain consistent between the main 15-round evaluation and [CodeClash Ladders](https://github.com/emagedoc/CodeClash/blob/main/updates/cc_ladder/report.md)' 7-round setup, the failure modes identified in Section 5.2 are strategic rather than interface-related, and the system prompt (Appendix B) is deliberately minimal.
> * The multi-agent study in Section 4.2 — where 6 models compete simultaneously in Core War using TrueSkill ratings — also produces consistent rankings despite very different competitive dynamics (48.4% lead changes vs. 18.2% in 2-player), further suggesting these reflect stable underlying capabilities.
>
> ---
>
> **> W4: The RobotRumble comparison is limited — only one arena and one expert bot.**
>
> [General Response #1](https://github.com/emagedoc/CodeClash/blob/main/updates/general_response_1.md) addresses this directly.
>
> * We introduce [CodeClash Ladders](https://github.com/emagedoc/CodeClash/blob/main/updates/cc_ladder/report.md), where models climb a ranked ladder of human expert solutions (58 for RobotRumble, 264 for Core War), advancing only when they defeat the current opponent.
> * The strongest performer, Claude Sonnet 4.6, only reaches rank 39/58 on RobotRumble and 177/264 on Core War, confirming that models still have a substantial gap to close against human experts.
>
> ---
>
> **> Q2: Do you plan to add larger, less game-like repositories or stronger human baselines?**
>
> For stronger human baselines, CodeClash Ladders (W4 above) is our direct answer.
> As discussed in W2, we show the extensibility of CodeClash with 9 training settings, and identify market simulators for various industries as a fertile ground for adapting more realistic, complex settings into CodeClash.
>
> ---
>
> **> Limitations: "There is little discussion of the overall impact on society."**
>
> We used the default ICML "Impact Statement" because CodeClash evaluates existing models on simulated settings and does not introduce new capabilities or training methods. We'd be happy to include a more extended discussion if there is consensus that this is important to address.

---

> > ### Author Rebuttal · Reviewer_5t8L · 2026-04-03
> >
> > Thank you for the rebuttal.

---

> > > ### Author Response · Authors · 2026-04-05
> > >
> > > Thank you once again for your review of our paper!

---

### Decision · Program_Chairs · 2026-04-30

**Decision:**

Accept (regular)

**Comment:**

The reviewers agreed that the paper proposes a novel benchmark for goal-directed software engineering with LLM-based agents. The extensive empirical analysis and findings of the work demonstrate both the usefulness of the benchmark and its utility for further use by the community. However, the reviewers also raised several concerns and questions in their initial reviews, particularly regarding the size of the benchmark, stylized tasks in the benchmark, the reliability of automated assessments, and the limited comparison with human experts. We want to thank the authors for their detailed responses and active engagement with the reviewers. The reviewers appreciated the responses, which helped in answering their key questions. Based on the authors' responses and follow-up discussions, the reviewers are generally positive and the recommended decision is to accept the paper. The reviewers have provided detailed feedback, and we strongly encourage the authors to incorporate this feedback when preparing an updated version of the paper.